# Repulsions instruct synaptic partner matching in an olfactory circuit

Zhuoran Li[1,3], Cheng Lyu[1,3], Chuanyun Xu[1], Ying Hu[1], David J. Luginbuhl[1], Asaf B. Caspi-Lebovic[2], Jessica M. Priest[2], Engin Özkan[2] & Liqun Luo[1✉]

Neurons exhibit extraordinary precision in selecting synaptic partners. Although cell-surface proteins (CSPs) that mediate attractive interactions between developing axons and dendrites have been shown to instruct synaptic partner matching[1,2], the degree to which repulsive interactions have a role is less clear. Here, using a genetic screen guided by single-cell transcriptomes[3,4], we identified three CSP pairs, Toll2–Ptp10D, Fili–Kek1 and Hbs/Sns–Kirre, that mediate repulsive interactions between non-partner olfactory receptor neuron (ORN) axons and projection neuron (PN) dendrites in the developing *Drosophila* olfactory circuit. Each CSP pair exhibits inverse expression patterns in the select ORN–PN partners. Loss of each CSP in ORNs led to similar synaptic partner matching deficits as the loss of its partner CSP in PNs, and mistargeting phenotypes caused by overexpressing one CSP could be suppressed by loss of its partner CSP. All CSP pairs are also differentially expressed in other brain regions. Together, our data reveal that multiple repulsive CSP pairs work together to ensure precise synaptic partner matching during development by preventing neurons from forming connections with non-cognate partners.

A fundamental question in neural development is how the vast number of neurons precisely select their synaptic partners to form functional circuits. Neural circuit wiring involves multiple coordinated developmental steps: axon guidance to target regions, dendrite patterning and synaptic partner matching followed by synaptogenesis[5–7]. Even though axon guidance and dendrite patterning can greatly reduce the number of potential partners a neuron encounters at a given time and region[8], a developing axon must select specific partners among multiple nearby non-partners[1,2]. The mechanisms by which neural systems reduce multiple candidate synaptic partners to a specific one remain poorly understood.

It is well established that axon guidance involves both attraction towards the target region and repulsion away from non-target regions[9,10]. Repulsion mediated by CSPs is also used in establishing topographic maps[11], subregion target selection[12], and dendritic and axonal self-avoidance[2]. However, most known CSPs that instruct the final steps of synaptic partner selection act through attraction. These include homophilic attraction of teneurins (Ten-m and Ten-a) in *Drosophila* olfactory and neuromuscular systems[13,14], heterophilic attractions among members of the immunoglobulin superfamily of CSPs in multiple *Drosophila* circuits[15–22], and homophilic attraction mediated by immunoglobulin[23] or cadherin[24,25] families of CSPs in the vertebrate retina. The few examples of repulsion include *Drosophila* motor axon target selection, controlled by Wnt4 from non-target muscles[26], and olfactory neuron target selection by Fish-lips (Fili) from non-cognate partners[27]. How general repulsion is utilized as a guiding force in synaptic partner matching remains to be examined.

In the *Drosophila* olfactory circuit, axons of about 50 types of ORNs form one-to-one precise synaptic connections with dendrites of 50 types of PNs in 50 glomeruli in the antennal lobe[28]. During development, PN dendrites coarsely pattern the antennal lobe first[29,30]. While extending across the antennal lobe prepatterned by PN dendrites, each ORN axon sends multiple transient branches along its trajectory. ORN axon branches that contact partner PN dendrites are stabilized and branch further, whereas the rest retract[31,32]. Since synaptic partner matching involves retraction of transient ORN axon branches in contact with non-partner PNs, we aimed to identify repulsive CSPs that might function to prevent the formation of misconnections between non-partner PNs and ORNs.

## Inverse expression of three CSP pairs

VA1d and VA1v are neighbouring glomeruli that sense distinct pheromones[33,34]. Known homophilic attraction molecules that mediate matching between synaptic partners, Ten-m and Ten-a, cannot distinguish VA1d-PNs and VA1d-ORNs from VA1v-PNs and VA1v-ORNs, as they all express Ten-m at high levels and Ten-a at low levels[13]. We hypothesized that additional CSPs are differentially expressed and instruct synaptic partner matching in these adjacent glomeruli. To identify such CSPs, we performed a genetic screen focusing on PN–ORN matching in the VA1d and VA1v glomeruli (Fig. 1a). We first analysed the existing single-cell transcriptome data for developing PNs and ORNs[3,4] at 24–30 h after puparium formation (APF), shortly before matching between ORN axons and PN dendrites occurs. We focused on CSPs (including both transmembrane and secreted proteins[35]) that are differentially expressed in VA1d-PNs and VA1v-PNs or in VA1d-ORNs and VA1v-ORNs. We identified 36 candidate genes with assistance from existing literature, including the list of top 100 CSPs enriched in

[1]Department of Biology and Howard Hughes Medical Institute, Stanford University, Stanford, CA, USA. [2]Department of Biochemistry and Molecular Biology, The Neuroscience Institute and Institute for Biophysical Dynamics, The University of Chicago, Chicago, IL, USA. [3]These authors contributed equally: Zhuoran Li, Cheng Lyu. ✉e-mail: lluo@stanford.edu

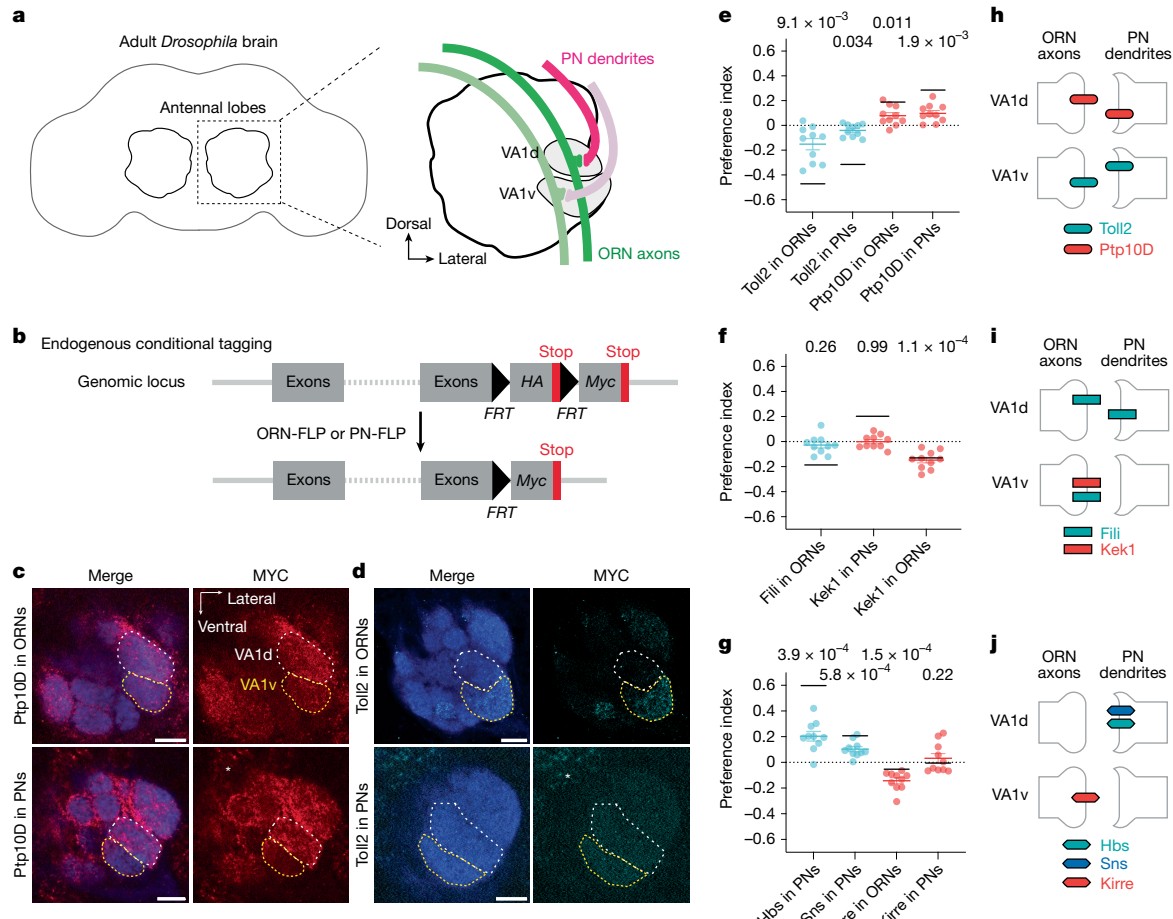

**Fig. 1 | Inverse expression of three CSP pairs in the VA1d and VA1v glomeruli.** **a**, Schematics of adult *Drosophila* brain and the antennal lobe. Axons of VA1d-ORNs and VA1v-ORNs (green) match with dendrites of VA1d-PNs and VA1v-PNs (magenta), respectively. **b**, Schematic of conditional tagging of CSPs to reveal their endogenous protein expression pattern (top) before−and in specific cell types (bottom) after−FLP-mediated recombination. **c**,**d**, Confocal images showing neuropil staining (N-cadherin, blue) and MYC staining of tagged endogenous Ptp10D (red) (**c**) and Toll2 (cyan) (**d**) using ORN-specific FLP (top) or PN-specific FLP (bottom). VA1d and VA1v glomeruli are outlined in white and yellow, respectively, based on N-cadherin staining. Example images of other proteins are shown in Extended Data Fig. 2c. Asterisks indicate PN cell bodies. Scale bars, 10 μm. **e**–**g**, Quantification of the preference index of Toll2

and Ptp10D (**e**), Fili and Kek1 (**f**) and Hbs, Sns and Kirre (**g**) mRNA (black horizontal lines) and protein (coloured data points) expression levels in ORNs and PNs that innervate VA1d versus VA1v. A preference index above 0 means that the expression level in VA1d is higher than in VA1v. For all genotypes, $n = 10$ antennal lobes. Cyan or red lines indicate geometric mean. Whiskers extend to the most extreme data points within 1.5× the interquartile range. One-sample two-side $t$-test comparing to zero; $P$ values are shown. **h**–**j**, Summary of relative expression of Toll2 and Ptp10D (**h**), Fili and Kek1 (**i**) and Hbs, Sns and Kirre (**j**) in the VA1d and VA1v PN−ORN pairs during development, based on mRNA and protein data in **e**–**g** (see Extended Data Fig. 2 for caveats). If the expression levels in VA1d- and VA1v-ORNs/PNs differ significantly, we only drew bars for the cell types with higher expression level as a simplification.

developing antennal lobes revealed by proteomic profiling[36]. We then performed tissue-specific RNA interference (RNAi) against candidate genes selectively in PNs, ORNs and/or all neurons.

In wild-type flies, axons of VA1d-ORNs and VA1v-ORNs only innervated the VA1d and VA1v glomeruli, respectively (Extended Data Fig. 1a,d). Fourteen out of the 36 candidate genes from the screen showed mistargeting phenotypes in either VA1d-ORNs or VA1v-ORNs[37] (Extended Data Fig. 1 and Extended Data Table 1; accompanying article[37]). We note that loss-of-function of single CSP usually resulted in subtle phenotypes: mistargeting of a small fraction of axons or dendrites. In the accompanying article, we show that only by simultaneously manipulating multiple CSPs in ORNs could we substantially change their matching specificity[37]. Among these candidate genes, three pairs of CSPs−Toll2−Ptp10D, Fili−Kek1 and Hbs/Sns−Kirre (Hbs−Kirre and Sns−Kirre)−exhibited largely inverse expression patterns in ORN-PN synaptic partners based on single-cell transcriptome data, particularly in ORNs and PNs that target the VA1d and VA1v glomeruli (Extended Data Fig. 2a). For example, *Toll2* is more highly expressed in VA1v-ORNs than VA1d-ORNs,

whereas its partner *Ptp10D* is more highly expressed in VA1d-PNs than VA1v-PNs (Extended Data Fig. 2a, left). Such inverse expression patterns suggest a potential role for these CSP pairs to promote repulsion during synaptic partner matching. We therefore focused the remainder of this study on these three CSP pairs.

To validate the mRNA-based inverse expression patterns (Extended Data Fig. 2a), we examined the endogenous protein expression levels at 42−48 h APF, when glomerular identities first become identifiable (and the matching between partner ORN axons and PN dendrites is mostly complete). To determine cell-type-specific expression patterns, we knocked into the endogenous loci of the three CSP pairs a modified conditional tag[36,38] (Fig. 1b). In the absence of the FLP recombinase, these proteins were tagged with haemagglutinin (HA), and no MYC signal was detected in the antennal lobe (Extended Data Fig. 2b). We found that all seven CSPs were differentially expressed across the antennal lobe (Supplementary Videos 1−7). With ORN-specific FLP or PN-specific FLP, we could visualize endogenous protein expression only in ORNs or PNs by MYC staining (Fig. 1c,d and Extended Data Fig. 2c).

In the VA1d and VA1v glomeruli, Toll2 exhibited higher expression in VA1v-PNs and VA1v-ORNs, whereas Ptp10D had higher expression in VA1d-PNs and VA1d-ORNs (Fig. 1c–e). Similarly, Kek1 exhibited higher expression in VA1v-ORNs than in VA1d-ORNs and low expression in both VA1d-PNs and VA1v-PNs (Fig. 1f and Extended Data Fig. 2c). Fili did not show preferential expression in VA1d-ORNs and VA1v-ORNs, but is more highly expressed in VA1d-PNs than in VA1v-PNs, based on data from a previous study[27]. For the third CSP pair, Hbs and Sns were minimally expressed in ORNs and exhibited higher expression in VA1d-PNs than VA1v-PNs, whereas Kirre exhibited higher expression in VA1v-ORNs than VA1d-ORNs (Fig. 1g and Extended Data Fig. 2c). Using the same preference index to quantify the relative expression in ORNs or PNs that target VA1d versus VA1v glomerulus, we found that mRNA and protein expression patterns were mostly consistent for all three CSP pairs (Fig. 1e–g). We note that the magnitudes of differential expression for some CSPs, although significant, were modest; nevertheless, our genetic analysis below suggests that such differential expression was used to instruct synaptic partner matching.

In summary, on the basis of the relative expression levels of mRNAs and proteins, the Toll2–Ptp10D, Fili–Kek1 and Hbs/Sns–Kirre pairs are expressed in inverse patterns in PN–ORN partners at the VA1d and/or VA1v glomeruli (Fig. 1h–j). Furthermore, all these CSPs are present at the terminals of ORN axons and/or PN dendrites at the nascent glomeruli, consistent with a role in synaptic partner matching.

## Loss of Toll2 or Ptp10D disrupts matching

We first examined the function of Toll2 and Ptp10D in PN–ORN synaptic partner matching. Ptp10D is an evolutionarily conserved member of the type III receptor tyrosine phosphatase family (Fig. 2a) that is involved in axon guidance at the midline, tracheal tube formation and cell competition, and was reported to be a receptor for the CSP Sas[39–43]. However, single-cell transcriptomic data indicate that Sas is minimally expressed in the antennal lobe[3,4], suggesting the existence of additional Ptp10D-interacting CSPs.

To validate the *Ptp10D* RNAi phenotypes from our screen (Extended Data Fig. 1b), we labelled VA1d-ORN axons in *Ptp10D* hemizygous mutant flies and observed similar phenotype as pan-ORN *Ptp10D* RNAi: VA1d-ORN axons mistargeted to the VA1v glomerulus (Fig. 2b,c,k). Given the high Ptp10D expression in VA1d-ORNs (Fig. 1h), we tested whether Ptp10D is autonomously required in VA1d-ORNs for their axon targeting. Knocking down *Ptp10D* using a VA1d-ORN-specific-GAL4 driver[32] and multiple RNAi lines caused similar mistargeting of VA1d-ORN axons to the VA1v glomerulus and mismatching with VA1v-PN dendrites (Fig. 2d,k, Extended Data Fig. 3a,b and Extended Data Table 2). Additional experiments argued against Ptp10D mediating homophilic attraction (Extended Data Fig. 3f,g). Furthermore, using a sparse VA1d-ORN GAL4 driver (Extended Data Fig. 4a–d) to knock down *Ptp10D* in single VA1d-ORN also caused axon branches to mistarget to the VA1v glomerulus (Fig. 2e,k), indicating that Ptp10D acts cell-autonomously in VA1d-ORNs to prevent their mismatching with VA1v-PNs.

On the basis of the inverse expression pattern of Ptp10D and Toll2 (Fig. 1h), we tested whether Toll2 has a similar role in VA1d-ORN axon targeting. Toll2 is a single-pass transmembrane protein belonging to the Toll-like receptor family, with leucine-rich repeats (LRRs) extracellularly and a Toll/interleukin-1 receptor domain intracellularly (Fig. 2a). Toll2 has an evolutionarily conserved roles in innate immunity and regulate tissue morphogenesis[44–46], but its role in neural development is unclear. We found that both pan-PN RNAi-mediated knockdown of *Toll2* and *Toll2* heterozygous mutation caused mistargeting of VA1d-ORN axons to the VA1v glomerulus (Fig. 2f,k and Extended Data Fig. 1c).

Given the high expression of Toll2 in VA1v-PNs (Fig. 1h), we hypothesized that Toll2 in VA1v-PN dendrites sends a trans-cellular repulsive signal to VA1d-ORN axons to prevent misconnection between them. To manipulate Toll2 specifically in VA1v-PNs, we identified a VA1v-PN

driver that labels VA1v-PNs across developmental stages (Extended Data Fig. 4e–i). Indeed, *Toll2* knockdown in VA1v-PNs caused VA1d-ORN axon mistargeting to the VA1v glomerulus (Fig. 2g,k), phenocopying *Ptp10D* knockdown in VA1d-ORNs. Thus, Toll2 acts in VA1v-PNs whereas Ptp10D acts in VA1d-ORNs to prevent misconnections between VA1d-ORN axons and VA1v-PN dendrites (Fig. 2m).

Since Ptp10D and Toll2 were also highly expressed in VA1d-PNs and VA1v-ORNs, respectively (Fig. 1h), we examined whether they are similarly required for preventing mismatching between VA1v-ORNs and VA1d-PNs. Indeed, cell-type-specific knockdown of *Ptp10D* in VA1d-PNs and Toll2 in VA1v-ORNs caused similar phenotypes: VA1d-PN dendrites mistargeted to the VA1v glomerulus (Fig. 2h–j,l and Extended Data Fig. 3c). Conversely, no mistargeting phenotype was observed in VA1d- or VA1v-ORN axons when *Toll2* was knocked down in ORNs (Extended Data Fig. 3d,e), suggesting that Toll2 does not function cell-autonomously in VA1d-ORNs or VA1v-ORNs and does not mediate axon-axon interactions between VA1d-ORNs and VA1v-ORNs. No mistargeting phenotype was observed when knocking down *Toll2* or *Ptp10D* where they had low expression (Extended Data Table 2). Together, these data suggest that both Ptp10D and Toll2 act in both PNs and ORNs to prevent mismatching between non-partners, with Toll2 sending and Ptp10D receiving a repulsive signal to non-partner neurons (Fig. 2m).

## Toll2 and Ptp10D interact trans-cellularly

We next tested whether Toll2 and Ptp10D work together to prevent mismatching between non-partner PNs and ORNs via trans-cellular interactions. To do so, we first overexpressed Toll2 specifically in VA1d-ORNs, where the endogenous Toll2 level is low. This caused some of their partner VA1d-PN dendrites (with high Ptp10D) to mismatch with DA4l-ORN axons (with low Toll2) (Fig. 3a,b,e and Extended Data Fig. 5b); the same manipulation did not cause mistargeting phenotype in VA1d-ORN axons or VA1v-PN dendrites (Extended Data Fig. 5a and Extended Data Table 2). This result supports the repulsion hypothesis: misexpressed Toll2 in VA1d-ORN axons sent a trans-cellular signal to repel the partner VA1d-PN dendrites away from them. Similarly, over-expressing Ptp10D specifically in VA1v-ORNs (whose synaptic partner VA1v-PNs expressed high Toll2) caused their axons to mismatch with VA1d-ORNs (with low Toll2; Extended Data Fig. 5c).

Next, we tested whether the Toll2 repulsive signal was received by Ptp10D, which was highly expressed in VA1d-PN dendrites. We combined Toll2 overexpression in VA1d-ORNs with loss of Ptp10D. The mistargeting level of VA1d-PN dendrites to the DA4l glomerulus was significantly reduced in *Ptp10D* hemizygous mutant flies (Fig. 3d,e). *Ptp10D* hemizygosity itself did not cause VA1d-PN dendrite mistargeting to DA4l, even though some VA1d-PN dendrites mistargeted to VA1v (Fig. 3c,e). This suppression indicates that Ptp10D is necessary to mediate the Toll2 overexpression phenotype, and thus Toll2 and Ptp10D function together to mediate repulsion.

As Ptp10D showed high expression in both VA1d-PNs and VA1d-ORNs, the experiments above did not distinguish whether the suppression by Ptp10D knockout was a result of *cis-* or *trans*-interaction between Toll2 and Ptp10D, or a result of loss of Ptp10D in glomeruli other than VA1d. To distinguish between these possibilities, we overexpressed Toll2 in VA1d-ORNs and knocked down *Ptp10D* in VA1d-PNs simultaneously using two orthogonal binary expression systems. In wild-type flies, dual-labelled VA1d-ORN axons and VA1d-PN dendrites largely intermingled with each other (Fig. 3f,j). Overexpressing Toll2 in VA1d-ORNs caused VA1d-PN dendrites to segregate from VA1d-ORN axons within the VA1d glomerulus and mistarget to the nearby DC3 glomerulus (Fig. 3g,j). Simultaneous knockdown of *Ptp10D* in VA1d-PNs and overexpression of Toll2 in VA1d-ORNs suppressed the VA1d-PN dendrite phenotypes caused by Toll2 overexpression alone (Fig. 3i,j), whereas *Ptp10D* knockdown in VA1d-PNs alone did not cause a similar

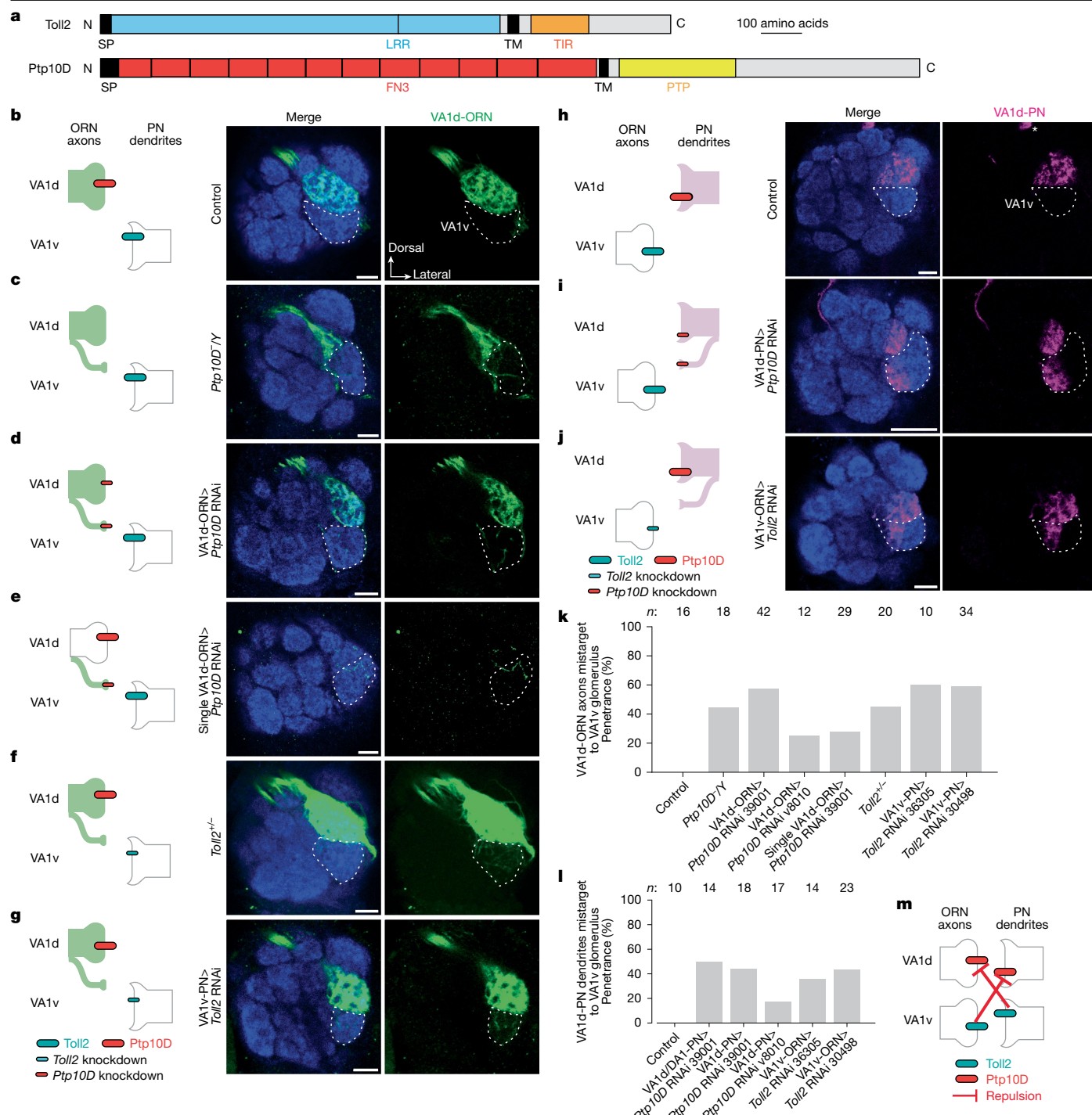

**Fig. 2 | Loss of Ptp10D or Toll2 causes similar mismatching between non-partner PNs and ORNs. a**, Domain composition of Toll2 and Ptp10D. FN3, fibronectin type III; PTP, protein tyrosine phosphatase domain; TM, transmembrane domain; TIR, Toll/interleukin-1 receptor; SP, signal peptide. **b–j**, Left, experimental schematic. Right, confocal images of adult antennal lobes showing neuropil staining (N-cadherin, blue) and VA1d-ORN axons (green) (**b–g**) or VA1d-PN dendrites (magenta) (**h–j**). VA1v is outlined based on N-cadherin staining. Scale bars, 10 μm. **b**, Control VA1d-ORN axons innervate VA1d. **c–g**, Some VA1d-ORN axons mistarget to VA1v in *Ptp10D* hemizygous mutant

flies (**c**), or in flies expressing *Ptp10D* RNAi in all (**d**) or individual (**e**) VA1d-ORNs, in *Toll2* heterozygous mutant flies (**f**) or in flies expressing *Toll2* RNAi in VA1v-PNs (**g**). **h–i**, Control VA1d-PN dendrites only innervate VA1d (**h**). Some VA1d-PN dendrites mistarget to VA1v when *Ptp10D* RNAi was expressed in VA1d-PNs (**i**) or when *Toll2* RNAi was expressed in VA1v-ORNs (**j**). Asterisks indicate PN cell bodies. **k,l**, Penetrance of the mistargeting phenotypes in **b–g** (**k**) and **h–j** (**l**). *n* refers to the total number of antennal lobes examined. **m**, Schematic summary for the function of Ptp10D and Toll2. Inhibition arrow indicates repulsive signalling from sender to receiver.

phenotype (Fig. 3h,j). Together with the inverse expression of Toll2 and Ptp10D and their similar loss-of-function phenotypes, these trans-cellular interaction data support a model in which Toll2 sends and Ptp10D receives a repulsive signal to prevent matching between non-partner ORNs and PNs (Fig. 2m).

## Non-partner repulsion by Fili–Kek1

To study the function of the other CSP pairs, we performed similar loss-of-function and suppression experiments as with Toll2–Ptp10D. A previous study showed that when *Fili* is knocked out or knocked

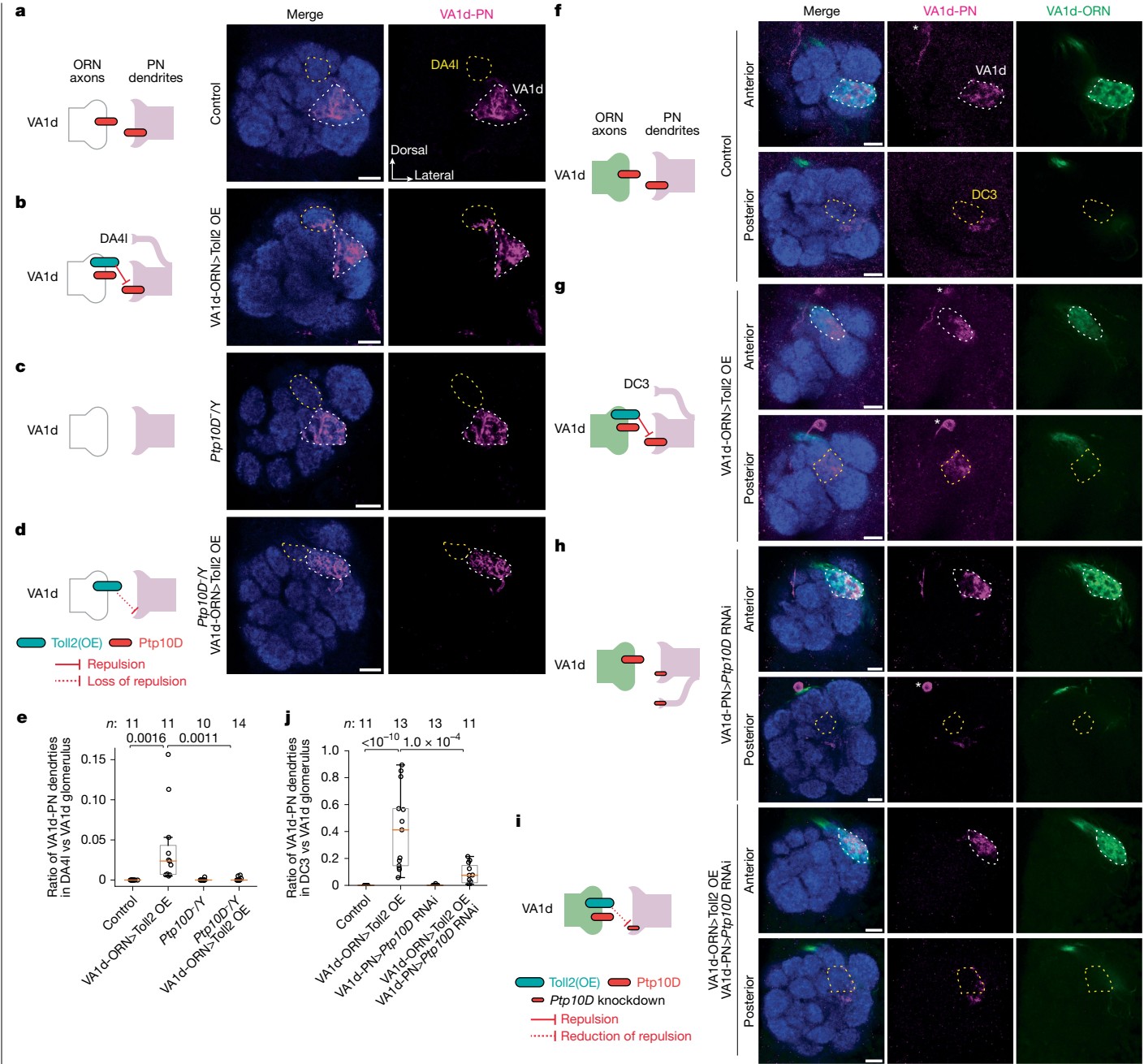

**Fig. 3 | Toll2–Ptp10D promote trans-cellular repulsive interactions.**
**a**–**d**, Left, experimental schematic. Right, confocal images showing neuropil staining (N-cadherin, blue) and VA1d-PN dendrites (magenta). VA1d (white dashed lines) and DA4l (yellow dashed lines) are outlined based on N-cadherin staining. OE, overexpression. Scale bars, 10 μm. **a**, Control VA1d-PN dendrites only innervate VA1d. **b**, Some VA1d-PN dendrites mistarget to DA4l following Toll2 overexpression in VA1d-ORNs. **c**, No VA1d-PN dendrites mistarget to DA4l in *Ptp10D* hemizygous mutant flies. **d**, Almost no VA1d-PN dendrites mistarget to DA4l when Toll2 is overexpressed in VA1d-ORNs of *Ptp10D* hemizygous mutant flies. **e**, Mistargeting ratio of VA1d-PN dendrites in the DA4l versus VA1d glomerulus for **a**–**d**. *P* values are shown. **f**–**i**, Same as **a**–**d** except that VA1d-ORN axons (green) are also visualized. Two optical sections of each antennal lobe are shown, with VA1d and DC3 outlined in white and yellow based on N-cadherin staining. Asterisks indicate PN cell bodies. Scale bars, 10 μm. **f**, In the control, VA1d-PN dendrites only innervate VA1d and fully overlap with VA1d-ORN axons.

**g**, VA1d-PN dendrites overlap less with VA1d-ORN axons within VA1d and mistarget to DC3 following Toll2 overexpression in VA1d-ORNs. **h**, No VA1d-PN dendrites mistarget to DC3 when *Ptp10D* RNAi was expressed in VA1d-PNs. **i**, VA1d-PN dendrites overlap more with VA1d-ORN axons and mistarget less to DC3 glomerulus when Toll2 overexpression in VA1d-ORNs combines with *Ptp10D* knockdown in VA1d-PNs. The different mistargeting regions in **b**,**e** and **g**,**j** are likely to result from different Toll2 overexpression levels in the different binary systems. DA4l-ORNs and DC3-ORNs express low levels of Toll2, consistent with our repulsion model. **j**, Quantification of the mistargeting ratio of VA1d-PN dendrites in DC3 versus VA1d represented in **f**–**i**. *P* values are shown. *n* refers to total antennal lobes examined. Boxes in **e**,**j** indicate geometric mean and 25th to 75th centiles and whiskers extend to the most extreme data points within 1.5× the interquartile range. Kruskal–Wallis test with Bonferroni's multiple comparison.

down in VA1d-PNs and DC3-PNs (VA1d/DC3-PNs), VA1v-ORN axons mistarget to the VA1d glomerulus whereas VA1d-ORN axons and VA1d/DC3-PN dendrites are unaffected. This result suggests that Fili is

required in VA1d/DC3-PNs to prevent mistargeting of VA1v-ORN axons to the VA1d glomerulus[27]. However, the CSP partner of Fili partner remained unknown. Kek1 was a top candidate on the basis of its high

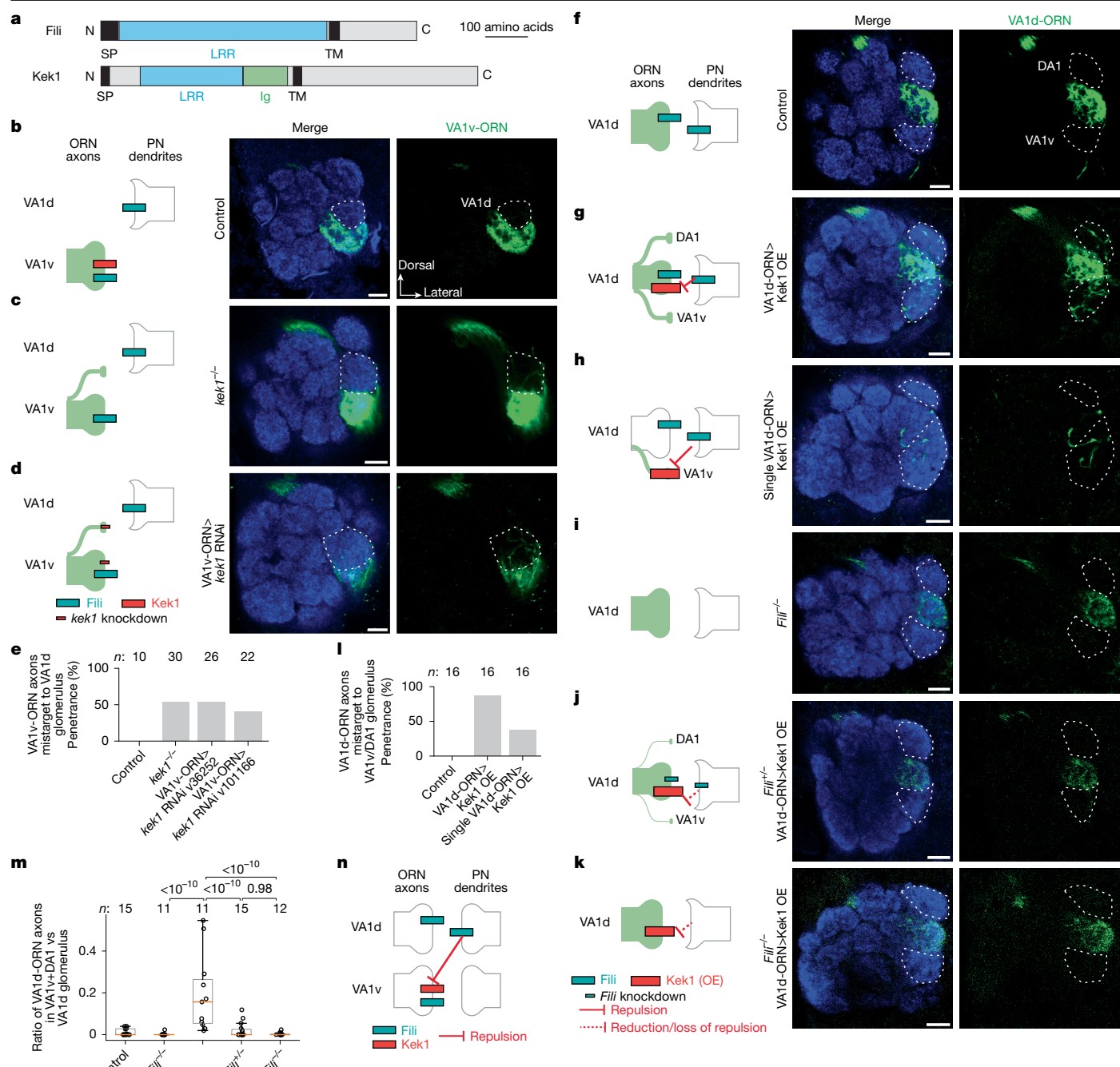

**Fig. 4 | Repulsive Fili–Kek1 interactions. a**, Domain composition of Fili and Kek1. Ig, immunoglobulin domain. **b–d**, Left, experimental schematic. Right, confocal images showing adult antennal lobe neuropil (N-cadherin, blue) and VA1v-ORN axons (green). VA1d is outlined based on N-cadherin staining. Scale bars, 10 μm. **b**, Control VA1v-ORN axons only innervate VA1v ventral to VA1d. **c,d**, Some VA1v-ORN axons mistarget to VA1d in *kek1* mutant flies (**c**) or with *kek1* RNAi expression in VA1v-ORNs (**d**). **e**, Penetrance of the mistargeting phenotypes in **b–d**. *n* refers to the total antennal lobes examined and is shown above the bars. **f–k**, Same as **b–d**, except VA1d-ORN axons (green) are visualized. DA1 and VA1v glomeruli are outlined based on N-cadherin staining. Scale bars, 10 μm. **f**, Control VA1d-ORN axons only innervate VA1d. **g,h**, Some VA1d-ORN axons mistarget to DA1 and VA1v following Kek1 overexpression in all (**g**) or individual (**h**) VA1d-ORNs. **i**, No VA1d-ORN axons mistarget to the DA1 or VA1v glomeruli in *Fili* mutant. **j,k**, Almost no VA1d-ORN axons mistarget to the DA1 and VA1v glomeruli when Kek1 overexpression in VA1d-ORNs was performed in *Fili* heterozygous (**j**) or homozygous (**k**) mutant. **l**, Penetrance of the mistargeting phenotypes in **f–h**. *n* refers to total antennal lobes examined and is shown above the bars. **m**, Quantification of the mistargeting ratio of VA1d-ORN axons in DA1 and VA1v glomeruli versus VA1d glomerulus. *n* refers to total antennal lobes examined. Boxes indicate geometric mean and 25th to 75th centiles and whiskers extend to the most extreme data points within 1.5× the interquartile range. Kruskal–Wallis test with Bonferroni's multiple comparison. *P* values are shown. **n**, Schematic summary for the function of Kek1 and Fili.

expression in VA1v-ORNs (Fig. 1i and Extended Data Fig. 2) and mistargeting of VA1v-ORN axons to the VA1d glomerulus caused by pan-ORN knockdown of *kek1* (Extended Data Fig. 1e). Kek1 and Fili both contain LRRs in their extracellular domain (Fig. 4a). Kek1 inhibits epidermal growth factor receptor (EGFR) activity through the LRRs during eye development[47] and it is expressed in developing CNS, but its function is poorly defined[48]. We found that homozygous deletion of *kek1* or *kek1* knockdown in VA1v-ORNs caused VA1v-ORN axons to mistarget

to the VA1d glomerulus (Fig. 4b–e), phenocopying the loss of Fili in VA1d/DC3-PNs[27].

To further investigate whether Fili and Kek1 work together to prevent misconnections between non-partner PNs and ORNs, we overexpressed Kek1 specifically in VA1d-ORNs, whose synaptic partner VA1d-PNs expressed high Fili[27]. This caused VA1d-ORN axons to mistarget to the neighbouring VA1v and DA1 glomeruli (Fig. 4f,g,m), whose PN dendrites mostly do not express Fili[27]. To test whether the Kek1 overexpression phenotype was caused by its interaction with Fili, we overexpressed Kek1 in *Fili* mutant flies (Fig. 4i,m). The overexpression phenotype was much less severe in *Fili* heterozygous mutant flies (Fig. 4j,m) and was nearly fully suppressed in *Fili* homozygous mutant flies (Fig. 4k,m). In addition, overexpressing Kek1 in single VA1d-ORN using the sparse driver produced a similar mistargeting phenotype (Fig. 4h,l), indicating that Kek1 acts cell-autonomously. Although VA1d-ORNs and VA1v-ORNs also expressed Fili (Fig. 1i), a previous study showed that Fili knockout in ORNs does not cause any mistargeting phenotype of VA1d-ORN and VA1v-ORN axons[27]. Together, these data suggest that Fili and Kek1 work together to prevent the misconnections between non-partner PNs and ORNs, with Fili sending and Kek1 receiving the trans-cellular repulsive signal (Fig. 4n).

## Non-partner repulsion by Hbs/Sns–Kirre

Hbs/Sns and Kirre are members of an evolutionarily conserved family of immunoglobulin ligand–receptor pairs with conserved binding sites[49] (Fig. 5a). *Drosophila* Hbs/Sns and Kirre regulate myoblast fusion[50], nephrocytes functions[51] and neural circuit wiring[52]. Whereas previous studies have suggested that this ligand–receptor pair (and sometimes homophilic interactions) mediates attraction[53,54], the expression patterns in the fly olfactory system (Fig. 1j) and RNAi phenotypes (Extended Data Fig. 1f–h) raised the possibility that Hbs/Sns and Kirre might also mediate repulsion. We found that *kirre*, *hbs* or *sns* mutants, as well as VA1v-ORN-specific knockdown of *kirre* and VA1d-PN-specific knockdown of *hbs* or *sns*, all caused a similar phenotype: mistargeting of VA1v-ORNs to the VA1d glomerulus (Fig. 5b–f and Extended Data Fig. 6a–c). Thus, Kirre, Hbs and Sns are all required to prevent VA1v-ORNs from matching with VA1d-PNs, with Kirre acting in VA1v-ORNs and Hbs/Sns acting in VA1d-PNs to prevent VA1v-ORNs to mistarget to the VA1d glomerulus.

To examine whether Hbs/Sns and Kirre work together to instruct synaptic partner matching, we performed genetic interaction experiments. We first overexpressed Kirre in VA1d-ORNs, whose partner, VA1d-PNs, expressed high levels of Hbs and Sns (Fig. 1j). Kirre overexpression in all VA1d-ORNs or single VA1d-ORN led to mistargeting of some VA1d-ORN axons to the VA1v and DA1 glomeruli (Fig. 5g–i,n), whose PNs expressed low levels of Hbs and Sns, suggesting cell-autonomous function of Kirre as a repulsive receptor. Overexpressing Kirre in *hbs* mutant, *sns* mutant or *hbs*/*sns* double mutant background reduced mistargeting phenotypes of VA1d-ORN axons (Fig. 5j–l,n). None of the mutants alone had any VA1d-ORN axon mistargeting phenotype in the absence of Kirre overexpression (Extended Data Table 2). Since Kirre can also mediate homophilic binding[49], we tested whether Kirre homophilic attraction is responsible for the mistargeting phenotype we observed. Overexpressing Kirre specifically in VA1d-ORNs caused the same phenotype in *kirre* hemizygous mutant flies as in the wild type (Fig. 5m,n), arguing against a contribution for trans-cellular Kirre homophilic interaction in synaptic partner matching. Together, these data support that heterophilic repulsion between Hbs/Sns as the ligands and Kirre as the receptor prevents VA1v-ORNs from mistargeting to the VA1d glomerulus (Fig. 5o).

As the genetic interactions of Fili–Kek1 and Hbs/Sns–Kirre both functions to prevent misconnection of VA1v-ORNs with VA1d-PNs, we tested whether there is crosstalk between these interactions. We found

that knocking down *sns* and *hbs* did not suppress the mistargeting phenotype caused by Kek1 overexpression (Extended Data Fig. 6d,e), suggesting that Fili–Kek1 and Hbs/Sns–Kirre are likely to act in distinct pathways, and validating the specificity of these genetic suppression experiments. In an additional set of experiments, we co-expressed Bruchpilot-Short, a presynaptic active zone marker, and found that it was enriched in mistargeted ORN axons when we perturbed each of the three CSP pairs (Extended Data Fig. 7). These data suggest that mistargeted ORN axons may form ectopic synaptic connections with new PN partners.

As previous biochemical and structural studies have shown direct binding between Hbs/Sns and Kirre[49], we also tested whether the other two CSP pairs that we identified directly bind each other. However, we did not detect direct binding between Fili and Kek1, or between Toll2 and Ptp10D, in in vitro (Extended Data Fig. 8) or tissue-based (Extended Data Fig. 9) binding assays. Thus, the biochemical basis for the repulsive interactions mediated by Fili–Kek1 and Toll2–Ptp10D remains an open question. Possibilities include requirements for unidentified co-factor(s), post-translational modifications or specific physiological conditions such as multimerization[55,56] that were not recapitulated in our binding assays.

## Broad usage of the three CSP pairs

Our results (Figs. 2–5) and additional control experiments (Extended Data Table 2) suggest that the three repulsive CSP pairs could prevent mismatching between VA1d-ORNs with VA1v-PNs (Toll2–Ptp10D), VA1d-PNs with VA1v-ORNs (Toll2–Ptp10D), and VA1v-ORNs with VA1d-PNs (Fili–Kek1 and Hbs/Sns–Kirre) (Fig. 5o). Thus, the three CSP pairs work in concert, with partial redundancy, to ensure robust repulsion between non-matching ORNs and PNs at the VA1d and VA1v glomeruli.

The antennal lobe has 50 ORN-PN synaptic partner pairs that need to be specified, potentially requiring many CSP pairs to mediate repulsions. One way to alleviate this is to repeatedly use the same repulsive CSP pairs across the antennal lobe in a combinatorial fashion. Indeed, analysis of previously published single-cell transcriptomes during development[3] revealed that mRNAs encoding all three CSPs implicated in sending the repulsive cues—*Toll2*, *Fili* and the maximum of *hbs* and *sns*—are expressed in multiple PN types. Notably, most PN types expressed one or two, but not all three repulsive cues at high levels (Fig. 5p). For each PN type, this property reduces the errors of mismatching with non-partner ORN types (as all PN types express at least one repulsive cue at high levels) while making room for the match with its partner ORN type (as few PN type expresses all three repulsive cues at high levels).

On the basis of their expression patterns, we further investigated whether the repulsions of these three CSP pairs have similar roles in additional ORN and PN types using loss- or gain-of-function experiments. A previous study showed that Fili acts in ORNs to prevent mistargeting of VM5-PN dendrites[27]. We observed a similar mistargeting phenotype of VM5-PN dendrites when we knocked down *kek1* using a pan-PN driver (Extended Data Fig. 10a,b), suggesting that Fili in ORNs and Kek1 in PNs are both required for proper targeting of VM5-PN dendrites. As Hbs is highly expressed in DA1-PNs and DA4l-PNs, we overexpressed Kirre in DA1-ORNs or DA4l/VA1d-ORNs using available drivers with early developmental onset. We observed mistargeting of each ORN group(s) to neighbouring glomeruli, whose PNs expressed low Hbs and Sns (Extended Data Fig. 10c–e), suggesting that Hbs and Kirre mediate repulsions in these PN−ORN pairs. In the accompanying article, we show that all three repulsive CSP pairs have a key role in preventing the mismatch of both VA1d-ORNs and DA1-ORNs with PNs of nearby glomeruli, as manipulating the expression of these CSP pairs is essential in the rewiring of VA1d- and DA1-ORNs to non-cognate PNs[37]. Together, these data support a model in which these repulsive

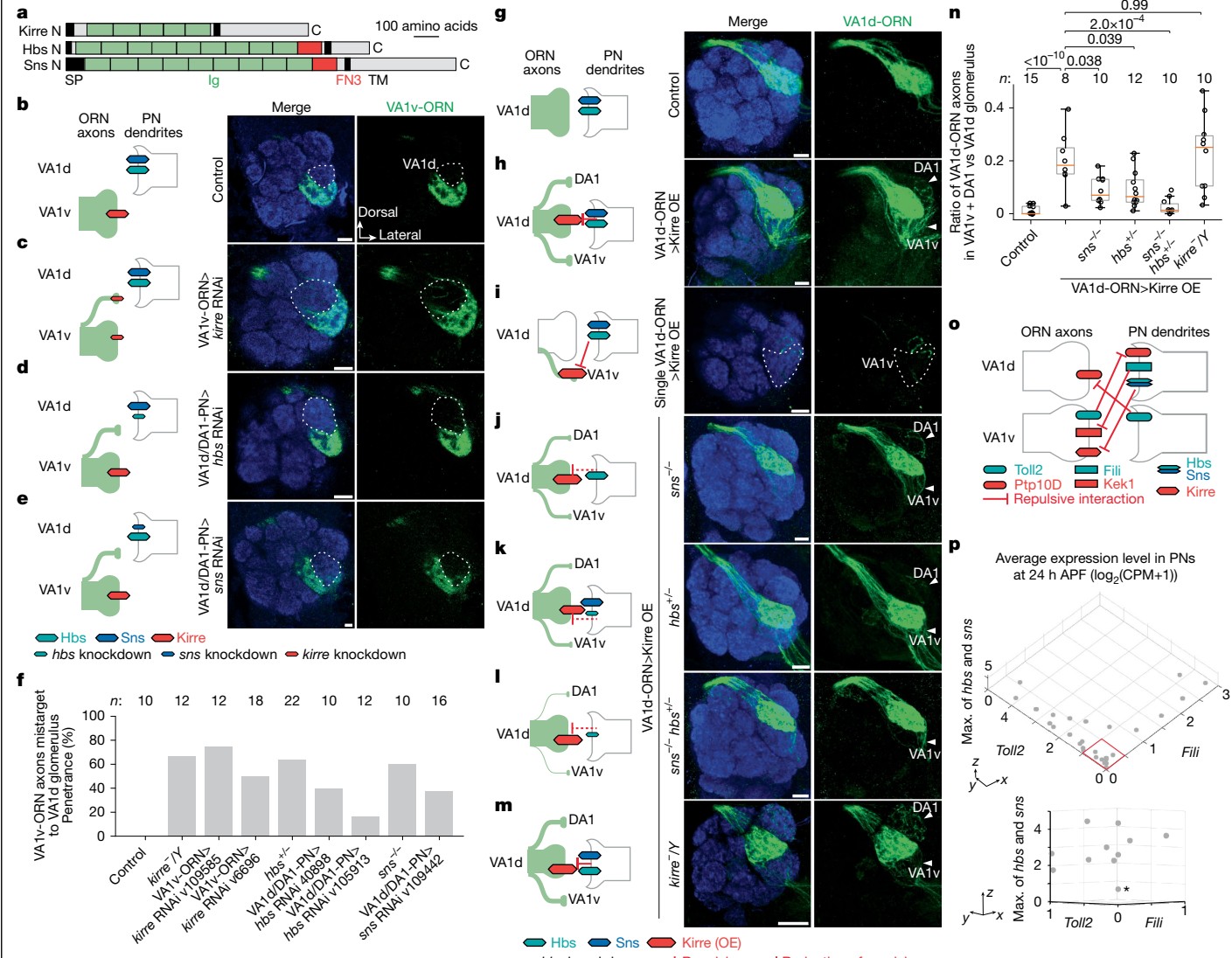

**Fig. 5 | Repulsive Hbs/Sns–Kirre interactions and combinatorial expression of three CSP pairs. a**, Domain composition of Kirre, Hbs and Sns. **b–e**, Left, experimental schematic. Right, confocal images of adult antennal lobes showing neuropil staining (N-cadherin, blue) and VA1v-ORN axons (green). VA1d is outlined based on N-cadherin staining. Scale bars, 10 μm. **b**, Control VA1v-ORN axons only innervate VA1v. **c–e**, Some VA1v-ORN axons mistarget to VA1d when VA1v-ORN axons express *kirre* RNAi (**c**), DA1/VA1d-PNs express *hbs* RNAi (**d**) or DA1/VA1d-PNs express *sns* RNAi (**e**). **f**, Penetrance of the mistargeting phenotypes in **b–e**. *n* refers to total antennal lobes examined. **g–m**, Same as **b–e** except VA1d-ORN axons (green) are visualized. Arrowheads indicate DA1 and VA1v based on N-cadherin staining. Scale bars, 10 μm. **g**, Control VA1d-ORN axons only innervate VA1d ventral to DA1 and dorsal to VA1v. **h,i**, Some VA1d-ORN axons mistarget to DA1 and VA1v when Kirre is overexpressed in all (**h**) or individual (**i**) VA1d-ORNs. **j–l**, Compared with Kirre overexpression alone, fewer VA1d-ORN axons mistarget to DA1 and VA1v when Kirre is overexpressed in VA1d-ORNs of *sns* homozygous mutant flies (**j**), *hbs* heterozygous mutant flies (**k**) or *sns* homozygous, *hbs*

heterozygous double mutants (**l**). **m**, VA1d-ORN axons still mistarget to DA1 and VA1v when Kirre is overexpressed in VA1d-ORNs of *kirre* hemizygous mutant flies. **n**, Quantification of the mistargeting ratio of VA1d-ORN axons in DA1 and VA1v glomeruli versus VA1d glomerulus. *n* refers to total antennal lobes examined. Boxes indicate geometric mean and 25th to 75th centiles and whiskers extend to the most extreme data points within 1.5× the interquartile range. Kruskal–Wallis test with Bonferroni's multiple comparison. **o**, Schematic summary of repulsive interactions of three CSP pairs that distinguish ORNs and PNs that target VA1v and VA1d glomeruli. Hbs and Sns are combined as they perform similar functions. **p**, Top, single-cell RNA-sequencing data showing the average expression levels of the non-cell-autonomous cues *Toll2*, *Fili* and the higher expression of *hbs* or *sns* in PNs at 24–30 h APF. Each dot represents a PN type. Bottom, closer view of PN types outlined in red from a different angle, showing that PN types that express low levels of *Toll2* and *Fili* tend to express high levels of *hbs* or *sns* (the asterisk highlights a possible exception). CPM, counts per million reads. Data adapted from previous studies[3,4].

interactions are used broadly in the antennal lobe to ensure synaptic partner matching specificity.

Finally, to explore the possibility that these three CSP pairs work elsewhere in the fly brain to regulate connection specificity, we used HA staining to examine the expression patterns of the 7 CSPs (Fig. 1b) during the period when many circuits are establishing wiring specificity (42–48 h APF). We found that all CSPs had broad but differential expression across the *Drosophila* brain (Supplementary Videos 1–7). For example, in the optic lobe, all CSPs showed differential expressions in

specific neuropil layers (Extended Data Fig. 11a,c). To better examine the endogenous expression pattern of CSP pairs in brain regions without layer structures, we also produced transgenic flies with endogenous CSPs tagged with V5 and co-labelled each pair of CSPs in the same brain. Extended Data Fig. 11b showcased the differential expression of these CSP pairs in the suboesophageal zone, ellipsoid body and protocerebrum. These data support the notion that combinatorial repulsive interactions serve as a generalizable mechanism in instructing wiring specificity of neural circuits.

## Discussion

Previous high-throughput extracellular interactome screenings in vitro have identified novel molecular pairs with direct interactions, including Dpr/DIP and Beat/Side families of immunoglobulin-containing CSPs whose in vivo functions in neuronal wiring have subsequently been validated[15–17,20–22,57,58]. Here we took an alternative approach of using transcriptome-informed in vivo genetic screens, which enabled us to identify known binding partners (Hbs/Sns–Kirre) as well as proteins that may not interact directly (Fili–Kek1 and Toll2–Ptp10D). Thus, transcriptome-informed in vivo screening complements in vitro biochemical approach to identify CSP pairs that mediate trans-cellular interactions in neuronal wiring.

The inverse expression patterns of the three CSP pairs that we identified, their cell-type-specific loss-of-function phenotypes, and suppression assays strongly suggest that they mediate repulsion between non-partner PNs and ORNs. Some of the CSPs only show modest differential expression, suggesting that synaptic partner choice is likely to be regulated by the relative levels of repulsive interactions. We note that orthologues of Hbs/Sns and Kirre in other species control synaptic site choice in *C. elegans*[53] and axon sorting in the mouse olfactory bulb[54] via heterophilic and homophilic attraction, respectively. However, our results suggest they instruct synaptic partner matching in the fly olfactory system via heterophilic repulsion. These different mechanisms could potentially be mediated by engaging distinct intracellular signalling pathways in specific cellular context.

Repulsion could be combined with attraction to enhance the selection process of synaptic partners. For example, for neuron A to match its synaptic partner A' but not non-partners, one strategy is to express attractive CSP pairs in A and A'. However, as the CSP number on each synaptic partner increases, an attraction-only strategy can cause ambiguity (for example, to distinguish the matching of two versus three attractive pairs), and the addition of repulsion can reduce errors. Repulsion can also increase the searching efficiency by ruling out non-partners during the simultaneous searching process, as in the case of ORN-PN matching[31,32]. Conversely, a repulsion-only strategy may have difficulty exploring a larger space owing to excessive branch retraction. In the accompanying article, we showed that only by simultaneously manipulating both attractive and repulsive CSPs in ORNs could we substantially switch their partner PNs[37]. Thus, attractive and repulsive interactions work in concert to ensure precise synaptic partner matching.

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

## Methods

### Fly husbandry and stocks

*Drosophila melanogaster* were reared on a standard cornmeal medium at 25 °C under a 12 h:12 h light:dark cycle. To enhance transgene expression levels, flies from all genetic perturbation experiments, including control groups, were shifted to 29 °C shortly before puparium formation. Both male and female flies were used. The *w[1118]* strain was used as wild type, with ages ranging from larva stage to seven days old adults. Detailed genotypes for each experiment are listed in Supplementary Table 1.

### Generation of cDNA constructs and transgenic flies

Complementary DNA (cDNA) encoding proteins used in this study were obtained from different resources. *Toll2* cDNA was amplified from the cDNA library of *w1118* pupal brain extracts using Q5 hot-start high-fidelity DNA polymerase (New England Biolabs) as previously described[32]; *Ptp10D* cDNA was amplified from clone RE52018 (DGRC 9073; https://dgrc.bio.indiana.edu//stock/9073; RRID:DGRC_9073); *Fili* cDNA was amplified from pUAST-attB-SP-V5-Fili-Flag plasmid[27]; *kek1* cDNA was amplified from clone GH23277 (DGRC 1263019; https://dgrc.bio.indiana.edu//stock/1263019; RRID:DGRC_1263019); *kirre* cDNA was amplified from genomic DNA extraction of *UAS-kirre.C-HA* fly[59] (RRID:BDSC_92196) using DNeasy blood and tissue kit (QIAGEN). Sequence-verified coding regions were assembled into pUAST-attB-mtdT-3xHA[60], pUAST-attB-SP-V5-Fili-Flag[32] or pJFRC19-13XLexAop2-IVS-myr::GFP (Addgene plasmid #26224) backbones using the NEBuilder HiFi DNA assembly master mix (New England Biolabs) to generate pUAST-attB-Toll2-Flag, pUAST-attB-Ptp10D-3xHA, pUAST-attB-kek1-3xHA, pUAST-attB-kirre-3xHA, and pJFRC19-13XLexAop2-IVS-Toll2-Flag plasmids. Transgenic flies for overexpression experiments were generated by BestGene with microinjection of plasmids into the *VK5* site.

### Generation of conditional tags

Endogenous conditional tag flies were generated using CRISPR knock-in with modifications from a previous strategy[38]. To increase the efficiency of knock-in, we incorporated the short repair templates flanked by guide RNA (gRNA) target sites[61] and the gRNA into a single plasmid TOPO-HR1-FRT-3xHA-Stop-FRT-3xMyc-Stop-loxP-mCherry-loxP-HR2-pU6-gRNA or TOPO-HR1-FRT-V5-Stop-FRT-Flag-Stop-loxP-mCherry-loxP-HR2-pU6-gRNA. In the plasmid, *HR1* and *HR2* are the 150-bp genomic sequences of upstream and downstream of the target genes stop codon, respectively; gRNA sequences were designed by the flyCRISPR Target Finder tool that targeting stop codons of the genes, and were cloned into the backbone of pU6-BbsI-chiRNA vector[62,63] to make the pU6-gRNA. The plasmids were synthesized by Synbio Technologies and were microinjected in house into *nos-Cas9* flies[64]. All *mCherry+* progenies were individually balanced and the *loxP*-flanked *mCherry* cassettes were then removed by crossing each line to balancer expressing Cre (Bloomington *Drosophila* Stock Center, RRID:BDSC 1092). To detect cell-type-specific expression level, we used *ey-Flp*[65] for ORNs and *VT033006-GAL4;UAS-FLP*[66] for PNs.

### Generation of the sparse driver and single-neuron genetic manipulations

The *FRT100-Stop-FRT100* element was cloned into the 78H05-p65AD plasmid (in backbone pBPp65ADZpUw) to generate plasmid 78H05-FRT100-Stop-FRT100-p65AD as previously reported[67]. The plasmid was integrated into the *VK27* site. To perform the sparse genetic manipulations, flies including VA1d-ORN sparse driver (31F09-GAL4DBD, 78H05-FRT100-Stop-FRT100-p65AD), *hsFLP* (heat shock protein promoter-driven FLP), reporter (*UAS-mCD8-GFP*), and knockdown or overexpression transgenes were raised at 29 °C.

To induce sparse manipulation, the flies were collected at 0–6 h APF and heat shocked for 1–2 h in a water bath[67] at 37 °C.

### Immunostaining

Fly brain dissection, fixation and immunostaining were performed according to the published protocol[68]. For primary antibodies, we used rat anti-N-cadherin (1:40; DN-Ex#8, Developmental Studies Hybridoma Bank), chicken anti-GFP (1:1000; GFP-1020, Aves Labs), rabbit anti-DsRed (1:500; 632496, Clontech), mouse anti-rat CD2 (1:200; OX-34, Bio-Rad), rabbit anti-HA (1:100, 3724S, Cell Signaling), mouse anti-HA (1:100, 2367S, Cell Signaling), rabbit anti-MYC (1:250, 2278S, Cell Signaling) and mouse anti-V5 (1:250, R960-25, Thermo Fisher Scientific). Donkey secondary antibodies conjugated to Alexa Fluor 405/488/568/647 or Cy3 (Jackson ImmunoResearch or Thermo Fisher) were used at 1:250. For the staining of conditional tag for Hbs and Fili in PNs, the routine protocol described above failed to detect MYC signal from the background, probably owing to low expression of endogenous proteins in vivo. Alexa 488 Tyramide SuperBoost kit (Thermo Fisher) was used to amplify the immunostaining signal by following the manufacture's protocol.

### Imaging, quantification and statistical analysis

Images were obtained using laser scanning confocal microscopy (Zeiss LSM 780 or LSM 900). Fiji was used to adjust brightness and contrast for representative images. Penetrance of phenotypes represents the percentage of antennal lobes showing a given phenotype among the total antennal lobes (two per fly) examined. To quantify the endogenous expression levels of the proteins, we manually outlined VA1d and VA1v glomeruli in Fiji based only on the N-cadherin signal (that is, blind to MYC signals), and use this as filter to calculate mean fluorescent density ($\bar{I}$) in the VA1d and VA1v glomeruli (total fluorescence intensity divided by the volume). The preference index is calculated in Python by $(\bar{I}_{\text{VA1d}} - \bar{I}_{\text{VA1v}})/(\bar{I}_{\text{VA1d}} + \bar{I}_{\text{VA1v}})$. Preference index for proteins was calculated based on MYC staining intensity throughout the glomeruli at 42–48 h APF. Preference index for mRNAs was calculated based on average expression levels in VA1d-ORN/PNs versus VA1v-ORN/PNs based on the single-cell-transcriptome data[3,4] at 24–30 h APF. To quantify the mistargeting ratio of VA1d-PNs or VA1d-ORNs in the trans-cellular assay, we defined PN dendritic or ORN axonal targeting area by smoothening ('gaussian blur' with radius = 1 pixels) and thresholding (based on the algorithm Otsu) the images in Fiji. We manually outlined VA1d, VA1v, DA1, DA4l and/or DC3 glomeruli in Fiji based only on the N-cadherin signal (that is, blind to PNs or ORNs signals), and used this as a filter to calculate PN dendritic or ORN axonal targeting volume ($V$) in each glomerulus. The mistargeting ratio is calculated in Python by $V_{\text{other glomerulus}}/V_{\text{VA1d}}$.

### Binding assays

For binding assays involving purified proteins with surface plasmon resonance, we expressed and purified Fili, Kek1, Ptp10D and Toll2 extracellular domains using the baculoviral expression system in *Trichoplusia ni* High Five cells (Thermo Fisher B855-02). *Sf*9 cells from *Spodoptera frugiperda* were used for baculovirus production (Thermo Fisher, 12659017). Proteins were tagged with C-terminal hexahistidine tags for purification, and Avi-tags for biotinylation using BirA biotin ligase. Proteins were purified to homogeneity with Ni-NTA metal affinity and size-exclusion chromatography in 10 mM HEPES pH 7.2, 150 mM NaCl. Biotinylated Fili and Toll2 extracellular domains were captured on a Streptavidin sensor chip in a Biacore T200 system (Cytiva) running a buffer containing 10 mM HEPES pH 7.2, 150 mM NaCl and 0.05% Tween-20. We observed no binding responses for Kek1 and Ptp10D extracellular domains flowing on Fili and Toll2 channels, respectively.

For the avidity-based extracellular binding assay (ECIA), we followed the published protocols[69]. In brief, we expressed and secreted each

ectodomain as bait and/or prey, where they were tagged with Fc, V5 and His tags (for bait) and Alkaline phosphatase, the COMP pentameric coiled coil region, Flag and His tags (for prey). All proteins were expressed in *Drosophila* S2 cells (DGRC #6) in Schneider's medium supplemented with the insect medium supplement (Sigma). Media from bait-expressing cells were incubated with Protein A-coated 96-well plates for capturing of bait onto plates overnight at room temperature, while media from prey-expressing cells were incubated with bait-captured plates for 3 h. Washes were performed with 1× phosphate-buffered saline, supplemented with 1 mM $CaCl_2$, 1 mM $MgCl_2$ and 0.1% bovine serum albumin. Binding of prey to bait was detected using the chromogenic alkaline phosphatase substrate BluePhos (KPL) by measuring absorbance at 650 nm after a two-hour incubation. We included the known interactions of Rst dimerization[49] and EGFR–Kek1 interaction as positive controls[70]. Cell lines (from commercial sources) were not authenticated, as they were only used as an exogenous production source of protein, and not studied for any biological functions. No mycoplasma contamination was observed. The raw data for the western blots are shown in Supplementary Data 1.

The tissue-based binding assays were performed as previously described[58,71,72]. In brief, brains or wing discs were dissected in Schneider's medium, then incubated with the conditioned medium of High Five cells expressing epitope-tagged extracellular domains of a specific protein for 18 h (for pupal brains) or 1 h (for wing discs) at 4 °C on a rotating platform. Medium of High Five cells without expressing any transgenes was used as a negative control. After the incubation, brains or wing discs were washed with the Schneider's medium and fixed with 4% paraformaldehyde in 1× PBS for 30 min, followed by the immunostaining protocol above using antibodies against the epitope tags.

## Statistics and reproducibility

For the representative images in Extended Data Figs. 4b–d,h,i, 5c, 7a–c and 9a–f at least eight samples were examined with similar results. For the representative images in Extended Data Fig. 11a,b, at least three samples were examined with similar results. The western blots in Extended Data Fig. 8d were performed once and utilized the proteins directly used in the ECIA in Extended Data Fig. 8c with the hexahistidine tag common in all constructs. There was a lack of detectable expression for Ptp10D ectodomain, which may be the reason for the lack of binding of Ptp10D in Extended Data Fig. 8c.

## Animal study design

No statistical tests were used to determine sample size. We used sample sizes (~4–20 flies per condition) that been shown to have sufficient statistical power in similar experiments in the past. We did not exclude flies or data from any analysis, unless brains stained for imaging appeared unsuitable (for example, broken) at the time of imaging. All experiments discussed in the paper were conducted on multiple flies, with sample size specified. In immunostaining, data across multiple days were collected and all imaged brains showed the same qualitative pattern of staining. Organisms were not allocated to control and experimental groups by the experimenter in this work, rather the flies' genotype determined their group. Thus, randomization of individuals into treatments groups is not relevant. The investigators were not blind to the flies' genotypes. All data collection and analyses were performed computationally. During this process, data from control groups and experimental groups were analysed equally using the same well-established protocols, therefore are less prone to investigator influence.

## Reporting summary

Further information on research design is available in the Nature Portfolio Reporting Summary linked to this article.

## Data availability

All data are included in the manuscript and the supplementary materials. Source data are provided with this paper.

## Code availability

Code used in the study is available on github: https://github.com/ZhuoranLi97/repulsive_interactions.

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

**Acknowledgements** We thank C. Desplan and Y.-C. D. Chen for sharing split-GAL4 drivers; M. Baylies for sharing *hbs/sns* double mutant flies; M. Ruiz for sharing *kirre* mutant flies; Bloomington *Drosophila* Stock Center (BDRC), the Vienna *Drosophila* Resource Center (VDRC), and Fly Stocks of National Institute of Genetics (NIG) for providing fly lines; *Drosophila* Genomics Resource Center (DGRC) supported by NIH grant 2P40OD010949 and Addgene for providing plasmids; all the Luo laboratory members, especially T. Hindmarsh Sten, D. Pederick, C. McLaughlin, K. L. Wong, Y. Wu, Q. Xie, J. Hui, J. Kalai, Y. Ge and M. Molacavage; the Özkan laboratory members N. Tsang, S. Usatyuk, W. Nawrocka and I. Weathers for technical support and valuable discussions; and K. Shen, T. Clandinin, X. Gao, J. Frydman and X. Xiong for feedback on the project and manuscript. C.L. was supported by the Stanford Science Fellows Program. L.L. is an investigator of Howard Hughes Medical Institute. This work was supported by National Institutes of Health grants R01-DC005982 (L.L.) and R01-NS139060 (E.Ö.), Wu Tsai Neuroscience Institute of Stanford University (L.L.) and Gatsby Foundation (L.L.).

**Author contributions** Z.L., C.L. and L.L. conceived the project. Z.L., C.L. and L.L. planned the experiments and interpreted the results. Z.L., C.L., C.X. and Y.H. performed the in vivo experiments. D.J.L. assisted in the generation of transgenic flies. A.B.C.-L., J.M.P. and E.Ö. performed biochemical experiments. Z.L., C.L. and L.L. wrote the paper, with inputs from all other co-authors. L.L. supervised the work.

**Competing interests** The authors declare no competing interests.

**Additional information**
**Correspondence and requests for materials** should be addressed to Liqun Luo.

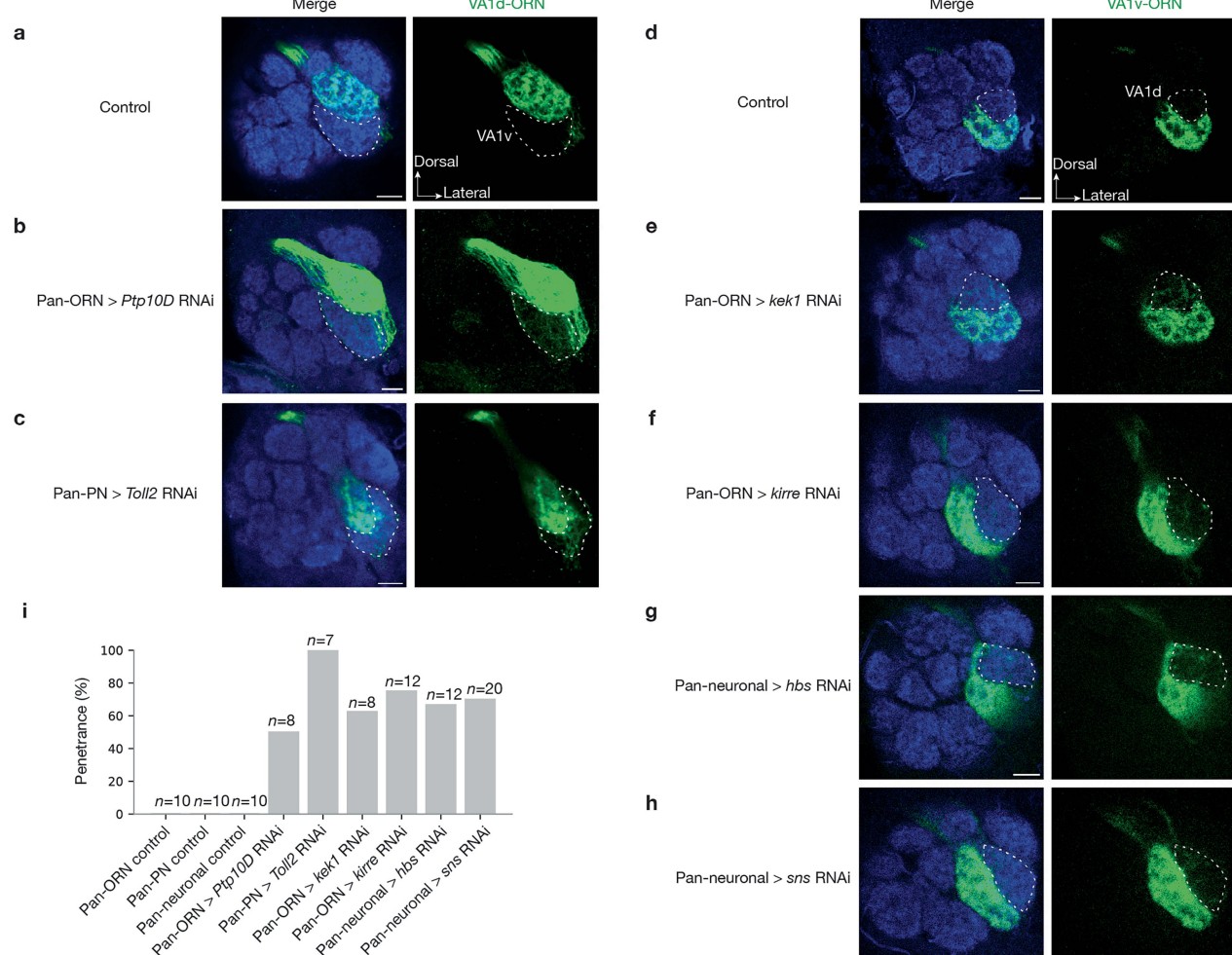

**Extended Data Fig. 1 | In vivo RNAi screen to identify CSPs required for synaptic partner matching. a–c**, Confocal images of adult antennal lobes showing neuropil staining by N-cadherin antibody (blue) and VA1d-ORN axons (green). The VA1v glomerulus is outlined based on N-cadherin staining. Control VA1d-ORN axons only innervate the VA1d glomerulus dorsal to the VA1v glomerulus (**a**). Some VA1d-ORN axons mistarget to the VA1v glomerulus in *Ptp10D* RNAi expressed from all ORNs (**b**, maximum projection of 3 sections with 1-μm interval), or *Toll2* RNAi expressed in all PNs (**c**). **d–h**, Confocal images of adult antennal lobes showing neuropil staining by N-cadherin antibody (blue) and VA1v-ORN axons (green). The VA1d glomerulus is outlined based on N-cadherin staining. Control VA1v-ORN axons only innervate the VA1v glomerulus ventral to the VA1d glomerulus (**d**). Some VA1v-ORN axons mistarget to the VA1d glomerulus in *kek1* RNAi expressed in all ORNs (**e**), *kirre* RNAi expressed in all ORNs (**f**), *hbs* RNAi expressed in all neurons (**g**), or *sns* RNAi expressed in all neurons (**h**). **i**, Penetrance of the mistargeting phenotypes in **a–h**. *n* refers to total antennal lobes examined. Scale bars = 10 μm.

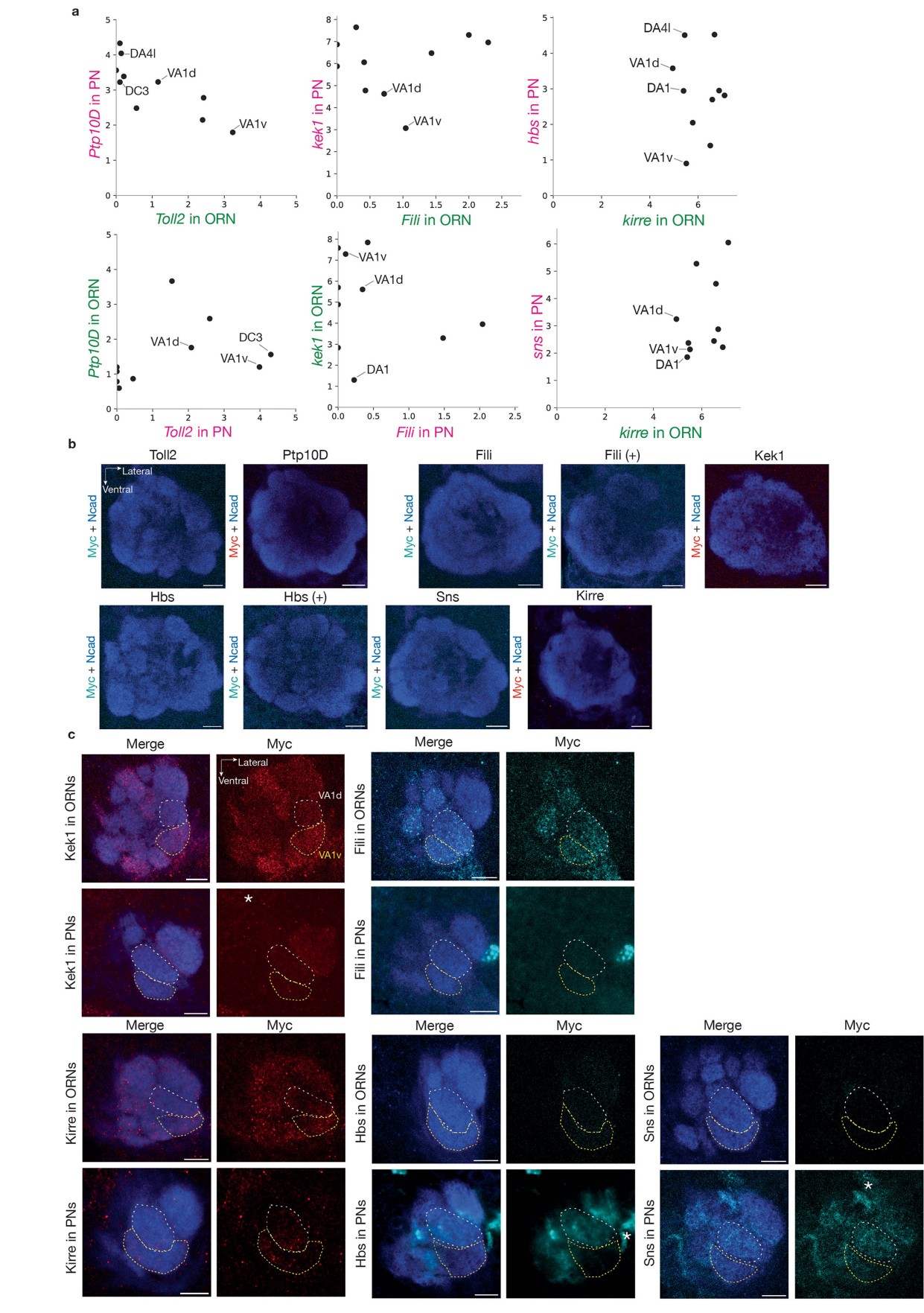

**Extended Data Fig. 2** | See next page for caption.

**Extended Data Fig. 2 | Cell-type-specific expression levels of all CSP pairs across the antennal lobe. a**, mRNA expression levels of the three CSP pairs at 24–30 h APF in ORNs and PNs that target 10 glomeruli (DA1, DA4l, DL1, DL3, DC3, VA1d, VA1v, DM2, VM3, and VA2) from single-cell RNA sequencing data[3,4]. The unit for x and y axis is $log_2$(CPM + 1). CPM: counts per million reads. Besides VA1d and VA1v, several other glomeruli relevant to experiments described in this study are indicated. **b**, Confocal images showing neuropil staining by N-cadherin antibody (blue) and lack of Myc staining of tagged endogenous CSPs (cyan for Toll2, Fili, Hbs, and Sns; red for Ptp10D, Kek1, and Kirre) without FLP. Annotation (+) indicate signal amplification for Myc staining. **c**, Confocal images showing neuropil staining by N-cadherin antibody (blue) and Myc staining of tagged endogenous CSPs (red for Kek1 and Kirre; cyan for Fili, Hbs, and Sns) using ORN-specific FLP (top tow) or PN-specific FLP (bottom row). The VA1d (white) and VA1v (yellow) glomeruli are outlined based on N-cadherin staining. Samples were chosen based on (1) their VA1d vs. VA1v preference indexes are close to the average, and (2) both VA1d and VA1v glomeruli occupy similar sizes in these single optic sections. Scale bars = 10 μm. Data from the above panels, along with data from Fig. 1c,d, contribute to the quantification of expression (Fig. 1e–g) and simplified schematic summary (Fig. 1h–j). Note that although Fili's expression in PNs was undetectable using the conditional tag, a previous study showed that Fili exhibits higher expression in VA1d-PNs than VA1v-PNs using immunostaining of Fili antibody and cell-type-specific expression pattern using intersection of ORN- or PN-FLP and *Fili-GAL4*[27]. And in DA1 glomerulus, Fili appears to be expressed in a small portion of DA1-PN dendrites neighboring the VA1d glomerulus[27]. For Kirre–Hbs/Sns, expression of Hbs and Sns in ORNs are not detectable, so we did not draw their expression in Fig. 1j. We did not draw the Kirre expression level in VA1d- or VA1v-PNs (Fig. 1j) given its preference index is highly variable (Fig. 1g). We also note that the differential expression patterns of mRNAs and proteins are largely consistent (Fig. 1e–g). Occasional discrepancies could be caused by (1) post-transcriptional regulations (e.g., protein translation, stability) and (2) different time windows from which mRNA (24–30 h APF) and protein (42–48 h APF) data were collected. Ideally, protein staining should be done around 30 h APF when synaptic partner matching initiates. However, as glomeruli have not formed at that stage, we could not distinguish cell types in which proteins are expressed. 42–48 h is the earliest window we could use glomerular identity to infer cell-type-specific expression. It is possible that the expression of some of the CSPs for synaptic partner matching is already downregulated by then. Although protein expression data is more directly relevant to the action of these genes, mRNA expression data is more temporally relevant, and thus these data provide complementary information.

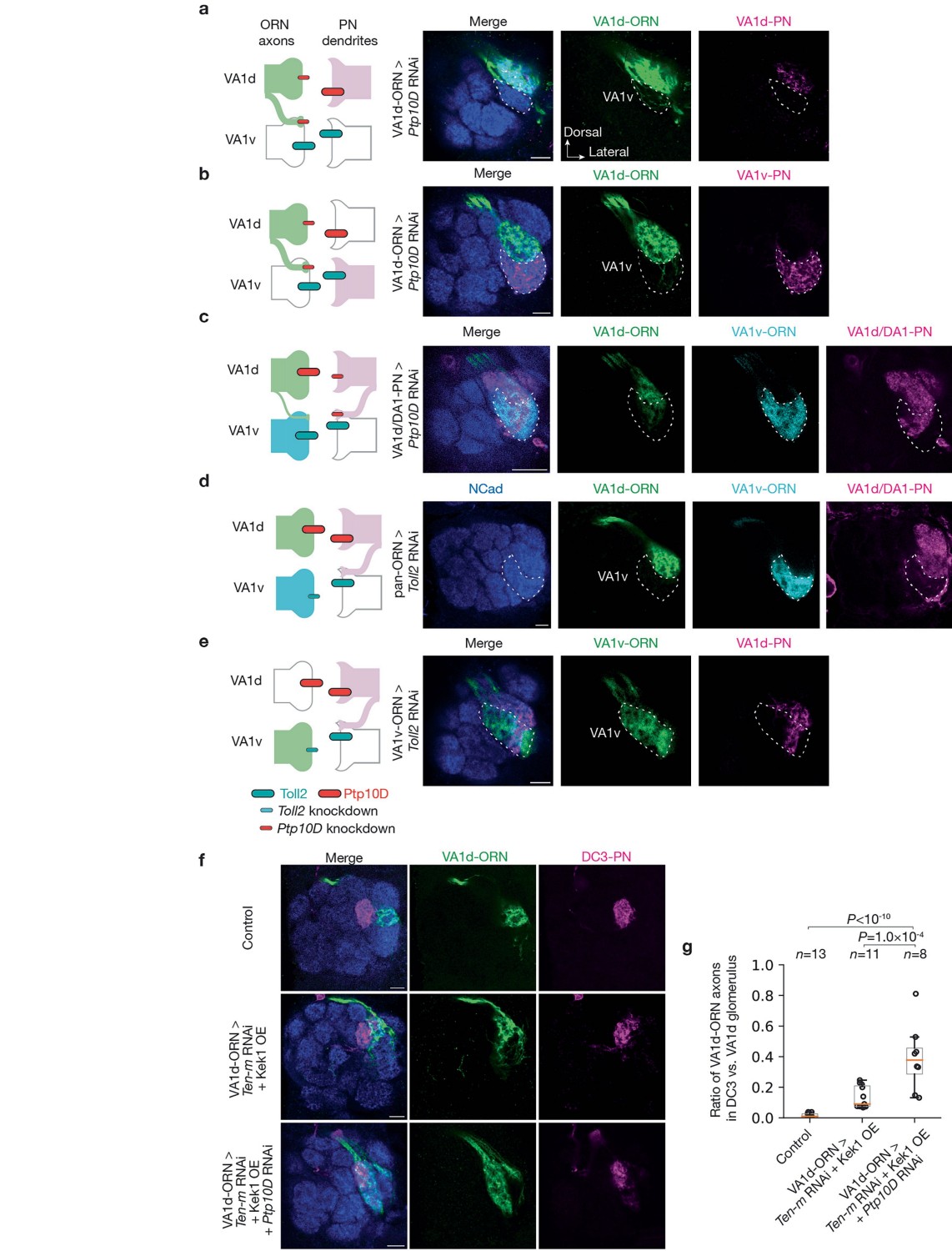

**Extended Data Fig. 3** | See next page for caption.

**Extended Data Fig. 3 | Additional loss-of-function experiments supporting that Ptp10D and Toll2 mediate PN-ORN repulsion.** Schematics on the left column show genetic labeling (color-filled neurites) and genetic manipulations (red bar, high expression level of Ptp10D; cyan bar, high expression level of Toll2; smaller bar, knockdown). Right columns are confocal images of representative single sections of adult antennal lobes showing neuropil staining by N-cadherin antibody (blue), and different types of ORN axons (green or cyan) or PN dendrites (magenta). Specific glomeruli are outlined based on N-cadherin staining. **a**, To examine whether the partner neurons of VA1d-ORNs were affected in experiments described in Fig. 2d, we simultaneously labeled VA1d-PN dendrites. Although VA1d-ORN axons (green) mistarget to the VA1v glomerulus in *Ptp10D* RNAi expressed in VA1d-ORNs as in Fig. 2d, the simultaneously dual-labeled VA1d-PN dendrites (magenta) only innervate the VA1d glomerulus. This result argues against the possibility that VA1d-ORN axons mistargeting is a secondary effect of VA1d-PN dendrites mistargeting. **b**, To confirm whether mistargeted VA1d-ORNs in Fig. 2d overlap with non-partner neurons innervating VA1v glomerulus, we simultaneously labeled VA1v-PN dendrites. Mistargeted VA1d-ORN axons (green) match with the simultaneously dual-labeled VA1v-PN dendrites (magenta) in *Ptp10D* RNAi expressed in VA1d-ORNs. **c**, To examine whether VA1d-PNs' partner and non-partner neurons were affected in Fig. 2i, we simultaneously labeled VA1d-ORN and VA1v-ORN axons. Some VA1d/DA1-PN dendrites (magenta) mistarget to the VA1v glomerulus and mismatch with VA1v-ORN axons (cyan) in *Ptp10D* RNAi expressed in VA1d/DA1-PNs as in Fig. 2i. A smaller fraction of VA1d-ORN axons (green) also mistarget to VA1v glomerulus but still intermingle with VA1d/DA1-PN dendrites. For the brains with no VA1d/DA1-PN dendrites mistargeting, no VA1d-ORN axons mistargeting was observed. This result suggests that VA1d-ORN axons mistargeting is likely to be a secondary effect of VA1d-PN dendrites mistargeting. **d**, To examine whether ORN knockdown of *Toll2* affects the correct targeting of any ORNs besides VA1d/DA1-PNs shown in Fig. 2j, we simultaneously labeled VA1d-ORN and VA1v-ORN axons with distinct markers. Whereas pan-ORN knockdown of *Toll2* causes some VA1d/DA1-PN dendrites (magenta) to mistarget to the VA1v glomerulus and mismatch with VA1v-ORN axons (cyan), consistent with VA1v-ORN knockdown of *Toll2* shown in Fig. 2j, VA1v-ORN axons only innervate the VA1v glomerulus, and VA1d-ORN axons (green) only innervate the VA1d glomerulus. This result suggests that Toll2 functions non-cell-autonomously and does not cause repulsion between VA1d-ORN axons and VA1v-ORN axons. **e**, Similar to **d**, but with a more specific manipulation as in Fig. 2j, except that we also simultaneously labeled VA1v-ORNs. Some VA1d-PN dendrites (magenta) mistarget to the VA1v glomerulus and mismatch with VA1v-ORN axons (green) when we knocked down *Toll2* only in VA1v-ORNs. This result suggests that Toll2 functions non-cell-autonomously in VA1v-ORNs to prevent VA1d-PN dendrites to mistarget to the VA1v glomerulus. **f**, To test the possibility that Ptp10D mediates homophilic attraction, we focus on DC3-PNs where Toll2 and Ptp10D are both highly expressed (Extended Data Fig. 2a). We would expect that knocking down *Ptp10D* in potential partner ORNs would decrease its matching with DC3-PNs if Ptp10D homophilic attraction plays a more dominant role in synaptic partner matching, but would increase its matching with DC3-PNs if Ptp10D–Toll2 repulsion plays a more dominant role. We used genetic manipulations of VA1d-ORNs to test this. Confocal images show neuropil staining by N-cadherin antibody (blue), DC3-PN dendrites (magenta), and VA1d-ORN axons (green). (Top row) In control, VA1d-ORN axons only innervate the VA1d glomerulus and does not overlap with DC3-PN dendrites. (Middle row) As knocking down *Ptp10D* alone cause VA1d-ORN axons to mistarget to the VA1v glomerulus and not to DC3 glomerulus, we incorporated manipulation of Ten-m and Kek1 to sensitize VA1d-ORN based on the results in the accompanying article[37] Fig. 5 CSP set2. VA1d-ORN axons overlap with DC3-PN dendrites following *Ten-m* knockdown and Kek1 overexpression in VA1d-ORNs. (Bottom row) VA1d-ORN axons overlap more with DC3-PN dendrites when *Ptp10D* is knocked down. This result argues against Ptp10D mediating homophilic attraction. **g**, Quantification of the mistargeting ratio of VA1d-ORN axons overlapping with DC3-PN dendrites versus not overlapping for experiments in **f**. *n* represents total antennal lobes examined. Boxes indicate geometric mean and 25% to 75% range. Whiskers extend to the most extreme data points within 1.5× the interquartile range. Kruskal–Wallis test with Bonferroni's multiple comparison. Scale bars = 10 μm.

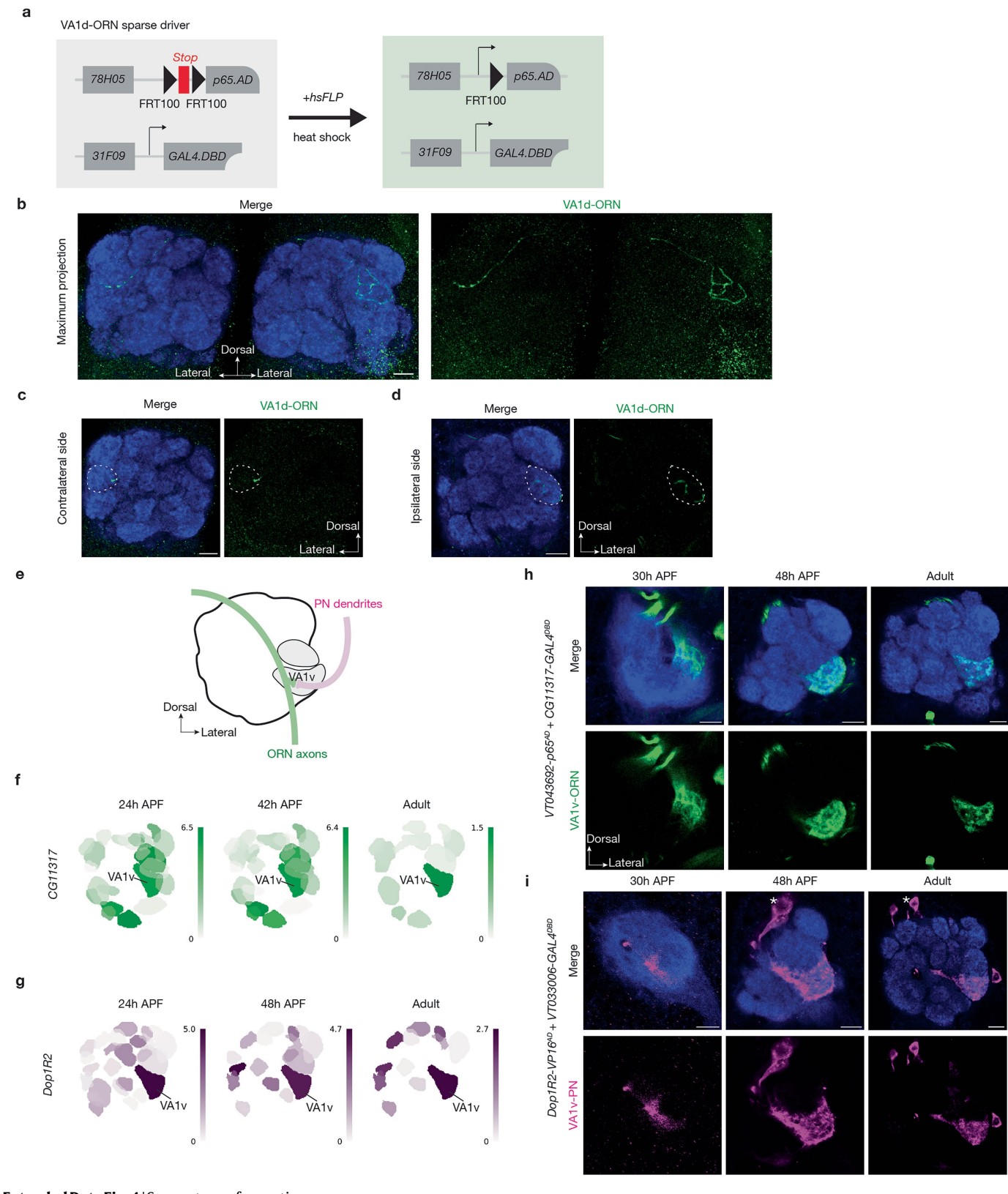

**Extended Data Fig. 4** | See next page for caption.

**Extended Data Fig. 4 | Characterization of new genetic drivers. a**, Schematic for VA1d-ORN sparse drivers using a previously described method[67]. The transcription activation domain (AD) is controlled by the enhancer *GMR78H05* and gated by *FRT100-STOP-FRT100*. *FRT100* sites are ~1% efficient as of wild-type FRT sites. STOP represent a transcription termination sequence. Heat-shock-induced FLP expression often enables split GAL4 expression in a single VA1d-ORN. **b**–**d**, VA1d-ORN sparse driver enable single VA1d-ORN labeling (green, labeled by a membrane-targeted GFP) as shown in the maximum projection (**b**), single contralateral section (**c**), or single ipsilateral section (**d**). The VA1d glomerulus location is verified by neuropil staining of N-cadherin (NCad) antibody (blue). **e**, Schematic of VA1v-ORN axons and VA1v-PN dendrites matching in the VA1v glomerulus (grey) in adult *Drosophila* antennal lobe (black solid line). Green, VA1v-ORNs; Magenta, VA1v-PNs. **f, g**, Single-cell RNA sequencing data showing the expression level of *CG11317* in VA1v-ORNs (**f**) and *Dop1R2* in VA1v-PNs (**g**) are consistently high across developmental stages. Heat map units: $\log_2(CPM + 1)$. CPM: counts per million reads. Data adapted from previous published studies[3,4]. **h, i**, Gene-based genetic drivers[73] express in VA1v-ORNs (green, labeled by a membrane-targeted GFP) (**h**) or VA1v-PNs (magenta, labeled by a membrane-targeted GFP) (**i**) at 30 h APF (left), 48 h APF (middle) and in adults (right). The VA1v glomerulus location is verified by neuropil staining of N-cadherin (NCad) antibody (blue). Both drivers enable expression of the transgenes in VA1v-ORNs or VA1v-PNs across developmental stages. Although they also express in ORNs or PNs targeting several other glomeruli, the expression is not detectable in glomeruli adjacent to VA1v (especially VA1d). *, PN cell bodies. Scale bars = 10 μm.

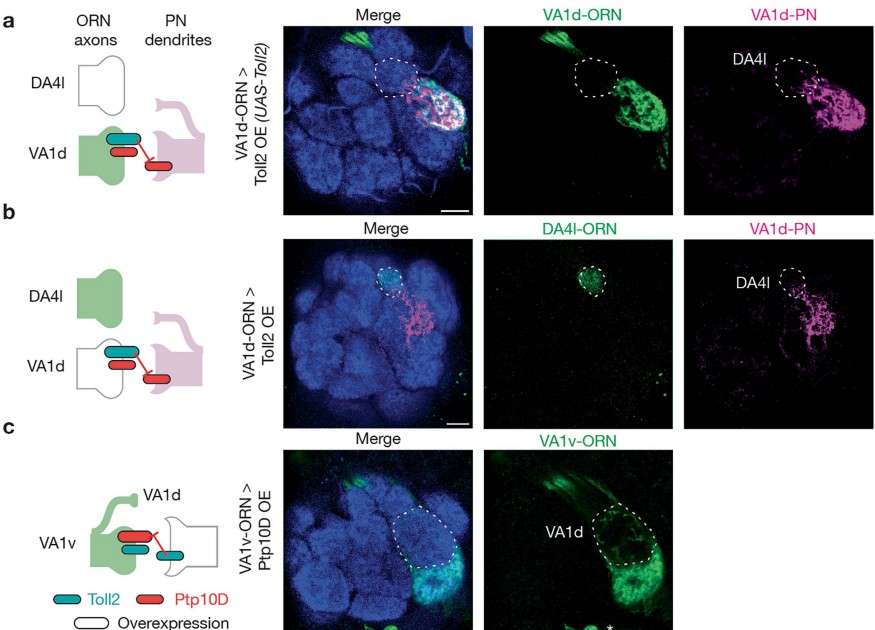

**Extended Data Fig. 5 | Additional gain-of-function experiments supporting that Ptp10D and Toll2 mediate PN-ORN repulsion.** Schematics on the left column show genetic labeling (color-filled neurites) and genetic manipulations (red bar, high expression level of Ptp10D; cyan bar, high expression level of Toll2; large bar, overexpression). Right columns are confocal images of representative single sections of adult antennal lobes showing neuropil staining by N-cadherin antibody (blue), and different types of ORN axons (green) or PN dendrites (magenta). Specific glomeruli are outlined based on N-cadherin staining. **a**, To examine whether the partner neurons of VA1d-PNs were affected in the experiment described in Fig. 3b, we simultaneously labeled VA1d-ORN axons. Although VA1d-PN dendrites (magenta) mistarget to the DA4l glomerulus following Toll2 overexpression in VA1d-ORNs as in Fig. 3b, VA1d-ORN axons (green) only innervate the VA1d glomerulus. This result supports a non-cell-autonomous function for Toll2. **b**, To test whether mistargeted VA1d-PN dendrites in Fig. 3b overlap with DA4l-ORN axons in DA4l glomerulus, we simultaneously labeled DA4l-ORN axons. Some VA1d-PN dendrites (magenta) mistarget to the DA4l glomerulus and mismatch with DA4l-ORN labeled by *Or43a-mCD8GFP* (green) following Toll2 overexpression in VA1d-ORNs. **c**, As Ptp10D expression in VA1v-ORNs is low and Toll2 expression in VA1v-PNs is high, we overexpressed Ptp10D in VA1v-ORNs, and observed that some VA1v-ORN axons (green) mistarget to the VA1d glomerulus, where both VA1d-PNs and VA1d-ORNs have low Toll2 expression (Fig. 1h). This result supports the repulsion model. Scale bars = 10 μm.

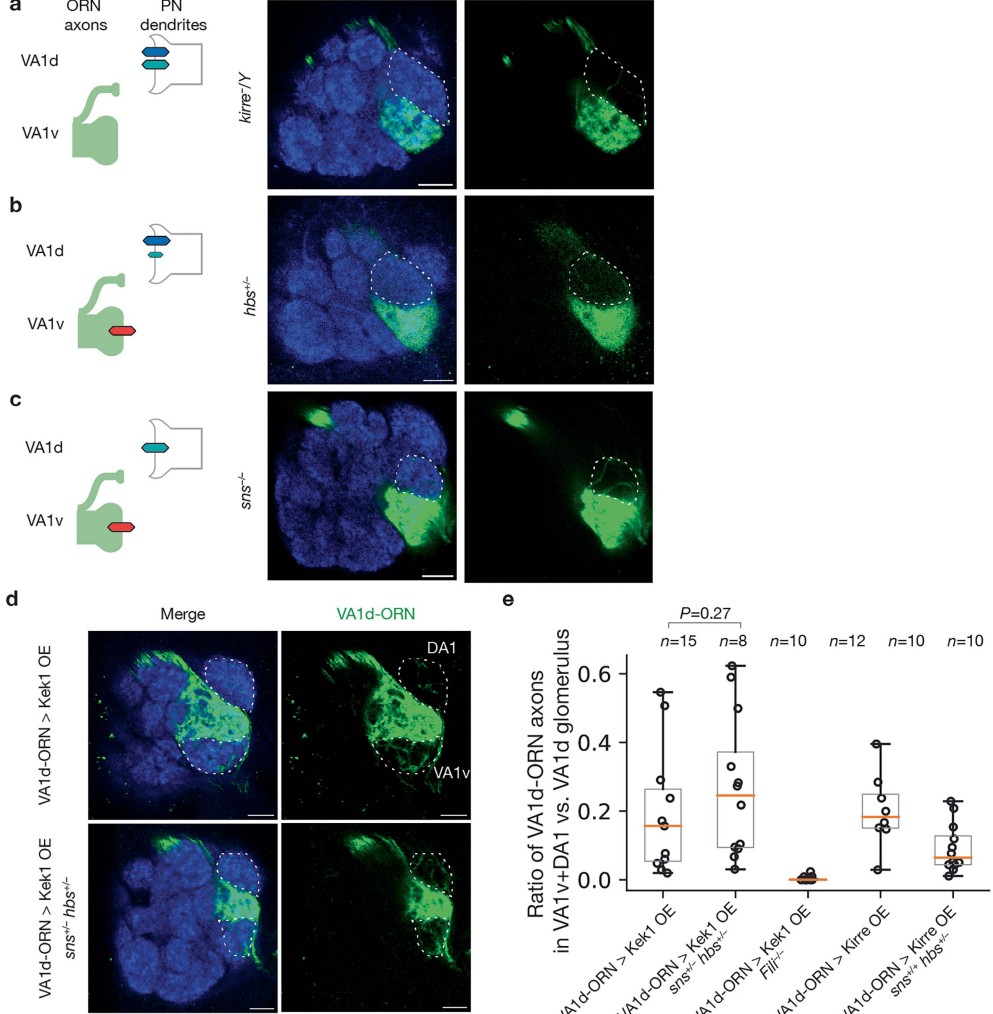

**Extended Data Fig. 6 | Additional loss-of-function experiments for Hbs/Sns–Kirre and lack of cross-interaction between Hbs/Sns–Kek1. a–c**, Left column shows experimental schematic. Middle and right columns are confocal images of adult antennal lobes showing neuropil staining by N-cadherin antibody (blue) and VA1v-ORN axons (green). The VA1d glomerulus is outlined based on N-cadherin staining. Control VA1v-ORN axons only innervate the VA1v glomerulus (Fig. 5b). Some VA1v-ORN axons mistarget to the VA1d glomerulus in *kirre* hemizygous mutant (**a**), *hbs* heterozygous mutant (**b**), or *sns* homozygous mutant (**c**). Penetrance was quantified in Fig. 5f. **d**, DA1 and VA1v glomeruli are outlined based on N-cadherin staining. Control VA1d-ORN axons only innervate the VA1d glomerulus ventral to the DA1 glomerulus and dorsal to the VA1v glomerulus (Fig. 4f). Some VA1d-ORN axons mistarget to the DA1 and VA1v glomeruli following Kek1 overexpression in VA1d-ORNs (top row). This phenotype is not suppressed in *hbs* and *sns* double heterozygous mutant (bottom row). Scale bars =10 μm. **e**, Quantification of the mistargeting ratio of VA1d-ORN axons in the DA1 + VA1v versus VA1d glomerulus. The first two columns are the quantification for (**d**). The rest are re-plotting of part of the data in Fig. 4m and Fig. 5n, showing that the Kek1 overexpression phenotype can be suppressed by *Fili* mutant (3rd column), and that *hbs* heterozygous mutant can suppress Kirre overexpression phenotype (4th and 5th columns). *n* represents total antennal lobes examined. Boxes indicate geometric mean and 25% to 75% range. Whiskers extend to the most extreme data points within 1.5× the interquartile range. Kruskal–Wallis test with Bonferroni's multiple comparison.

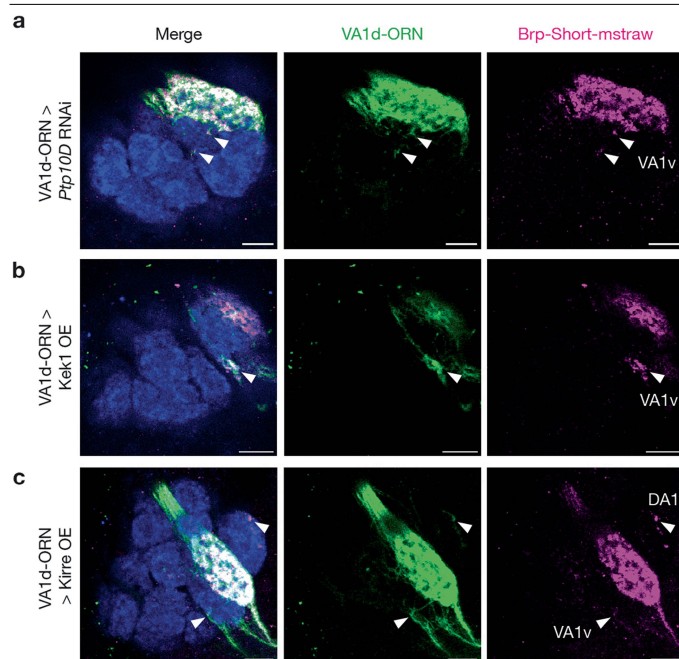

**Extended Data Fig. 7 | Mistargeted VA1d-ORN axons in neighboring glomeruli are enriched for a presynaptic terminal protein.** Confocal images of adult antennal lobes showing neuropil staining by N-cadherin antibody (blue), VA1d-ORN axons (green, labeled by a membrane-targeted GFP) and a presynaptic active zone marker Bruchpilot-Short[74] (Brp-Short, magenta). Knockdown of *Ptp10D* (**a**), overexpression of Kek1 (**b**), and overexpression of Kirre (**c**) in VA1d-ORNs caused their axons to mistarget to neighboring glomeruli (arrowheads). These mistargeted processes are enriched for Brp-Short, suggesting that mistargeted ORN axons might form synapses. Scale bars = 10 μm.

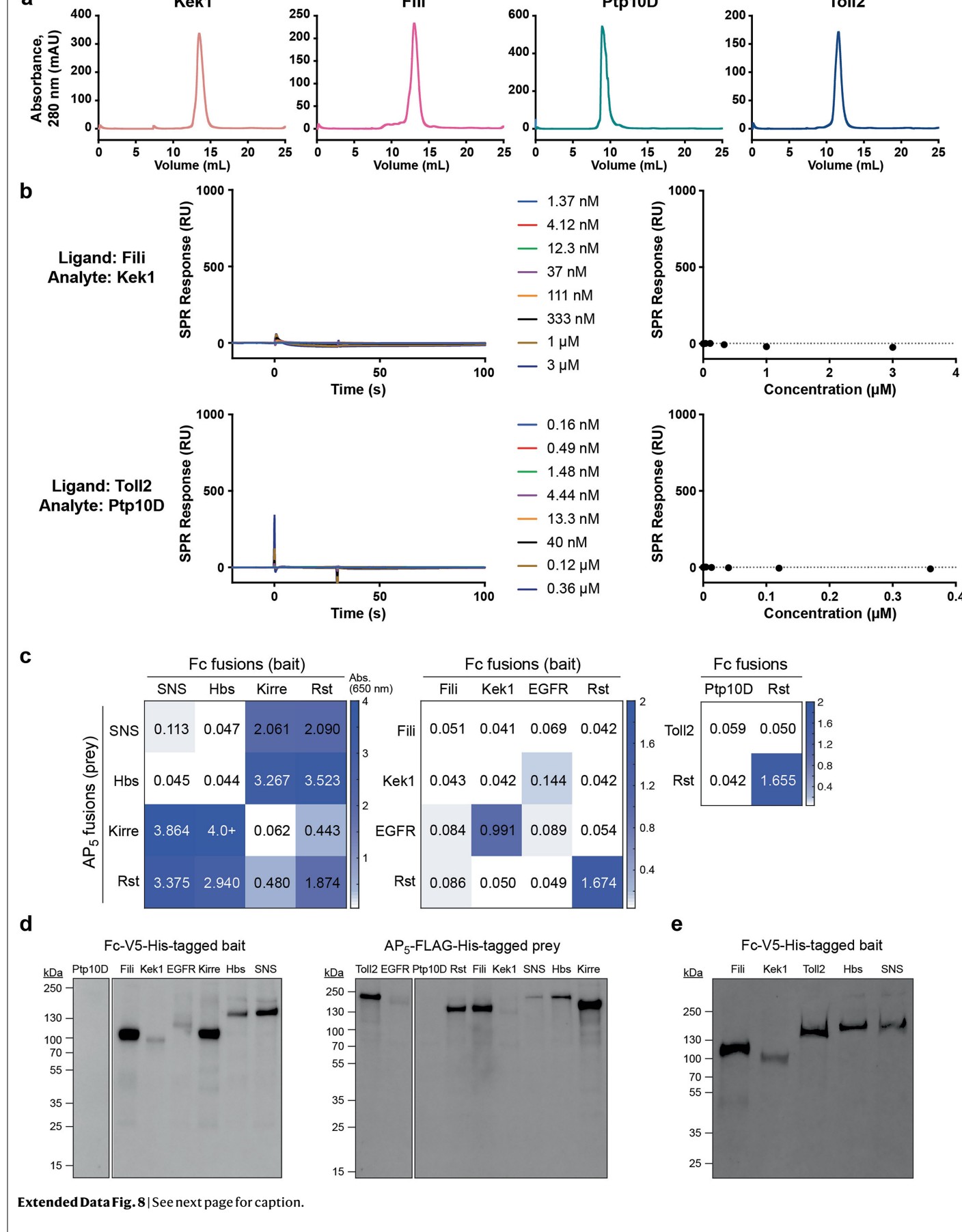

**Extended Data Fig. 8** | See next page for caption.

**Extended Data Fig. 8 | In vitro assays to test direct binding between CSP pairs. a**, Size-exclusion chromatography curves for four extracellular domains (ectodomains) purified on a Superdex 200 Increase 10/300 column on an ÄKTA Pure system, reporting absorbance at 280 nm with a path length of 0.2 cm. **b**, Surface plasmon resonance sensorgams of 30 s injections (left) and analyte concentration vs. response plots (right) show no direct physical interaction between Fili and Kek1, and between Toll2 and Ptp10D. **c**, Extracellular interactome assay (ECIA) to test binding between various ectodomains studied. We observe strong binding between Sns and Hbs with Kirre and its paralog Rst (left), and Kek1 with its previously described binding partner, EGFR (middle). No interaction between Fili and Kek1 (middle) or Ptp10D and Toll2 (right) were observed. **d**, Western blots of proteins used in the ECIA in (**c**) against the hexahistidine tag common in all constructs. There is a lack of detectable expression for Ptp10D ectodomain, which may be the reason behind no binding for Ptp10D in (**c**). **e**, Western blots of the same Fc fusions used directly in the tissue staining experiments in Extended Data Fig. 9.

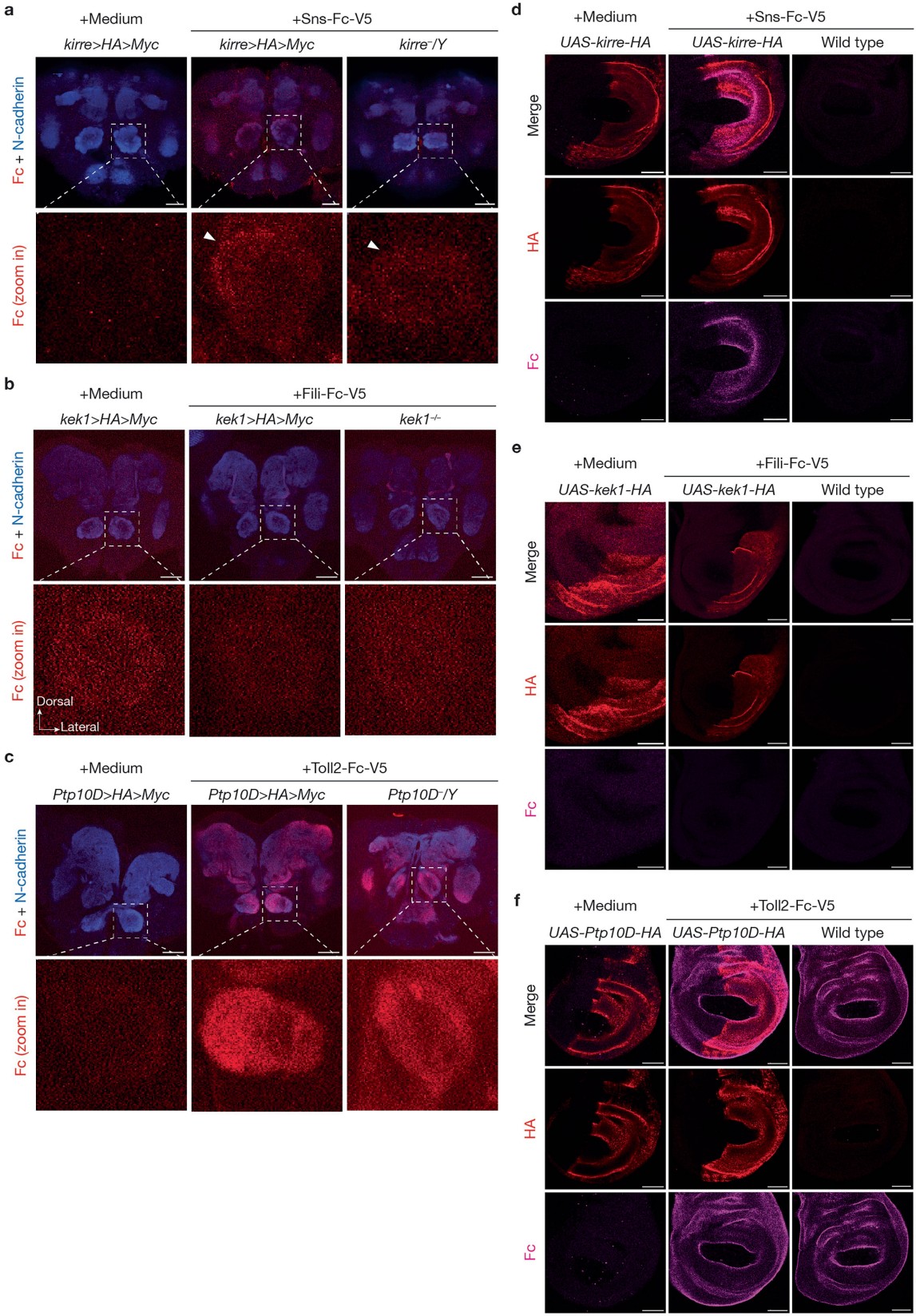

**Extended Data Fig. 9** | See next page for caption.

**Extended Data Fig. 9 | Live tissue staining assays to test direct binding between CSP pairs. a**, Confocal images of pupal brains (48 h APF) showing neuropil staining by N-cadherin antibody (blue), and binding of Sns-Fc-V5 in live pupa brains followed by fixing and staining against Fc (red). Control *kirre* conditionally tagged brains incubated with medium alone showed minimal background signal for Fc staining (left column). *kirre* conditionally tagged brains incubated with Sns-Fc-V5 proteins in medium have detectable signal for Fc staining in multiple brain regions and have differential signal in different glomeruli (middle column). These signals are substantially diminished in *kirre* mutant brains incubated with Sns-Fc-V5 proteins (right column), especially in the glomeruli where signal is high in the *kirre* conditionally tagged brains (arrowheads in the magnified images in the bottom rows). These results suggest Sns-Fc-V5 binds to endogenous Kirre, which serve as a positive control for this binding assay. **b**, Confocal images of pupal brains (48 h APF) showing neuropil staining by N-cadherin antibody (blue), and binding of Fili-Fc-V5 in live pupal brains followed by fixing and staining against Fc (red). *kek1* conditionally tagged brains (middle column) incubated with Fili-Fc-V5 proteins in medium do not exhibit an increase of Fc signal in the antennal lobe (magnified in the bottom row) comparing to the control without the addition of Fili-Fc-V5 (left column) or Fili-Fc-V5 in *kek1* homozygous mutant brain (right column). These results suggest that Kek1 does not bind to Fili in this assay. **c**, Confocal images of pupal brains (48 h APF) showing neuropil staining by N-cadherin antibody (blue), and binding of Toll2-Fc-V5 in live pupa brains followed by fixing and staining against Fc (red). Control *Ptp10D* conditionally tagged brains incubated with medium alone showed minimal background signal for Fc staining (left column). *Ptp10D* conditionally tagged brains incubated with Toll2-Fc-V5 proteins in medium have detectable signal for Fc staining in multiple brain regions (middle column). However, these signals are still present in *Ptp10D* mutant brains incubated with Toll2-Fc-V5 proteins (right column). These results suggest that Toll2-Fc-V5 binds to other proteins expressed in the brain, masking the detection of its potential binding to Ptp10D. **d**, Confocal images of wing discs dissected from 3rd instar larvae showing ectopic expression of *UAS-kirre-HA* in the posterior compartment driven by *engrailed (en)-GAL4* (HA staining, red), and binding of Sns-Fc-V5 in live wing discs followed by fixing and staining against Fc (magenta). Control Kirre-overexpressed wing disc incubated with medium alone shows minimal background signal for Fc staining (left column). Kirre-overexpressed wing disc incubated with Sns-Fc-V5 proteins in medium has specific binding signal in the posterior compartment where Kirre was overexpressed from *en-GAL4* (middle column). Interestingly, Sns binds strongest to regions with intermediate but not highest levels of Kirre overexpression. Without Kirre overexpression, no specific signal in the posterior compartment is detected (right column). These results suggest Sns-Fc-V5 binds to Kirre, which serves as a positive control for this binding assay. **e**, Confocal images of wing discs dissected from 3rd instar larvae showing ectopic expression of *UAS-kek1-HA* in the posterior compartment driven by *en-GAL4* (HA staining, red), and binding of Fili-Fc-V5 in live wing discs followed by fixing and staining against Fc (magenta). Fc signal is not detectable in Kek1-overexpressed wing disc incubated with medium alone, Kek1-overexpressed wing disc incubated with Fili-Fc-V5, or wild-type wing disc incubated with Fili-Fc-V5. These results suggest that Kek1 does not bind to Fili in this assay. **f**, Confocal images of wing discs dissected from 3rd instar larvae showing ectopic expression of *UAS-Ptp10D-HA* in the posterior compartment driven by *en-GAL4* (HA staining, red), and binding of Toll2-Fc-V5 in live wing discs followed by fixing and staining against Fc (magenta). Control Ptp10D-overexpressed wing disc incubated with medium alone showed minimal background signal for Fc staining (left column). Ptp10D-overexpressed wing disc incubated with Toll2-Fc-V5 proteins in medium has binding signal throughout the wing disc, not restricted to the posterior compartment where Toll2 was overexpressed (middle column). These signals are still present in wild-type wing disc incubated with Toll2-Fc-V5 proteins (right column). These results suggest Toll2-Fc-V5 binds to other proteins expressed in the wing disc, masking the detection of its potential binding to Ptp10D. The Fc-V5-tagged proteins used in this figure are visualized on Western blots in Extended Data Fig. 8e. Scale bar = 20 µm (**a**–**c**); Scale bar = 50 µm (**d**–**f**).

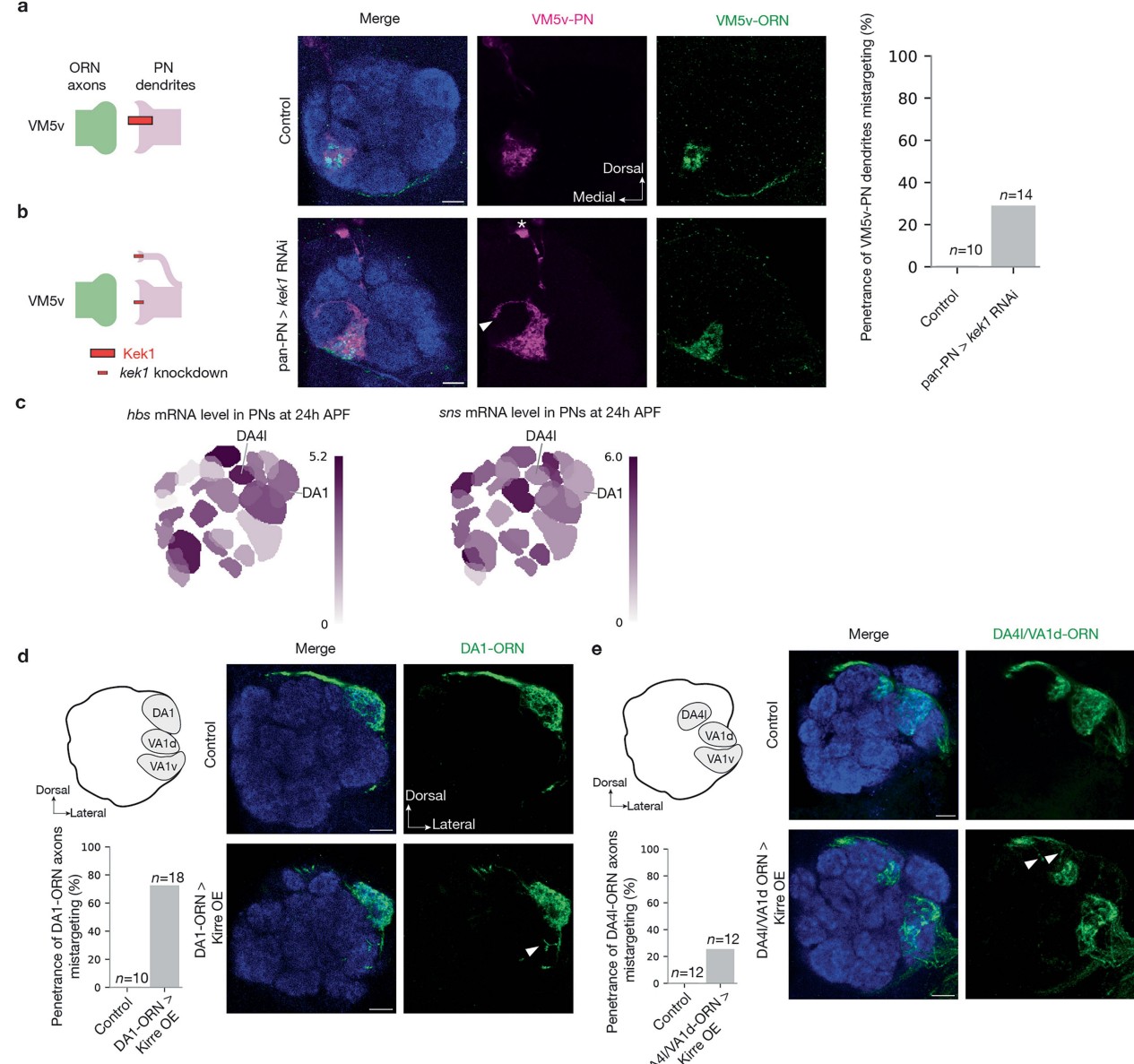

**Extended Data Fig. 10 | Genetic manipulation experiments of the CSPs in other parts of the antennal lobe. a,b,** Left column shows experimental schematic (red bar, high expression level of Kek1; smaller bar, knockdown). The middle columns are confocal images showing adult antennal lobe neuropil staining by N-cadherin antibody (blue), VM5v-PN dendrites (magenta), and VM5v-ORN axons (green). Control VM5v-PN dendrites labeled by *GMR86C10-LexA* only innervate the VM5v glomerulus and fully overlap with VM5v-ORN axons labeled by *Or98a-mCD8GFP* (**a**). Some VA5v-PN dendrites mistarget to other glomerulus (arrowheads) in *kek1* RNAi driven by the pan-PN driver, phenocopying ORN knockdown of *Fili*[27] (**b**). * denotes PN cell body. The right column shows penetrance of the mistargeting phenotypes. *n* represents total antennal lobes examined. **c**, Single-cell RNA sequencing data showing the expression level of *hbs* and *sns* throughout the antennal lobe at 24–30 h APF. Heat map units: $\log_2(\text{CPM}+1)$. CPM: counts per million reads. Data adapted from previous work[3,4]. **d,e**, Top left shows schematic of the adult antennal lobe with locations of three glomeruli highlighted in grey. Right columns are confocal images showing adult antennal lobe neuropil staining by N-cadherin antibody (blue), DA1-ORN axons (green in **d**), and DA4l/VA1d-ORN axons (green in **e**). Some DA1-ORN axons or DA4l/VA1d-ORN axons mistarget to neighboring glomeruli (bottom rows) following Kirre overexpression (arrowheads), which is not observed in control (top rows). Bottom left shows penetrance of mistargeting phenotypes. *n* represents total antennal lobes examined. Scale bars = 10 μm.

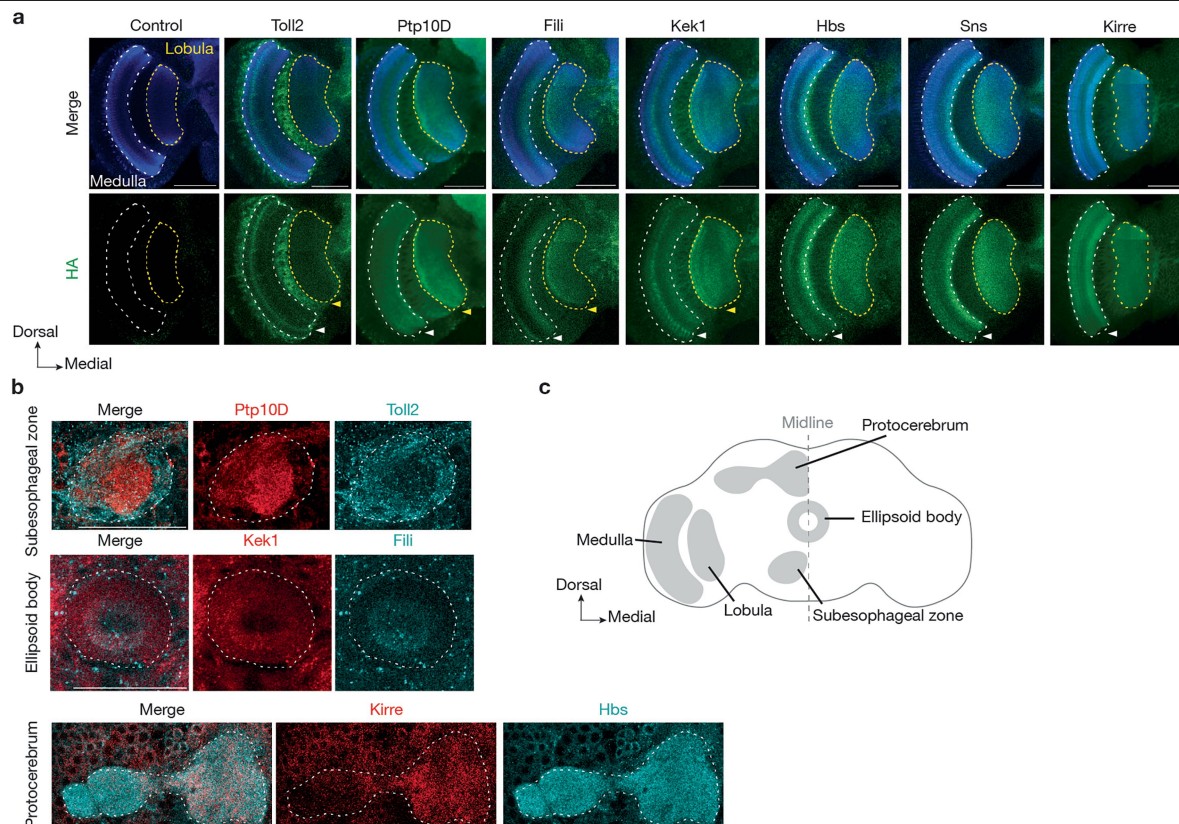

**Extended Data Fig. 11 | Three CSP pairs are differentially expressed in other brain regions. a**, Confocal images showing neuropil staining by N-cadherin antibody (blue) and HA staining (green) of tagged endogenous cell-surface proteins in the *Drosophila* optic lobe at 42–48 h APF. The medulla (white dotted lines) and lobula (yellow dotted lines) are outlined based on N-cadherin staining. Arrowheads indicate the layers with differential expression patterns. For example, Kek1 is highly expressed in a medulla layer that is low for Fili expression. Scale bars = 50 μm. **b**, Confocal images showing HA staining (for Ptp10D, Kek1, and Hbs) and V5 staining (for Toll2, Fili, and Kirre) of tagged endogenous CSP pairs in the same brain (each row) at 42–48 h APF. The subesophageal zone, ellipsoid body, and protocerebrum regions are outlined based on neuropil staining. Scale bars = 50 μm. **c**, Pupal *Drosophila* brain schematic with brain regions in **a** and **b** highlighted in grey in the left hemisphere.

**Extended Data Table 1 | Information for genes and manipulations used in the genetic screen**

| Gene | RNAi source | RNAi code | Manipulation | Penetrance | Gene | RNAi source | RNAi code | Manipulation | Penetrance |
|---|---|---|---|---|---|---|---|---|---|
| CCKLR-17D3 | BDSC | 67866 | Mz19-PN | 0/6 | klg | BDSC | 28746 | Mz19-PN | 0/7 |
| CCKLR-17D3 | BDSC | 67866 | VA1d-ORN | 4/5 | klg | BDSC | 28746 | pan-PN | 0/7 |
| CG10474 | BDSC | 51444 | pan-PN | 0/7 | klg | BDSC | 28746 | pan-ORN | 6/9* |
| CG14024 | VDRC | 100991 | pan-ORN | 0/5* | klg | VDRC | 102502 | pan-ORN | 0/6* |
| CG33143 | BDSC | 28823 | pan-PN | 0/12* | klg | VDRC | 108818 | pan-ORN | 0/7* |
| Con | VDRC | 17898 | pan-ORN | 7/7 | Mcr | BDSC | 65896 | pan-ORN | 0/3* |
| Con | BDSC | 28967 | pan-ORN | 4/5* | Mcr | BDSC | 65896 | pan-PN | 3/4* |
| Con | BDSC | 28967 | pan-PN | 3/7 | mgl | BDSC | 29324 | pan-ORN | 3/3* |
| Cow | BDSC | 38235 | pan-ORN | 1/7* | mgl | VDRC | 36389 | pan-ORN | 0/7* |
| Cow | BDSC | 44530 | pan-ORN | 0/9* | Nrt | VDRC | 8495 | pan-ORN | 0/8* |
| Cow | BDSC | 44530 | pan-PN | 1/8* | Nrt | BDSC | 28742 | pan-ORN | 0/10* |
| cue | VDRC | 1043 | pan-ORN | 5/5* | Nrt | VDRC | 106080 | pan-ORN | 0/6* |
| cue | VDRC | 104645 | pan-ORN | 0/5* | Nrt | BDSC | 28742 | pan-PN | 0/7* |
| cue | BDSC | 36875 | pan-ORN | 0/7 | PlexB | BDSC | 57813 | Mz19-PN | 3/8 |
| Diap2 | VDRC | 101187 | pan-ORN | 0/6* | PlexB | BDSC | 57813 | VA1d-ORN | 0/5 |
| Dop1R1 | BDSC | 62193 | pan-PN | 0/7 | Ptp10D | VDRC | 8010 | pan-ORN | 4/8 |
| ed | BDSC | 38209 | pan-ORN | 0/7* | Ptp10D | BDSC | 39001 | pan-ORN | 8/13* |
| ed | BDSC | 38243 | pan-ORN | 0/4* | Ptp10D | BDSC | 39001 | pan-PN | 7/7* |
| ed | BDSC | 38243 | pan-PN | 3/7* | Ptp99A | BDSC | 57299 | DA1-ORN | 7/7 |
| Fas2 | BDSC | 34084 | Mz19-PN | 0/7* | Ptp99A | BDSC | 57299 | Mz19-PN | 7/7 |
| Fas2 | BDSC | 34084 | pan-ORN | 0/10* | Ptp99A | BDSC | 57299 | pan-ORN | 2/10* |
| Fas2 | VDRC | 103807 | pan-ORN | 0/7* | Ptp99A | VDRC | 103457 | pan-ORN | 0/8* |
| Fas2 | BDSC | 28990 | pan-ORN | 0/8* | Ptp99A | VDRC | 108505 | pan-ORN | 0/7* |
| Fas3 | BDSC | 77396 | pan-PN | 0/7 | Ptp99A | BDSC | 57299 | pan-PN | 3/7 |
| Fas3 | BDSC | 77396 | pan-ORN | 0/5* | rst** | BDSC | 28672 | DA1-ORN | 0/7 |
| fra | BDSC | 40826 | pan-PN | 0/7 | rst** | BDSC | 28672 | Mz19-PN | 0/7 |
| fred | BDSC | 42621 | DA1-ORN | 0/6 | rst** | BDSC | 28672 | pan-PN | 0/7 |
| fred | BDSC | 42621 | Mz19-PN | 2/2 | sdk | BDSC | 33412 | Mz19-PN | 1/6 |
| fred | BDSC | 42621 | pan-ORN | 0/9* | sdk | BDSC | 33412 | VA1d-ORN | 0/5 |
| fred | VDRC | 46180 | pan-ORN | 1/6* | side-IV | BDSC | 82064 | VA1d-ORN | 10/10 |
| fred | VDRC | 101138 | pan-ORN | 0/8* | side-IV | VDRC | 102563 | VA1d-ORN | 5/7 |
| fred | BDSC | 42621 | pan-PN | 0/7* | side-IV | BDSC | 82064 | VA1d-PN | 1/8 |
| hbs | VDRC | 40898 | pan-neuronal | 8/12 | side-IV | VDRC | 102563 | VA1d-PN | 2/12 |
| hbs | BDSC | 57003 | pan-PN | 3/6* | side-V | BDSC | 61953 | Mz19-PN | 2/9 |
| ldit | VDRC | 101893 | pan-ORN | 0/6* | side-V | BDSC | 61953 | VA1d-ORN | 6/6 |
| ldit | BDSC | 28823 | pan-ORN | 0/12* | slit | VDRC | 20210 | pan-ORN | 0/7* |
| ldit | BDSC | 28823 | pan-PN | 0/4* | slit | BDSC | 31467 | pan-ORN | 3/10* |
| kek1 | VDRC | 36252 | pan-ORN | 5/6* | slit | BDSC | 31468 | pan-ORN | 2/8* |
| kek1 | BDSC | 57000 | pan-ORN | 5/8 | slit | VDRC | 108853 | pan-ORN | 0/7* |
| kek1 | VDRC | 101166 | pan-ORN | 1/8* | slit | BDSC | 31467 | pan-PN | 0/7* |
| kek5 | BDSC | 40830 | Mz19-PN | 0/7 | slit | BDSC | 31468 | pan-PN | 0/7* |
| kek5 | BDSC | 40830 | pan-ORN | 7/7* | sns | VDRC | 109442 | pan-neuronal | 14/20 |
| kek5 | BDSC | 40830 | pan-PN | 2/7* | TkR86C | VDRC | 13392 | pan-ORN | 0/5* |
| kek5 | BDSC | 40830 | VA1d-ORN | 4/5 | TkR86C | VDRC | 107090 | pan-ORN | 0/6* |
| kek6 | BDSC | 61212 | Mz19-PN | 0/7 | TkR86C | BDSC | 31884 | pan-ORN | 0/3* |
| kek6 | BDSC | 61212 | pan-ORN | 7/7 | tnc | BDSC | 60058 | pan-PN | 7/7* |
| kek6 | VDRC | 109681 | VA1d-ORN | 5/6 | Toll2 | VDRC | 965 | pan-ORN | 8/8* |
| kirre | VDRC | 109585 | pan-ORN | 9/12 | Toll2 | BDSC | 30498 | pan-ORN | 2/3* |
| | | | | | Toll2 | VDRC | 36305 | pan-ORN | 2/3* |
| | | | | | Toll2 | VDRC | 965 | pan-PN | 3/7 |

*indicates simultaneous knockdown of *Ten-m* using RNAi. Some of the screening experiments were done with the simultaneous knockdown of *Ten-m* to 'sensitize' the system because: (1) Knocking down *Ten-m* causes partial mismatching phenotype between ORN axons and PN dendrites[13] and (2) Ten-m is highly expressed in many of the glomeruli labeled in our screening. In some of the 'sensitized' screening experiments, we followed up the initial phenotype with single manipulation that did not contain *Ten-m* knockdown. In general, we observed equal or higher penetrance when *Ten-m* was simultaneously knocked down. For example, in the *Toll2* RNAi experiments, the penetrance is 3/7 when using RNAi line 965 from VDRC alone and is 8/8 when using both line 965 and *Ten-m* RNAi. **Rst (Roughest) binds with Hbs and Sns *in vitro*[49] and *rst* is also expressed in PNs and ORNs based on the single-cell transcriptome data[3,4]. However, we did not observe any mistargeting phenotype in cell-type-specific *rst* knockdown experiments.

**Extended Data Table 2 | Summary of genetic manipulation experiments that did not produce mistargeting phenotype**

| Exp.# | Genetic manipulation | Cell types examined | m/n[1] | Interpretation |
|---|---|---|---|---|
| 1 | *Ptp10D⁻/Y* | VA1v-ORN axons | 0/18 | Ptp10D is not required for VA1v-ORN axon targeting |
| 2 | VA1d/DA1-PN > *Ptp10D* RNAi | VA1d-ORN axons | 0/14 | Ptp10D does not affect targeting of partner neurons |
| 3 | VA1v-ORN > *Ptp10D* RNAi | VA1v-ORN axons | 0/10 | Ptp10D does not function in VA1v-ORNs for their axon targeting |
| 4 | VA1d/DA1-PN > *Toll2* RNAi | VA1d-ORN axons | 0/12 | Toll2 does not affect targeting of partner neurons |
| 5 | VA1d/DA1-PN > *Toll2* RNAi | VA1v-ORN axons | 0/12 | Toll2 does not function in VA1d-PNs to regulate VA1v-ORN axon targeting |
| 6 | VA1d/DA1-PN > *Toll2* RNAi | VA1d/DA1 PN dendrites | 0/12 | Toll2 does not function in VA1d-PNs to regulate their dendrite targeting |
| 7 | pan-ORN > *Toll2* RNAi | VA1v-PN dendrites | 0/10 | Toll2 does not affect targeting of partner neurons |
| 8 | VA1v-ORN > *Toll2* RNAi | VA1v-PN dendrites | 0/10 | Toll2 does not affect targeting of partner neurons |
| 9 | VA1d-ORN > Toll2 OE | VA1v-PN dendrites | 0/11 | Toll2 unlikely mediates homophilic attraction |
| 10 | VA1d/DA1-PN > *Fili* RNAi | VA1d-ORN axons | 0/14 | Fili does not affect targeting of partner neurons |
| 11 | *kek1⁻/⁻* | VA1d-ORN axons | 0/18 | Kek1 is not required for VA1d-ORN axon targeting |
| 12 | pan-ORN > *kek1* RNAi | VA1d-ORN axons | 0/10 | Kek1 is not required in ORNs for in VA1d-ORN axon targeting |
| 13 | VA1d-ORN > *kek1* RNAi | VA1d-ORN axons | 0/10 | Kek1 is not required in VA1d-ORNs for their axon targeting |
| 14 | pan-ORN > *kek1* RNAi | VA1v-PN dendrites | 0/11 | Kek1 does not affect targeting of partner neurons |
| 15 | VA1v-ORN > *kek1* RNAi | VA1v-PN dendrites | 0/10 | Kek1 does not affect targeting of partner neurons |
| 16 | VA1d-ORN > Kek1 OE | VA1v-PN dendrites | 0/10 | Kek1 unlikely mediates homophilic attraction |
| 17 | VA1v-ORN > Kek1 OE | VA1v-ORN axons | 0/10 | Kek1 endogenous expression level is higher in VA1v-ORNs than VA1d-ORNs and overexpression of Kek1 in VA1v-ORNs does not cause ORN axon mistargeting phenotype |
| 18 | *hbs⁺/⁻* | VA1d-ORN axons | 0/10 | Mutant alone does not have any VA1d-ORN axons mistargeting phenotype (Control for Fig. 5) |
| 19 | *hbs⁺/⁻, sns⁻/⁻* | VA1d-ORN axons | 0/10 | Mutant alone does not have any VA1d-ORN axons mistargeting phenotype (Control for Fig. 5) |
| 20 | VA1d/DA1-PN > *hbs* RNAi | VA1d-ORN axons | 0/10 | Hbs does not affect targeting of partner neurons |
| 21 | *sns⁻/⁻* | VA1d-ORN axons | 0/10 | Mutant alone does not have any VA1d-ORN axons mistargeting phenotype (Control for Fig. 5) |
| 22 | VA1d/DA1-PN > *sns* RNAi | VA1d-ORN axons | 0/10 | Sns does not affect targeting of partner neurons |
| 23 | *kirre⁻/Y* | VA1d-ORN axons | 0/10 | Mutant alone does not have any VA1d-ORN axons mistargeting phenotype (Control for Fig. 5) |
| 24 | pan-ORN > *kirre* RNAi | VA1d-ORN axons | 0/12 | Kirre is not required in ORNs for VA1d-ORN axon targeting |
| 25 | VA1d-ORN > *kirre* RNAi | VA1d-ORN axons | 0/10 | Kirre is not required in VA1d-ORNs for their axon targeting |
| 26 | pan-ORN > *kirre* RNAi | VA1v-PN dendrites | 0/15 | Kirre does not affect targeting of partner neurons |
| 27 | VA1v-ORN > *kirre* RNAi | VA1v-PN dendrites | 0/10 | Kirre does not affect targeting of partner neurons |

[1]m = number of antennal lobes with mistargeting phenotypes; n = number of total antennal lobes examined. Besides the results of genetic manipulation experiments shown in the figures and Extended Data figures, additional results with no mistargeting observed in certain ORN axons and PN dendrites are listed in this table. Collectively, these results argue against hypotheses alternative to our model (Fig. 5o) and suggest that: (1) the CSPs do not function in neurons with relatively low expression level; (2) the function of these CSPs are mainly to regulate interactions between non-partner neurons; (3) the CSPs unlikely mediate homophilic attraction. Despite the evidence supporting that the CSP pairs mediate repulsion between non-partner neurons, we cannot completely rule out the possibility that some of the CSPs have additional roles. As an example, the mouse teneurin-3 mediates both homophilic attraction with itself[75] and heterophilic repulsion with latrophilin-2[12] in the hippocampal networks.

# Reporting Summary

## Statistics

For all statistical analyses, confirm that the following items are present in the figure legend, table legend, main text, or Methods section.

| n/a | Confirmed | |
|---|---|---|
| ☐ | ☒ | The exact sample size (*n*) for each experimental group/condition, given as a discrete number and unit of measurement |
| ☐ | ☒ | A statement on whether measurements were taken from distinct samples or whether the same sample was measured repeatedly |
| ☐ | ☒ | The statistical test(s) used AND whether they are one- or two-sided<br>*Only common tests should be described solely by name; describe more complex techniques in the Methods section.* |
| ☐ | ☒ | A description of all covariates tested |
| ☐ | ☒ | A description of any assumptions or corrections, such as tests of normality and adjustment for multiple comparisons |
| ☐ | ☒ | A full description of the statistical parameters including central tendency (e.g. means) or other basic estimates (e.g. regression coefficient) AND variation (e.g. standard deviation) or associated estimates of uncertainty (e.g. confidence intervals) |
| ☐ | ☒ | For null hypothesis testing, the test statistic (e.g. *F*, *t*, *r*) with confidence intervals, effect sizes, degrees of freedom and *P* value noted<br>*Give P values as exact values whenever suitable.* |
| ☒ | ☐ | For Bayesian analysis, information on the choice of priors and Markov chain Monte Carlo settings |
| ☒ | ☐ | For hierarchical and complex designs, identification of the appropriate level for tests and full reporting of outcomes |
| ☒ | ☐ | Estimates of effect sizes (e.g. Cohen's *d*, Pearson's *r*), indicating how they were calculated |

*Our web collection on statistics for biologists contains articles on many of the points above.*

## Software and code

Policy information about availability of computer code

| Data collection | Immunostained brains were imaged using a laser-scanning confocal microscope (Zeiss LSM 780). |
|---|---|
| Data analysis | Images were processed using Fiji (version: 2.1.0/1.54j) and analyzed with custom code in python 2.7 and 3.6 and are available on github (https://github.com/ZhuoranLi97/repulsive_interactions). flyCRISPR (https://flycrispr.org/) was used for designing gRNA. |

For manuscripts utilizing custom algorithms or software that are central to the research but not yet described in published literature, software must be made available to editors and reviewers. We strongly encourage code deposition in a community repository (e.g. GitHub). See the Nature Portfolio guidelines for submitting code & software for further information.

## Data

Policy information about availability of data

All manuscripts must include a data availability statement. This statement should provide the following information, where applicable:
- Accession codes, unique identifiers, or web links for publicly available datasets
- A description of any restrictions on data availability
- For clinical datasets or third party data, please ensure that the statement adheres to our policy

All data are included in the manuscript and supplementary materials.

# Research involving human participants, their data, or biological material

Policy information about studies with human participants or human data. See also policy information about sex, gender (identity/presentation), and sexual orientation and race, ethnicity and racism.

| | |
|---|---|
| Reporting on sex and gender | Not applicable. |
| Reporting on race, ethnicity, or other socially relevant groupings | Not applicable. |
| Population characteristics | Not applicable. |
| Recruitment | Not applicable. |
| Ethics oversight | Not applicable. |

Note that full information on the approval of the study protocol must also be provided in the manuscript.

# Field-specific reporting

Please select the one below that is the best fit for your research. If you are not sure, read the appropriate sections before making your selection.

☒ Life sciences  ☐ Behavioural & social sciences  ☐ Ecological, evolutionary & environmental sciences

For a reference copy of the document with all sections, see nature.com/documents/nr-reporting-summary-flat.pdf

# Life sciences study design

All studies must disclose on these points even when the disclosure is negative.

| | |
|---|---|
| Sample size | No statistical tests were used to determine sample size. We used sample sizes (~4-20 flies per condition) that been previously shown to have sufficient statistical power in similar experiments in the past (e.g., Hong, Mosca, Luo 2012, Lyu, Abbott, Maimon 2022) |
| Data exclusions | We did not exclude flies or data from any analysis, unless brains stained for imaging appeared unsuitable (e.g., broken) at the time of imaging. |
| Replication | All experiments discussed in the paper were conducted on multiple animals with sample size specified. In immunostaining plots, data across multiple days were collected and all imaged brains showed the same qualitative pattern of staining. As the techinical variances are usually small, it is a standard in the field to focus on biological replicates. |
| Randomization | Organisms are not allocated to control and experimental groups by the experimenter in this work, rather the flies' genotype determines their group. Thus, randomization of individuals into treatments groups is not relevant. |
| Blinding | For counting the penetrance of the phenotype, the investigators were blind to the flies' genotypes. For the rest, the investigators were not blind to the genotype, and data collection and analysis was done computationally. |

# Reporting for specific materials, systems and methods

We require information from authors about some types of materials, experimental systems and methods used in many studies. Here, indicate whether each material, system or method listed is relevant to your study. If you are not sure if a list item applies to your research, read the appropriate section before selecting a response.

## Materials & experimental systems

| n/a | Involved in the study |
|---|---|
| ☐ | ☒ Antibodies |
| ☐ | ☒ Eukaryotic cell lines |
| ☒ | ☐ Palaeontology and archaeology |
| ☐ | ☒ Animals and other organisms |
| ☒ | ☐ Clinical data |
| ☒ | ☐ Dual use research of concern |
| ☒ | ☐ Plants |

## Methods

| n/a | Involved in the study |
|---|---|
| ☒ | ☐ ChIP-seq |
| ☒ | ☐ Flow cytometry |
| ☒ | ☐ MRI-based neuroimaging |

# Antibodies

| | |
|---|---|
| Antibodies used | rat anti-NCadherin (1:40; DN-Ex#8, Developmental Studies Hybridoma Bank), chicken anti-GFP (1:1000; GFP-1020, Aves Labs), rabbit anti-DsRed (1:500; 632496, Clontech), mouse anti-rat CD2 (1:200; OX-34, Bio-Rad), rabbit anti-HA (1:100, 3724S, Cell Signaling), mouse anti-HA (1:100, 2367S, Cell Signaling), rabbit anti-Myc (1:250, 2278S, Cell Signaling), and mouse anti-V5 (1:250, R960-25, Thermo Fisher Scientific). Secondary antibodies include: Fluorescein (FITC) AffiniPure Donkey Anti-Chicken IgY (IgG) (H+L) (Jackson ImmunoResearch 703-095-155); Donkey anti-Rabbit IgG (H+L) Highly Cross-Adsorbed Secondary Antibody, Alexa Fluor™ Plus 555 (ThermoFisher A32794); Alexa Fluor® 647 AffiniPure Donkey Anti-Mouse IgG (H+L) (Jackson Immunoresearch 715-605-151); DyLight™ 405 AffiniPure Donkey Anti-Rat IgG (H+L) (Jackson ImmunoResearch 712-475-153); Cy3-Donkey Anti-Rat IgG (H+L) (min X) (Jackson Immunoresearch 712-165-153); The anti-His-Tag Antibody coupled with iFlour 488, Genscript, A01800, used at a 1:500 dilution to detect secreted proteins from S2 cells in westerns (Ext. Data Fig. 8d,e) |
| Validation | All primary antibodies used in this study were validated as described at the following websites (and references therein): DSHB: https://dshb.biology.uiowa.edu/DN-Ex-8, Rockland: https://rockland-inc.com/store/Antibodies-to-GFP-and-Antibodies-to-RFP-600-901-215-O4L_23908.aspx, Takara: https://www.takarabio.com/products/antibodies-and-elisa/fluorescent-protein-antibodies/red-fluorescent-protein-antibodies?srsltid=AfmBOopUZqVextBqypoqsvRxsHH-H9rGlg0NFICn1UMie592NHF348BQ, Bio-Rad: https://www.bio-rad-antibodies.com/monoclonal/rat-cd2-antibody-ox-34-mca154.html?f=purified, Cell signaling: https://www.cellsignal.com/products/primary-antibodies/ha-tag-c29f4-rabbit-mab/3724?srsltid=AfmBOoo4fq323JT2OOUmSS57Ql81bkOHqgu_OEh2zrMiEFOwI8we-9yz; https://www.cellsignal.com/products/primary-antibodies/ha-tag-6e2-mouse-mab/2367?srsltid=AfmBOooyGtCSJAI0-x_0SG5HHyidV_FTUMprTD7bHWjkSghXfcqfT7Dw; https://www.cellsignal.com/products/primary-antibodies/myc-tag-71d10-rabbit-mab/2278?srsltid=AfmBOoqTY3_gUyCF-LzvhacFotUvpwIs3f3VWvmNzOVVtljc5Jwm6nl7; https://www.thermofisher.com/antibody/product/V5-Tag-Antibody-clone-SV5-Pk1-Monoclonal/R960-25; Validated by Genscript  (https://www.genscript.com/product/documents/down?doc_name=A01800_Datasheet_Rev05.pdf&file=scm_files/productFile_notes/2025/01/07/20250107091652_A01800.pdf), and by our lab, where expression cell lines (S2 and High Five) had complete absence of signal in the media of non-transfected cells. |

# Eukaryotic cell lines

Policy information about cell lines and Sex and Gender in Research

| | |
|---|---|
| Cell line source(s) | High Five cells (BTI-Tn-5B1-4) from Trichoplusia ni - for protein expression using baculoviruses - Thermo Fisher #B855-02; Sf9 cells from Spodoptera frugiperda, used for baculovirus production (Thermo Fisher, 12659017); S2 cells from Drosophila melanogaster (DGRC #6). |
| Authentication | Cell lines (from commercial source) were not authenticated, as they were only used as an exogenous production source of protein, and not studied for any biological functions. |
| Mycoplasma contamination | We regularly test our cell lines for mycoplasma contamination. None observed. |
| Commonly misidentified lines (See ICLAC register) | None used. |

# Animals and other research organisms

Policy information about studies involving animals; ARRIVE guidelines recommended for reporting animal research, and Sex and Gender in Research

| | |
|---|---|
| Laboratory animals | We used male and female Drosophila melanogaster. All fly strains and fly genotypes are described in details in the Methods. The w[1118] strain was used, with ages ranging from the larvae stage to 7 days old. |
| Wild animals | The study did not involve wild animals. |
| Reporting on sex | Experiments were performed on both sexes and reached similar conclusion. |
| Field-collected samples | The study did not involve samples collected from the field. |
| Ethics oversight | No ethical oversight was required because no vertebrates were used. |

Note that full information on the approval of the study protocol must also be provided in the manuscript.

## Plants

| | |
|---|---|
| Seed stocks | Not applicable. |
| Novel plant genotypes | Not applicable. |
| Authentication | Not applicable. |

