## [Peer Review File · Nature]

Repulsions instruct synaptic partner matching in an olfactory circuit

Corresponding Author: Dr Liqun Luo

Version 0:

Reviewer comments:

Referee #1

(Remarks to the Author)

This is an interesting paper describing multiple repulsive signaling cues controlling innervation of antennal lobe glomeruli. It is a companion to an excellent paper using these and other proteins to alter glomerular targeting in a programmed manner. This manuscript, however, has many problems. The experiments seem to be constructed assuming that the preferred model is correct, even though the differential expression of the repulsive cues between neurons is not convincing in most cases. Also, because they cannot demonstrate direct interactions between the proteins, it is possible that the repulsive effects are not interactions between these specific proteins, but indirect effects on signaling pathways in which they participate.

Fig. 1.

- a. Most of these differences in protein levels are not convincing in the images shown, even though they were presumably selected as demonstrations of the data reported in panel j. I see almost no difference between Ptp10D levels or Kirre levels in VA1d and VA1v ORNs. There are no detectable differences between Toll2 levels, Kek1 levels, and Kirre levels between VA1d and VA1v PNs. They also can't see Fili expression in PNs but say that it is there based on another study. There may be more differences in RNA levels, but protein levels are not strongly correlated with RNA levels measured in single-cell sequencing experiments. About all that one can say about protein levels based on RNA counts is that the protein is probably expressed at some level in a cell that has RNA.
- b. "cyan in e, g, i, j and red in d, f, h." Actually it is red in c, e, g and cyan in d, f, h, i. J is a graph.

Fig. 2.

- a. These phenotypes are not impressive, if I am reading the figure legend correctly. Mistargeting is scored if one axon or dendrite per sample is mistargeted, so in a case like Ptp10D VA1d ORN RNAi, where it is indicated as 30%, that presumably means that in 70% of the samples not a single axon mistargets.
- b. They don't say which RNAis are used in the images shown. Also, isn't it Toll $-/-$, not Toll $+/-$, in k? Panel f says $-/-$.
- c. Since the differences in Ptp10D levels between VA1d and VA1v ORNs are very small (Fig. 1), as are the differences in Toll2 levels in VA1d and VA1v PNs, it is not justified to confine the RNAi experiments to VA1d ORNs for Ptp10D and VA1v PNs for Toll2. The authors are assuming that their model is correct, and that therefore it is sufficient to knock down the protein in the glomerulus for which they say expression is higher and see if that produces a phenotype. In their model, Ptp10D in VA1v ORNs and Toll2 in VA1d PNs are not involved in these targeting decisions. But, since Ptp10D levels in VA1v ORNs are essentially the same as in VA1d, and Toll2 levels in VA1d PNs are essentially the same as in VA1v, they need to determine whether knocking down Ptp10D in VA1v ORNs and knocking down Toll2 in VA1d PNs produces a phenotype. If phenotypes are observed, this would argue that their model is incorrect or oversimplified.

Fig. 3.

- a. One might have expected that, if Ptp10D repulses Toll2-expressing axons, that the VA1d ORNs overexpressing Toll2 would innervate a glomerulus where PNs express low levels of Ptp10D. The fact that they target to the VA1d glomerulus but the VA1d PN dendrites now grow into other glomeruli suggests that the ORN axons are determined to grow to the VA1d glomerulus by other cues, and that once there they repel the VA1d PN dendrites. This is interesting.

b. Why should the VA1d PNs target to the DA41 glomerulus in the experiment of Fig. 3e, but to the DC3 glomerulus in the experiment of Fig. 3j? This needs to be explained.
c. In the text, they refer to “Simultaneous overexpression of Toll2 in VA1d-ORNs and knockdown of Ptp10D in VA1v-PNs”. Actually, they mean knockdown in VA1d-PNs.

Fig. 4

Since there is no detectable Fili in either VA1d or VA1v PNs, there is no reason to think that loss of Kek1 would specifically affect targeting of VA1v ORNs. Again, they are doing experiments based on the idea that their model is correct. They should knock down Kek1 in VA1d ORNs, or examine VA1d ORN targeting in a Kek1 mutant. I would think that this might also cause mistargeting of VA1d ORNs. Similarly, Kek1 OE in VA1v ORNs might also cause mistargeting.

Fig. 5

It does seem believable that Hbs and Sns levels are higher in VA1d PNs than in VA1v PNs. Kirre levels look essentially the same in VA1d and VA1v ORNs, however, so why should VA1d axons be able to innervate VA1d in wild-type, if Hbs/Sns levels are high there, given that when Kirre is knocked down in VA1v axons they now innervate VA1d? Loss of Kirre gives VA1v axons a preference for VA1d, but VA1d axons go there normally and they express Kirre.

Extended Data.

It is known that Kirre binds to Hbs and Sns. However, if it can't be shown that Ptp10D binds to Toll2, and that Fili binds to Kek1, it is hard to argue that these are repulsive cues. They might have indirect effects on targeting and have little to do with each other except that they influence signaling pathways that the other protein is involved in. The experiments shown in Extended Data 6c. are inadequate to determine whether these proteins bind to each other. First, Fc proteins almost always show binding to multiple partner proteins expressed in overlapping patterns, particularly in the brain, which is a thick and dirty tissue. These experiments should be done in embryo or larval dissections, where individual tissues (muscle, neuron, tracheae, etc.) are flat and spread out, so that they can be visualized separately from each other. Second, even in embryo or larval preps, unless these different binding proteins happen to be spatially segregated to different cells, one cannot expect to see a difference between wild-type and a LOF mutant for the candidate partner. This has only worked in one case that I know of. These experiments need to be done in a GOF manner, by overexpressing the candidate partner in one cell type (e.g. muscle) and looking for a change in the staining pattern. This should be done in both directions (protein A staining animals ectopically expressing protein B, and vice versa). If the bidirectional binding experiment works, one can tentatively conclude that A and B likely bind to each other even if binding cannot be detected in vitro. Even if binding is only observed in one direction, one can still conclude that A and B are likely to bind, but that a coreceptor or translational modification not present on the secreted Fc protein may be required for binding.

Referee #2

(Remarks to the Author)

The study by Dr. Luo and colleagues addresses the question of how neural circuits are precisely assembled during development. Here they focus on the roles of cell-surface proteins (CSPs) and take advantage the power of genetics to study the wiring logic in the *Drosophila* antennal lobe, where olfactory receptor neurons (ORNs) form one-to-one precise synaptic connection with projection neurons (PNs) in this olfactory circuit.

They uncovered the roles of three CSP pairs in mediating repulsive interactions between non-partner ORNs and PNs, paving way for the demonstration in the companion manuscript, in which they were able to genetically rewire the axons of one group of ORNs to a different glomerulus—a major advancement in the field. Specifically, they discovered that for a pair of CSPs, the expression of one molecule is high in the ORNs and PNs of one glomerulus, whereas the other molecule is highly expressed in the neighboring glomeruli. For example, Toll2 and Ptp10D are paired CSPs. Toll2 is highly expressed in the VA1d glomerulus, whereas Ptp10D is highly expressed in the VA1v, a neighboring glomerulus of VA1d. The repulsive interaction mediated by Toll2-Ptp10D ensures that VA1d ORNs avoids the non-cognate glomerulus. Overall, the study addresses an important gap in the field—the roles of CSPs in mediating repulsive interactions between pre- and post-synaptic neurons. This could be the last mile in establishing a complete picture of how precise synaptic connection is achieved during development. However, I have the following concerns that must be addressed.

Major concerns:

1. It is imperative to determine whether the function of the CSPs is limited to non-partner neurons. For example, the conclusion that “Ptp10D and Toll2 act in both PNs and ORNs to prevent mismatching between non-partners” (page 3 line 3 from bottom) needs more control experiments. First, it is necessary to examine whether Ptp10D knockdown in VA1d PNs causes any mistargeting of the partner ORNs, i.e., VA1d ORNs. Second, it is necessary to look at the arborization of VA1v PNs whether they exhibit any mistargeting phenotype when Toll2 is knocked down in VA1v ORNs.
2. Given that Toll2 or Ptp10D is expressed in the ORNs and PNs of the same glomerulus, it is important to test whether they have any role in homophilic attraction in the connection between partner ORNs and PNs. This question can be addressed, for example, by examining VA1v PNs when Toll2 is overexpressed in the VA1d ORNs.
3. These two concerns are also applicable to the other CSP pairs—Fili-Kek1 and Hbs/Sns-Kirre.

Referee #3

(Remarks to the Author)

In this study, the authors claim that repulsive interactions between cell surface proteins (CSPs) serve as a generalisable mechanism for synaptic partner matching during developmental wiring of neural networks. Their analysis focuses on three pairs of CSPs that restrict the targeting of olfactory receptor neurons and olfactory projection neurons to one of two specific glomeruli in the *Drosophila* antennal lobe. The experimental evidence for the action of these CSP pairs for appropriate targeting of these synaptic partners is solid, comprising expression pattern analysis, loss and gain of function, and suppression through genetic manipulation of the synaptic partner. However, as currently presented, it is perhaps too focused and too dependent on expertise in *Drosophila* developmental genetics to be suitable for a broad readership.

I would strongly recommend addressing this issue in the following ways:

1. Place more emphasis on the novelty and generalisability of the findings.

The authors analysed transcriptome data to identify two pairs of CSPs that appear to mediate repulsion without direct interaction. How unexpected was this? Did any of the other genes differentially expressed in VA1d or VA1v ORNs or PNs suggest plausible mechanisms for this repulsion, even if testing them would be beyond the scope of this study?

Also, at present, the fly brain-wide expression patterns are relegated to supplemental videos. There is space in the final figure to include a still image from each of the videos, highlighting some interesting aspect(s) of differential gene expression, before focusing on the optic lobe layers. The central brain expression patterns are very suggestive and will be of interest to a wider readership, ideally prompting future experiments by experts in those other neuropils.

2. Make the narrative as clear and accessible as possible.

The abstract is clear and succinct (although it omits the novel point that two of the three CSP pairs do not appear to interact directly). However, I identified several possible errors in the figures and accompanying text. There are also places where more information could be provided for clarity.

Fig. 1j-l What is the significance of cyan vs red? Should the 2nd column in 1k be red for Kek1 instead of cyan, or should the label read "Fili in PNs"? Should the 2nd column in 1l be darker blue for Sns instead of cyan?

Fig. 1 Legend: c-i Should this read "red in c, e, g" and "cyan in d, f, h, i"?)

Extended Data Figure 3

3a VA1d ORN and VA1d PN panels appear to have been swapped

Main text (last paragraph of Trans-cellular interactions of Toll2 and Ptp10D): "Simultaneous overexpression of Toll2 in VA1d-ORNs and knockdown of Ptp10D in VA1v-PNs suppressed the VA1d-PN dendrite phenotypes caused by Toll2 overexpression alone" - I believe that this should be "knockdown of Ptp10D in VA1d-PNs"

Fig. 6b Please clarify: Is the ssRNAseq data from ORNs, PNs, both, etc?

Please add a gene label, either to each file name or to the beginning of each supplemental movie. Perhaps this was done originally as indicated in the accompanying legend, but at present there is nothing to indicate which gene's expression is shown.

Fig. 6c Could different styles of arrowheads or other symbols be used to draw the reader's attention to the specific differences in expression that you believe could prove significant for targeting?

Extended Data Figure 6. The authors cite other studies that support direct binding of Kirre to Hbs/Sns, but they did not detect direct binding between Fili and Kek1 or between Toll2 and Ptp10, so the biochemical basis for their repulsion remains unknown. Please show a positive control for this method in your hands, for example the binding of Kirre to Hbs/Sns.

Discussion: "site choice" not "cite choice" in 2nd paragraph

Version 1:

Reviewer comments:

Referee #1

(Remarks to the Author)

I thought that there were a lot of problems with this paper, in that the authors present subtle differences in expression between different ORNs and PNs, and then assume that this means that the proteins must function in the ORN and PN groups that have higher expression. However, they have done a lot of experiments in response to my review, and they all appear to show that the relative expression levels are what determine which proteins control synapse formation. That is, even if the differences are small, they can be read by the axons and dendrites to control innervation.

It is disappointing that none of the proposed new interactions can be detected in vitro, but they did an extensive set of

experiments in response to my review to test out some alternative methods to demonstrate binding. None of these worked, however. Thus, we are left with the idea that the proteins mediate repulsive interactions, but cannot be demonstrated to bind to each other due to other factors, such as a requirement for a coreceptor or for some specific protein modification, such as a particular kind of glycosylation. We have obtained similar results for other pairs of proteins that clearly function together in vivo, but can't be demonstrated to bind in vitro.

In summary, the authors appear to have adequately addressed my criticisms, and the paper could be published in its present form.

Referee #2

(Remarks to the Author)

The authors have successfully addressed my previous concerns.

Referee #3

(Remarks to the Author)

In this study, the authors claim that repulsive interactions between cell surface proteins (CSPs) serve as a generalisable mechanism for precise synaptic partner matching during developmental wiring of neural networks. Their analysis focuses on three pairs of CSPs that restrict the targeting of olfactory receptor neurons and olfactory projection neurons to one of two specific glomeruli in the *Drosophila* antennal lobe. The experimental evidence for the action of these CSP pairs for appropriate targeting of these synaptic partners is extensive, comprising in vitro assays, expression pattern analysis, loss and gain of function, and suppression through genetic manipulation of the synaptic partner. There is also evidence for inversely correlated expression of these CSP pairs in other brain neuropils including the optic lobes, central complex, and suboesophageal zone, suggesting that repulsive interactions are indeed deployed as a generalisable mechanism for synaptic partner matching.

In response to reviewer feedback following the initial submission, the authors performed and reported numerous controls to support specific interactions between each pair of CSPs. They also corrected errors in the figures and legends and revised the main text to provide a clearer narrative for the non-expert reader. I agree that my earlier concerns have been addressed by the resulting revised manuscript together with the rebuttal to reviewers, although there are many minor infelicities of language that would benefit from careful copyediting.

We appreciate the compliments from all reviewers on the importance and general interest of our study, as well as their constructive criticisms. Below, we first outline two major revisions. We then follow with a point-by-point response to all comments by the reviewers.

1. To address specific critiques from Reviewers #1 and #2, we have performed many additional genetic manipulation experiments. We combined results in which we did not observe a phenotype, along with other experiments we previously performed, in a new Extended Data Table 2. We copy it here for convenience, with the new experiments we performed during revision highlighted in green. We will refer to it repeatedly in our point-by-point response below.

Extended Data Table 2 | Summary of genetic manipulation experiments that did not produce mistargeting phenotypes.

Exp. #	Genetic manipulation	Cell types examined	m/n ¹	Interpretation
1	Ptp10D ^{-Y}	VA1v-ORN axons	0/18	Ptp10D is not required for VA1v-ORN axon targeting
2	VA1d/DA1-PN > Ptp10D RNAi	VA1d-ORN axons	0/14	Ptp10D does not affect targeting of partner neurons
3	VA1v-ORN > Ptp10D RNAi	VA1v-ORN axons	0/10	Ptp10D does not function in VA1v-ORNs for their axon targeting
4	VA1d/DA1-PN > Toll2 RNAi	VA1d-ORN axons	0/12	Toll2 does not affect targeting of partner neurons
5	VA1d/DA1-PN > Toll2 RNAi	VA1v-ORN axons	0/12	Toll2 does not function in VA1d-PNs to regulate VA1v-ORN axon targeting
6	VA1d/DA1-PN > Toll2 RNAi	VA1d/DA1 PN dendrites	0/12	Toll2 does not function in VA1d-PNs to regulate their dendrite targeting
7	pan-ORN > Toll2 RNAi	VA1v-PN dendrites	0/10	Toll2 does not affect targeting of partner neurons
8	VA1v-ORN > Toll2 RNAi	VA1v-PN dendrites	0/10	Toll2 does not affect targeting of partner neurons
9	VA1d-ORN > Toll2 OE	VA1v-PN dendrites	0/11	Toll2 unlikely mediates homophilic attraction
10	VA1d/DA1-PN > Fili RNAi	VA1d-ORN axons	0/14	Fili does not affect targeting of partner neurons
11	kek1 ^{-/-}	VA1d-ORN axons	0/18	Kek1 is not required for VA1d-ORN axon targeting
12	pan-ORN > kek1 RNAi	VA1d-ORN axons	0/10	Kek1 is not required in ORNs for in VA1d-ORN axon targeting
13	VA1d-ORN > kek1 RNAi	VA1d-ORN axons	0/10	Kek1 is not required in VA1d-ORNs for their axon targeting
14	pan-ORN > kek1 RNAi	VA1v-PN dendrites	0/11	Kek1 does not affect targeting of partner neurons
15	VA1v-ORN > kek1 RNAi	VA1v-PN dendrites	0/10	Kek1 does not affect targeting of partner neurons

16	VA1d-ORN > Kek1 OE	VA1v-PN dendrites	0/10	Kek1 unlikely mediates homophilic attraction
17	VA1v-ORN > Kek1 OE	VA1v-ORN axons	0/10	Kek1 endogenous expression level is higher in VA1v-ORNs than VA1d-ORNs and overexpression of Kek1 in VA1v-ORNs does not cause ORN axon mistargeting phenotype
18	hbs ^{+/-}	VA1d-ORN axons	0/10	Mutant alone does not have any VA1d-ORN axons mistargeting phenotype (Control for Fig. 5)
19	hbs ^{+/-} , sns ^{+/-}	VA1d-ORN axons	0/10	Mutant alone does not have any VA1d-ORN axons mistargeting phenotype (Control for Fig. 5)
20	VA1d/DA1-PN > hbs RNAi	VA1d-ORN axons	0/10	Hbs does not affect targeting of partner neurons
21	sns ^{+/-}	VA1d-ORN axons	0/10	Mutant alone does not have any VA1d-ORN axons mistargeting phenotype (Control for Fig. 5)
22	VA1d/DA1-PN > sns RNAi	VA1d-ORN axons	0/10	Sns does not affect targeting of partner neurons
23	kirre ^{-Y}	VA1d-ORN axons	0/10	Mutant alone does not have any VA1d-ORN axons mistargeting phenotype (Control for Fig. 5)
24	pan-ORN > kirre RNAi	VA1d-ORN axons	0/12	Kirre is not required in ORNs for VA1d-ORN axon targeting
25	VA1d-ORN > kirre RNAi	VA1d-ORN axons	0/10	Kirre is not required in VA1d-ORNs for their axon targeting
26	pan-ORN > kirre RNAi	VA1v-PN dendrites	0/15	Kirre does not affect targeting of partner neurons
27	VA1v-ORN > kirre RNAi	VA1v-PN dendrites	0/10	Kirre does not affect targeting of partner neurons

¹ m = number of antennal lobes with mistargeting phenotypes; n = number of total antennal lobes examined.

Besides the results of genetic manipulation experiments shown in the figures, additional results with no mistargeting observed in certain ORN axons and PN dendrites are listed in this table. Collectively, these results argue against the hypotheses alternative to our model (Fig. 6a) and suggest that: (1) the CSPs do not function in neurons with relatively low expression level; (2) the function of these CSPs are mainly to regulate interactions between non-partner neurons; (3) the CSPs unlikely mediate homophilic attraction. Despite the evidence supporting that the CSP pairs mediate repulsion between non-partner neurons, we cannot completely rule out the possibility that some of the CSPs have additional roles. As an example, the mouse teneurin-3 mediates both homophilic attraction with itself (Berns et al., 2018) and heterophilic repulsion with latrophilin-2 (Pederick et al., 2021) in the hippocampal networks.

2. At the suggestions of Reviewer #3, we worked systematically to streamline our manuscript, from main text to main and Extended Data figures and their legends, to ensure that our study can be appreciated by a general audience while also being informative to experts. To enhance the readability and shorten the manuscript, we placed many technical details in the Methods section and the legends of Extended Data Figures. In the process, we have also addressed some critiques from reviewers due to misunderstandings caused by dense writing of our original manuscript.

Point-by-point responses (original review copied in blue):

Referee #1 (Remarks to the Author):

This is an interesting paper describing multiple repulsive signaling cues controlling innervation of antennal lobe glomeruli. It is a companion to an excellent paper using these and other proteins to alter glomerular targeting in a programmed manner. This manuscript, however, has many problems. The experiments seem to be constructed assuming that the preferred model is correct, even though the differential expression of the repulsive cues between neurons is not convincing in most cases. Also, because they cannot demonstrate direct interactions between the proteins, it is possible that the repulsive effects are not interactions between these specific proteins, but indirect effects on signaling pathways in which they participate.

We thank the reviewer for appreciating our study. Regarding the critique that “the experiments seem to be constructed assuming that the preferred model is correct,” we want to clarify that we started this study by performing pan-neuronal, pan-PN, or pan-ORN RNAi experiments (where both VA1d and VA1v are manipulated). At that time, we did not have any specific hypothesis, so we examined multiple cell types, but found mistargeting of ORN axons and PN dendrites only in specific cell types. For example, when we performed pan-ORN knockdown of *Ptp10D*, we observed VA1d-ORN axons mistargeting to the VA1v glomerulus (Extended Data Fig. 1), whereas VA1v-ORN axons and VA1d-PN dendrites targeted normally.

To address Reviewer #1’s concern directly, we tested whether manipulating the CSPs in other cell types caused any phenotypes. These additional data are summarized in the new Extended Data Table 2. These new experiments confirm that the targeting phenotypes we described in the original manuscript are specific to those cell types (see details in our specific responses).

We also performed another experiment to test the specificity of genetic interactions. As Fili-Kek1 and Hbs/Sns-Kirre have a similar function in preventing misconnection of VA1v-ORNs with VA1d-PNs, we tested the specificity of these genetic suppression experiments. We found that reducing the dose of *sns* and *hbs* did not suppress the mistargeting phenotype caused by Kek1 overexpression, suggesting that Fili-Kek1 and Hbs/Sns-Kirre participate in different pathways and confirming the specificity of the trans-cellular assay. This experiment is described in the new Extended Data Fig. 6.

Regarding the critique on direction interaction vs. indirect effects, please see our response to the more specific critiques the reviewer raised below.

Fig. 1.

a. Most of these differences in protein levels are not convincing in the images shown, even though they were presumably selected as demonstrations of the data reported in panel j. I see almost no difference between Ptp10D levels or Kirre levels in VA1d and VA1v ORNs. There are no detectable differences between Toll2 levels, Kek1 levels, and Kirre levels between VA1d and VA1v PNs. They also can't see Fili expression in PNs but say that it is there based on another study. There may be more differences in RNA levels, but protein levels are not strongly correlated with RNA levels measured in single-cell sequencing experiments. About all that one can say about protein levels based on RNA counts is that the protein is probably expressed at some level in a cell that has RNA.

We wish to clarify that for the representative images, we always chose brains that had preference indexes near the average and showed single optical sections where both VA1d and VA1v glomeruli occupy similar sizes. We agree with the reviewer that differential expression of some of the proteins is not very apparent based on single confocal sections shown. Nevertheless, the differences are statistically significant based on the quantification of fluorescence intensity in the 3D glomerular volume, which is more accurate than single section sample images. Specifically, the preference index for Ptp10D or Kirre in ORNs is significantly above or below zero, respectively, indicating that Ptp10D is preferentially expressed in VA1d-ORNs compared to VA1v-ORNs, whereas Kirre is preferentially expressed in VA1v-ORNs compared to VA1d-ORNs. It seems that these small but significant differences can be detected and used for partner matching, as indicated by our genetic analysis. In other words, synaptic partner choice is likely regulated by the relative levels of repulsive interactions. We added the following sentence in Discussion in the revised manuscript to emphasize this point: *“Some of the CSPs only have modest differential expression, suggesting that synaptic partner choice is likely regulated by the relative levels of repulsive interactions.”*

We agree with the reviewer that there is no detectable difference of Kek1 levels and Kirre levels between VA1d and VA1v PNs (the preference indexes are not significantly different from 0), so we did not draw any conclusions from them. We note that protein expression levels in PNs are usually weaker and harder to detect compared to ORNs, as VA1d- and VA1v-PNs consist of a few cells each, while VA1d- and VA1v-ORNs consist of several dozen cells each.

We also agree with the reviewers that RNA expression levels cannot accurately predict protein expression levels, even though there is a weak positive correlation (see Figure 3F in Li et al., 2020). We included the preference index for RNA levels because they provide information about differential expression at earlier developmental stages (than the protein staining data we have), which is more relevant for the synaptic partner matching process. Because of technical limitations, the earliest developmental stages we can

distinguish VA1d and VA1v neurites for protein staining is 42–48 h after puparium formation (APF) when nascent glomeruli emerge—i.e., when matching between PN dendrites and ORN axons is nearly complete. The RNA levels were measured at 24–30h APF, shortly before the start of the matching between PN dendrites and ORN axons. We found that for ORNs or PNs that have preference index of a specific protein significantly different from 0, the preference index of the corresponding RNA usually has the same sign as the protein. Thus, these data complement each other. We have explained some of these limitations at the end of Extended Data Fig. 2 legend in the original manuscript.

In the revised manuscript, we have clarified some of the above points in Fig. 1 legend and refer readers to Extended Data Fig. 2 legend for further explanation. We have also reworded the original text in Extended Data Fig. 2 legend to make our points clearer. Furthermore, we have moved some of the original images to Extended Data Fig. 2, such that we can (1) streamline the main text/figure; (2) make the figure panels larger to visualize the staining more clearly.

b. “cyan in e, g, i, j and red in d, f, h.” Actually it is red in c, e, g and cyan in d, f, h, i. J is a graph.

We thank the reviewer for pointing this out. We have corrected the errors in our revised manuscript.

Fig. 2.

a. These phenotypes are not impressive, if I am reading the figure legend correctly. Mistargeting is scored if one axon or dendrite per sample is mistargeted, so in a case like Ppt10D VA1d ORN RNAi, where it is indicated as 30%, that presumably means that in 70% of the samples not a single axon mistargets.

The reviewer read our data correctly, and we agree that mistargeting phenotypes in RNAi manipulation for some of the individual genes are subtle. This is consistent with many of the single-gene manipulation experiments in the companion manuscript. As we stated in the companion manuscript: “*disrupting individual CSPs, even with complete loss-of-function mutations, usually leads to partial phenotypes at specific wiring steps, particularly in synaptic partner selection, suggesting considerable redundancy*”. In support of this, we showed in the companion manuscript that by manipulating multiple CSPs, the mistargeting phenotype can be much stronger (companion manuscript Fig. 2).

To further experimentally verify the phenotype, we expressed the presynaptic active zone marker Bruchpilot-Short to examine whether mistargeted VA1d-ORN axons, although small in quantity, likely form synaptic connections in the VA1v glomerulus. The results

showed that Bruchpilot-Short is present in the mistargeted ORN axons when we manipulated each of three CSP pairs, suggesting that these mistargeted axons form synaptic connections with new PN dendrite partners. We summarized these results in the new Extended Data Fig. 7 in the revised manuscript.

b. They don't say which RNAis are used in the images shown. Also, isn't it Toll $-/-$, not Toll $+/-$, in k? Panel f says $-/-$.

We apologize for not making the experimental conditions clearer. We included the code for the RNAi lines as well as reagents for mutant and overexpression experiments in Extended Data Table 3, and the example images come from one of the RNAi lines being quantified in Fig. 2k and l.

The only exception is the *Toll2* mutant experiment. While we obtained $n > 10$ brains for *Toll2* $^{+/-}$ and about half of them showed the VA1d-ORN mistargeting phenotype as quantified in Fig. 2k, we only had $n = 2$ after collecting hundreds of flies for *Toll2* $^{-/-}$ with 1 out of 2 showing the VA1d-ORN mistargeting phenotype. The sickness of the mutant lines is consistent with what was reported previously in Eldon et al., 1994.

In the revised manuscript, we changed the example images to a *Toll2* $^{+/-}$ brain to be consistent with the quantification. We also removed the data from *Toll2* $^{-/-}$ brains because of the small n .

c. Since the differences in Ptp10D levels between VA1d and VA1v ORNs are very small (Fig. 1), as are the differences in Toll2 levels in VA1d and VA1v PNs, it is not justified to confine the RNAi experiments to VA1d ORNs for Ptp10D and VA1v PNs for Toll2. The authors are assuming that their model is correct, and that therefore it is sufficient to knock down the protein in the glomerulus for which they say expression is higher and see if that produces a phenotype. In their model, Ptp10D in VA1v ORNs and Toll2 in VA1d PNs are not involved in these targeting decisions. But, since Ptp10D levels in VA1v ORNs are essentially the same as in VA1d, and Toll2 levels in VA1d PNs are essentially the same as in VA1v, they need to determine whether knocking down Ptp10D in VA1v ORNs and knocking down Toll2 in VA1d PNs produces a phenotype. If phenotypes are observed, this would argue that their model is incorrect or oversimplified.

We performed a new set of experiments to address this concern from the reviewer. To examine whether Ptp10D in VA1v-ORNs is involved, we examined VA1v-ORN axons in *Ptp10D* mutant animals (Extended Data Table 2 Exp. #1) or when *Ptp10D* was knocked down in VA1v-ORNs (Extended Data Table 2 Exp. #3). No VA1v-ORN axon mistargeting was observed in any of the conditions. Similarly, we examined whether Toll2 in VA1d-PNs is

involved by knocking down *Toll2* in VA1d/DA1-PNs. No mistargeting was observed in VA1d-ORN axons (Extended Data Table 2 Exp. #4), VA1v-ORN axons (Extended Data Table 2 Exp. #5), or VA1d/DA1 PN dendrites (Extended Data Table 2 Exp. #6). These results support our repulsive model and suggest that the relative expression levels are more relevant to the repulsive function than whether the protein is expressed or not. We summarized the results in the new Extended Data Table 2.

Fig. 3.

a. One might have expected that, if Ptp10D repulses Toll2-expressing axons, that the VA1d ORNs overexpressing Toll2 would innervate a glomerulus where PNs express low levels of Ptp10D. The fact that they target to the VA1d glomerulus but the VA1d PN dendrites now grow into other glomeruli suggests that the ORN axons are determined to grow to the VA1d glomerulus by other cues, and that once there they repel the VA1d PN dendrites. This is interesting.

We thank the reviewer for raising this interesting point. Indeed, we found that in all experiments done so far, including both loss-of-function and gain-of-function experiments, Toll2 always *send* the repulsive signal and Ptp10D always *receive* the repulsive signal, regardless of whether PNs or ORNs express them. An example is that the knockdown of *Toll2* in VA1v-ORN caused VA1d-PN mistargeting to VA1v (Fig. 2j), while VA1v-ORN axons remain in the VA1v glomerulus (Extended Data Fig. 3e). In the revised manuscript, we added “with Toll2 sending and Ptp10D receiving a repulsive signal to non-partner neurons in both PNs and ORNs” at the conclusion of Fig. 2 to emphasize this point.

b. Why should the VA1d PNs target to the DA41 glomerulus in the experiment of Fig. 3e, but to the DC3 glomerulus in the experiment of Fig. 3j? This needs to be explained.

We explained this difference in Figure 3 legend and Methods. We used GAL4/UAS system to overexpress Toll2 by the *P{GSV1}18wEP-709* line, but LexA/LexAop system to overexpress Toll2 by cDNA lines we made. We suggest that Toll2 expression level differences from using different binary expression systems and reagents may have caused them to target to different glomeruli, especially considering that there are other CSPs working in concert with Toll2 as suggested by the companion manuscript. This phenomenon was also reported previously in Xu et al., 2024, where we raised flies at different temperatures to adjust the overexpression levels of Ten-m and observed a level-dependent difference in the mistargeting phenotypes. We note that DA4l-ORNs and DC3-ORNs both express low levels of Toll2 as suggested by the scRNA-seq data (Extended Data Fig. 2a), which is consistent with our model where Ptp10D-high VA1d-PNs mistarget to Toll2-low ORN types.

In the revised manuscript, we have updated the figure legend “*The different mistargeting regions in (b, e) and (g, j) likely result from different Toll2 overexpression levels using different binary systems (Methods).*”

c. In the text, they refer to “Simultaneous overexpression of Toll2 in VA1d-ORNs and knockdown of Ptp10D in VA1v-PNs”. Actually, they mean knockdown in VA1d-PNs.

We thank the reviewer for pointing out this typo and we have corrected it in the revised manuscript.

Fig. 4

Since there is no detectable Fili in either VA1d or VA1v PNs, there is no reason to think that loss of Kek1 would specifically affect targeting of VA1v ORNs. Again, they are doing experiments based on the idea that their model is correct. They should knock down Kek1 in VA1d ORNs, or examine VA1d ORN targeting in a Kek1 mutant. I would think that this might also cause mistargeting of VA1d ORNs. Similarly, Kek1 OE in VA1v ORNs might also cause mistargeting.

The functions of Fili in VA1d-PNs were characterized in detail in Xie et al., 2019. When *Fili* is specifically knocked out or knocked down in VA1d/DC3-PNs, VA1v-ORN axons mistarget to the VA1d glomerulus whereas VA1d-ORN axons remain intact (Xie et al., 2019, Figure 5, E and G). These results suggest Fili expressed in VA1d-PNs is sending a repulsive signal to VA1v-ORNs but not an attractive signal to VA1d-ORNs. This inspired us to identify CSPs that have higher expression levels in VA1v-ORNs than VA1d-ORNs as the potential partner of Fili. As we did not repeat these Fili experiments in our manuscript, we added more detailed descriptions of the crucial experiments and the results from Xie et al., 2019 in the section for Fili and Kek1.

To test Kek1’s function without any bias, as the reviewer suggested, we performed the following experiments: in *kek1* mutant animals, we labeled VA1d-ORNs and did not observe any mistargeting phenotype of their axons (Extended Data Table 2 Exp #11); when we knocked down *kek1* in VA1d-ORNs, we did not observe mistargeting phenotypes of VA1d-ORN axons (Extended Data Table 2 Exp #13); when we overexpressed Kek1 in VA1v-ORNs, we also did not observed mistargeting phenotypes of VA1v-ORN axons (Extended Data Table 2 Exp #17). These results are summarized in the new Extended Data Table 2.

Fig. 5

It does seem believable that Hbs and Sns levels are higher in VA1d PNs than in VA1v PNs. Kirre levels look essentially the same in VA1d and VA1v ORNs, however, so why should

VA1d axons be able to innervate VA1d in wild-type, if Hbs/Sns levels are high there, given that when Kirre is knocked down in VA1v axons they now innervate VA1d? Loss of Kirre gives VA1v axons a preference for VA1d, but VA1d axons go there normally and they express Kirre.

As we stated in our response to reviewer's critiques on Fig. 1, the level of Kirre is significantly higher in VA1v-ORNs than VA1d-ORNs in volume-based quantification. Based on our results, we suggest that it is the relative expression level instead of the absolute expression level that is more relevant to the loss-of-function phenotypes.

To experimentally address this concern, we labeled VA1d-ORNs in *kirre* mutant, knocked down *kirre* in all ORNs, or knocked down *kirre* in VA1d-ORNs. None of the manipulations resulted in mistargeting phenotypes of VA1d-ORN axons (Extended Data Table 2 Exp. #23–25).

Extended Data.

It is known that Kirre binds to Hbs and Sns. However, if it can't be shown that Ptp10D binds to Toll2, and that Fili binds to Kek1, it is hard to argue that these are repulsive cues. They might have indirect effects on targeting and have little to do with each other except that they influence signaling pathways that the other protein is involved in. The experiments shown in Extended Data 6c. are inadequate to determine whether these proteins bind to each other. First, Fc proteins almost always show binding to multiple partner proteins expressed in overlapping patterns, particularly in the brain, which is a thick and dirty tissue. These experiments should be done in embryo or larval dissections, where individual tissues (muscle, neuron, tracheae, etc.) are flat and spread out, so that they can be visualized separately from each other. Second, even in embryo or larval preps, unless these different binding proteins happen to be spatially segregated to different cells, one cannot expect to see a difference between wild-type and a LOF mutant for the candidate partner. This has only worked in one case that I know of. These experiments need to be done in a GOF manner, by overexpressing the candidate partner in one cell type (e.g. muscle) and looking for a change in the staining pattern. This should be done in both directions (protein A staining animals ectopically expressing protein B, and vice versa). If the bidirectional binding experiment works, one can tentatively conclude that A and B likely bind to each other even if binding cannot be detected in vitro. Even if binding is only observed in one direction, one can still conclude that A and B are likely to bind, but that a coreceptor or translational modification not present on the secreted Fc protein may be required for binding.

We greatly appreciate the detailed suggestions from the reviewer for the binding

experiments. The method we used in the original manuscript is based on a body of previous studies: (1) a deficiency screen using secreted fusion protein in live embryos successfully identified a ligand for RPTP LAR (Fox and Zinn, 2005); (2) an ectopic expression screen (where Toll2 is also one of the candidates that did not show binding) using similar strategy identified a ligand for Ptp10D (Lee et al., 2013); and (3) a study using 48-hour pupal brains and 0–24 hour adult brains for live tissue staining, which is the protocol we followed, revealed the sexual dimorphism of Dpr and DIP in the brain (Brovero et al., 2021). We chose to use the 48-hour pupal brains for the live staining based on the hypothesis that if there is an unidentified co-factor, it is likely to be present in 48-hour pupal brains where these CSPs normally function.

In response to the suggestion of the reviewer that overexpression assays may increase sensitivity for the binding assay, we overexpressed Kek1 and Ptp10D in DM6/DL4-ORNs or DA1-ORNs and performed live staining experiments using Fili-Fc-V5 and Toll2-Fc-V5 in 48-hour pupal brains. We still did not find any detectable differences in binding to brains with Kek1 or Ptp10D overexpression compared to wild-type brains.

To further validate our method, we stained wild-type (*kirre-ctag* flies) and *Kirre* mutant pupal brains with Sns-Fc-V5 as controls. In *kirre>HA>Myc* pupal brains, Sns-Fc-V5 showed binding in the antennal lobe with strong and specific pattern, and this pattern was not seen in the *Kirre* mutant pupal brains. This control is added as the new Extended Data Fig. 9a.

Following the reviewer's suggestion that the brain might have too high background for live staining, we performed several additional binding experiments using simpler tissues with CSP overexpression. The first tissue we tried was the wing disc (based on our previous protocol in Sweeney et al., 2011). We performed the live staining experiment in the wing disc dissected from 3rd instar larvae, where *engrailed (en)-GAL4* was used to drive ectopic expression of candidate CSPs in the posterior compartment of the wing disc. When we applied Sns-Fc-V5 for live staining in the *en-GAL4 > UAS-Kirre-HA* wing disc, we observed binding of Sns-Fc-V5 specifically to the posterior compartment (Extended Data Fig. 9d), thus validating the binding between Sns and Kirre. (Interestingly, Sns seems to bind strongest to regions with intermediate but not highest levels of Kirre-HA expression.) Using the same assay, we tested the binding between Toll2 and Ptp10D, and between Fili and Kek1. Toll2-Fc-V5 bound to some endogenous protein strongly throughout the wing disc, making it difficult to see the changes due to Ptp10D overexpression (Extended Data Fig. 9f). No detectable binding was observed for Fili-Fc-V5 in control or wing disc overexpressing Kek1 (Extended Data Fig. 9e). The results from this assay are consistent with the results we obtained using the pupal brain binding assay. Together, they suggest that Toll2 and Ptp10D do not bind in reduced conditions; neither do Fili and Kek1. Results from this new assay are added as new panels to Extended Data Fig. 9.

A second tissue we tried was the neuromuscular junction (NMJ). We performed the binding assay in the 3rd larvae dissected body wall by overexpressing the candidate partner in the motor neurons using *D42-GAL4*, and we did not observe apparent difference in overexpression vs. wild type even for Kirre and Sns, likely because the background is too high.

Although we only showed the results with CSPs playing cell-autonomous roles (Ptp10D, Kek1, and Kirre as shown in Fig. 2e, Fig. 4h, and Fig. 5l respectively) being expressed in the brain, we indeed tried to do both directions of binding. However, we cannot get high-quality expression of Ptp10D because of the size of the protein and the aggregation effect. When we used Kek1-Fc-V5 to perform the live binding in the brain, we did not observe any signal throughout the brain, so we did not use it for the LOF experiment. In addition, as we do not have any tagged Fili overexpression reagent (unlike the HA staining we showed in Extended Data Fig. 9 for Kirre, Ptp10D, Kek1), it is difficult to perform convincing live staining experiment using Kek1-Fc-V5 when we cannot visualize cells overexpressing Fili.

Finally, we added new experiments using an additional *in vitro* assay, where oligomerized ectodomains are tested in a bait-prey format, which were successfully used before for the discovery of cognates of neuronal surface receptors and ligands (Özkan et al., 2013). Using this methodology, we confirmed binding between Sns and Kirre, and Hbs and Kirre as expected (Extended Data Fig. 8). Furthermore, we observed binding between Kek1 and its previously identified partner, EGFR (Ghiglione et al., 1999). Despite these positive results, we observed no binding between Kek1 and Fili, or between Ptp10D and Toll2 (Extended Data Fig. 8).

In summary, despite all the *in vitro* assays we tried so far, whether the Toll2–Ptp10D or the Fili–Kek1 CSP pairs bind to each other as ligand–receptor pairs are still open questions. That’s the reason we used “interaction” instead of “ligand-receptor” even though Ptp10D, Kek1 and Kirre showed “receptor-like” function based on the single neuron manipulation experiments. As Reviewer 3 mentioned “the novel point that two of the three CSP pairs do not appear to interact directly”, we believe that our transcriptome-informed *in vivo* screening complements traditional binding assays in identifying CSP pairs regulating neuronal wiring, especially if there are co-receptors, post-translational modifications, or specific physiological conditions like multimerization to make it difficult to show binding in reduced preparations. We have emphasized this point in Discussion.

Several lines of evidence strongly suggest that the CSP pairs we identified are highly likely to be major players with each other: (1) they are transmembrane proteins and thus less likely to be downstream signaling molecules, which are usually cytosolic proteins; (2) our genetic screens did not yield consistent phenotypes for other CSPs that exhibit inverse expression of a candidate CSP of interest (Extended Data Table 1), suggesting that the CSP

pairs we reported are the most likely ones; (3) in the trans-cellular experiments, for example, *Ptp10D* mutant suppressed the Toll2 overexpression phenotype to a wild-type level (Fig. 3e), suggesting that Ptp10D is likely to be the key repulsive partner of Toll2. If Ptp10D only regulates the Toll2 signaling pathway, it is unlikely to have such a strong suppression effect.

Referee #2 (Remarks to the Author):

The study by Dr. Luo and colleagues addresses the question of how neural circuits are precisely assembled during development. Here they focus on the roles of cell-surface proteins (CSPs) and take advantage the power of genetics to study the wiring logic in the *Drosophila* antennal lobe, where olfactory receptor neurons (ORNs) form one-to-one precise synaptic connection with projection neurons (PNs) in this olfactory circuit.

They uncovered the roles of three CSP pairs in mediating repulsive interactions between non-partner ORNs and PNs, paving way for the demonstration in the companion manuscript, in which they were able to genetically rewire the axons of one group of ORNs to a different glomerulus—a major advancement in the field. Specifically, they discovered that for a pair of CSPs, the expression of one molecule is high in the ORNs and PNs of one glomerulus, whereas the other molecule is highly expressed in the neighboring glomeruli. For example, Toll2 and Ptp10D are paired CSPs. Toll2 is highly expressed in the VA1d glomerulus, whereas Ptp10D is highly expressed in the VA1v, a neighboring glomerulus of VA1d. The repulsive interaction mediated by Toll2-Ptp10D ensures that VA1d ORNs avoids the non-cognate glomerulus. Overall, the study addresses an important gap in the field—the roles of CSPs in mediating repulsive interactions between pre- and post-synaptic neurons. This could be the last mile in establishing a complete picture of how precise synaptic connection is achieved during development. However, I have the following concerns that must be addressed.

Major concerns:

1. It is imperative to determine whether the function of the CSPs is limited to non-partner neurons. For example, the conclusion that “Ptp10D and Toll2 act in both PNs and ORNs to prevent mismatching between non-partners” (page 3 line 3 from bottom) needs more control experiments. First, it is necessary to examine whether Ptp10D knockdown in VA1d PNs causes any mistargeting of the partner ORNs, i.e., VA1d ORNs. Second, it is necessary to look at the arborization of VA1v PNs whether they exhibit any mistargeting phenotype when Toll2 is knocked down in VA1v ORNs.

We thank the reviewer for the suggestion to test whether partner neurons are influenced. If the partner neurons are affected by these manipulations, it could be caused by the loss of attractive interaction between partner neurons.

To test this, we performed a set of control experiments for the Toll2–Ptp10D pair. When we knocked down *Ptp10D* in VA1d-ORNs, VA1d-ORN axons mistarget to VA1v glomerulus whereas VA1d-PN dendrites do not have any mistargeting phenotype (Extended Data Fig. 3a), suggesting that Ptp10D in VA1d-ORNs does not affect VA1d-PNs.

For the other direction, we knocked down *Ptp10D* in VA1d/DA1-PNs and observed that both VA1d/DA1-PN dendrites and VA1d-ORN axons mistarget to VA1v glomeruli (Extended Data Fig. 3c). We think VA1d-ORN axon mistargeting is likely caused by a secondary effect to follow the mistargeted VA1d/DA1-PN dendrites because of the following reasons: (1) VA1d/DA1-PN dendrites mistargeting is more severe than VA1d-ORN axons mistargeting, and all the mistargeted VA1d-ORN axons stay together with VA1d/DA1-PN dendrites; (2) for the brains with no VA1d/DA1-PN dendrites mistargeting, we did not observe VA1d-ORN axons mistargeting.

Similarly, we used the pan-ORN driver or the VA1v-ORN specific driver to knock down *Toll2* and did not observe VA1v-PN axons mistargeting (Extended Data Table 2 Exp #7, 8).

We have also performed analogous experiments for the other 2 CSP pairs. No VA1d-ORN axons mistargeting phenotype was observed in mutant of *Fili*, *hbs*, or *sns* (Xie et al., 2019; Extended Data Table 2 Exp #18, 19, 21). We knocked down *Fili*, *hbs*, or *sns* in VA1d/DA1-PNs and did not observe any VA1d-ORN axons mistargeting phenotype (Extended Data Table 2 Exp #10, 20, 22). Furthermore, knocking down *kek1* or *kirre* in pan-ORNs or VA1v-ORNs did not cause any VA1v-PN dendrites mistargeting phenotype (Extended Data Table 2 Exp #14, 15, 26, 27). These results argue against *Fili*, *Hbs*, or *Sns* having a role in regulating the targeting of their synaptic partners in the VA1d and VA1v glomeruli. The results of these new experiments are also summarized in the new Extended Data Table 2.

2. Given that Toll2 or Ptp10D is expressed in the ORNs and PNs of the same glomerulus, it is important to test whether they have any role in homophilic attraction in the connection between partner ORNs and PNs. This question can be addressed, for example, by examining VA1v PNs when Toll2 is overexpressed in the VA1d ORNs.

We agree with the reviewer that based on the expression pattern of Toll2 and Ptp10D in VA1d and VA1v glomeruli alone, it is possible that they might mediate homophilic attraction. The evidence from our previous data and newly added data against homophilic attraction is as follows:

- (1) To test this possibility experimentally for Toll2, we overexpressed Toll2 in VA1d-ORNs. We did not observe any mistargeting phenotype for VA1d-ORN axons (Extended Data Fig. 5a) or VA1v-PN dendrites (where Toll2 is highly expressed; Extended Data Table 2 Exp #9), arguing against Toll2 playing a homophilic attraction role.
- (2) To test whether homophilic attraction between Ptp10D-expressing cells play a role in the wiring process, we focus on DC3-PNs which express both Toll2 and Ptp10D at a relatively high level (Extended Data Fig. 2a). We would expect that knocking down Ptp10D in a potential partner ORNs would decrease its matching with DC3-PNs if Ptp10D homophilic attraction played a more dominant role in synaptic partner matching, but would increase its matching with DC3-PNs if Ptp10D–Toll2 repulsion played a more dominant role. We used genetic manipulation of VA1d-ORNs to test this. *Ptp10D* knockdown in VA1d-ORNs alone led to their axons mistargeting to the VA1v glomerulus and not DC3. Inspired by the rewiring experiments (companion manuscript Fig. 5, CSP set 2), we sensitized VA1d-ORNs by overexpressing *Kek1* and knocking down *Ten-m*, and then knocked down *Ptp10D* in VA1d-ORNs. Comparing to *Kek1* OE + *Ten-m* RNAi only group, *Kek1* OE + *Ten-m* RNAi + *Ptp10D* RNAi resulted in significantly more mistargeting to DC3 glomerulus (new Extended Data Fig. 3f, g). This suggests that a decrease of Ptp10D in VA1d-ORN axons leads to more mistargeting to Ptp10D-high/Toll2-high DC3 glomerulus, which favors the heterophilic repulsion model of Ptp10D-low VA1d-ORN axons mistarget to Toll2-high DC3.
- (3) When we performed the trans-cellular assay (Figure 3), VA1d-PN dendrites mistargeting phenotype caused by Toll2 overexpression can be nearly fully suppressed by *Ptp10D* mutant, suggesting that Ptp10D's interaction with Toll2 is the major interaction that caused the phenotype we observed.
- (4) Previous Toll2 binding assays in cell lines suggest that Toll2 does not bind with itself in the *in vitro* assay (Paré et al., 2014) and might bind to other cell-expressing CSPs (Eldon et al., 1994).

For the Fili–Kek1 pair, previous work analyzed the expression pattern of Fili in ORNs and PNs and argued against Fili mediating homophilic interaction (Xie et al., 2019). As for *Kek1*, in the trans-cellular assay, *Fili* mutant nearly fully suppressed *Kek1* overexpression phenotypes, suggesting that *Fili*'s interaction with *Kek1* is the major contributor to the phenotype we observed. We further tested homophilic effect experimentally by labeling VA1v-PN dendrites (high *Kek1*) when overexpressing *Kek1* in VA1d-ORNs (endogenous low *Kek1*), and did not observe any mistargeting phenotype of VA1v-PN dendrites (Extended Data Table 2 Exp. #16).

For the Kirre–Hbs/Sns pair, since Hbs and Sns are minimally expressed in ORNs (Extended Data Fig. 2), they unlikely mediate homophilic attraction between PN and ORN partners. Furthermore, overexpressing Kirre specifically in VA1d-ORNs in *kirre* hemizygous mutant animals did not change the overexpression phenotype (Fig. 5p, q), arguing against a contribution for trans-cellular Kirre homophilic interaction in synaptic partner matching.

Despite the evidence, we cannot completely rule out the possibility that some of the CSPs have additional roles besides mediating repulsive interactions. As an example, our lab previously found that mouse teneurin-3 (Ten3) mediates both homophilic attraction with itself (Berns et al., 2018) and heterophilic repulsion with latrophilin-2 (Lphn2) (Pederick et al., 2021) in the hippocampal networks.

In the revised manuscript, we summarized these results and discussion in the new Extended Data Table 2 and its footnote.

3. These two concerns are also applicable to the other CSP pairs—Fili-Kek1 and Hbs/Sns-Kirre.

We already discussed all CSP pairs in points 1 and 2 above.

Referee #3 (Remarks to the Author):

In this study, the authors claim that repulsive interactions between cell surface proteins (CSPs) serve as a generalisable mechanism for synaptic partner matching during developmental wiring of neural networks. Their analysis focuses on three pairs of CSPs that restrict the targeting of olfactory receptor neurons and olfactory projection neurons to one of two specific glomeruli in the *Drosophila* antennal lobe. The experimental evidence for the action of these CSP pairs for appropriate targeting of these synaptic partners is solid, comprising expression pattern analysis, loss and gain of function, and suppression through genetic manipulation of the synaptic partner. However, as currently presented, it is perhaps too focused and too dependent on expertise in *Drosophila* developmental genetics to be suitable for a broad readership.

I would strongly recommend addressing this issue in the following ways:

1. Place more emphasis on the novelty and generalisability of the findings.

The authors analysed transcriptome data to identify two pairs of CSPs that appear to mediate repulsion without direct interaction. How unexpected was this? Did any of the other genes differentially expressed in VA1d or VA1v ORNs or PNs suggest plausible mechanisms for this repulsion, even if testing them would be beyond the scope of this study?

We thank the reviewer for appreciating our work. As mentioned at the beginning, we have streamlined our manuscript so our findings can be better appreciated by a broad readership. We have also emphasized the novelty and generalizability of our findings.

Regarding reviewer’s specific questions, we used Toll2 as an example to show how unexpected this was: (1) To find out Toll2’s repulsive interaction partner, we first narrowed down the candidates to CSPs only (~950 molecules). (2) Given that Toll2 is highly expressed in VA1v-ORNs and VA1v-PNs, its repulsive partner should be expressed lower in VA1v than VA1d in both PNs and ORNs during development. Based on the transcriptome data, about 120 candidate CSPs fulfill these criteria. (3) We used a simple model to calculate matching index between Toll2 and molecule A. We punished “high-high” expression and awarded “high-low” expression in each PN-ORN pair, and ranked the candidates based on this matching index. (4) We took a closer look at the top candidates to exclude the CSPs with too low expression, included the CSPs that are in top 100 CSPs presenting on the surface of developing PNs (Li et al., 2020), and checked expression pattern across developmental stages (at ~24h APF and ~48h APF). Finally, we came up with top 5 candidates to perform the RNAi screen—*Ptp10D*, *sdk*, *sideV*, *PlexB*, and *CCKLR-17D3*. We showed their inverse expression pattern (Extended Data Fig. 2a and Figure R1). (5) As summarized in the Extended Data Table 1, we used the VA1d-ORN driver and the VA1d/DA1-PN driver to perform the *in vivo* screen. Only *Ptp10D* and *sideV* showed consistent phenotypes in both VA1d-ORNs and VA1d-PNs. Follow-up experiments suggest *Ptp10D* genetically interacts with Toll2.

We also checked the expression of some known binding molecule pairs (Özkan et al., 2013); none of them except Kirre–Hbs/Sns shows good inverse expression pattern.

Figure R1: Inverse expression of Toll2 with top candidate genes

Besides Ptp10D, these plots show other four top candidate genes at 24–30h APF that show inverse expression with Toll2 among the 10 partner PN-ORN pairs from our single-cell RNA sequencing data (Xie et al., 2021 and McLaughlin et al., 2021). CPM: counts per million reads.

Also, at present, the fly brain-wide expression patterns are relegated to supplemental videos. There is space in the final figure to include a still image from each of the videos, highlighting some interesting aspect(s) of differential gene expression, before focusing on the optic lobe layers. The central brain expression patterns are very suggestive and will be of interest to a wider readership, ideally prompting future experiments by experts in those other neuropils.

We appreciate the suggestions. We previously focused on the optic lobe because its stereotypic layer structures make it easy to compare between flies. To better visualize the differential expression of CSP pairs, we generated new genetic reagents (*Toll2>V5>Flag*, *Fili>V5>Flag*, *Kirre>V5>Flag*) during the revision, which allowed us to visualize distributions of repulsive CSP pairs with double labeling in the same brain. The results suggest all CSPs have differential expression during development. We highlighted the differential expression of three CSP pairs in three example brain regions in the new Fig. 6d.

2. Make the narrative as clear and accessible as possible.

The abstract is clear and succinct (although it omits the novel point that two of the three CSP pairs do not appear to interact directly). However, I identified several possible errors in the figures and accompanying text. There are also places where more information could be provided for clarity.

We appreciate the suggestions from the reviewer on the errors and unclear descriptions. We have modified our manuscript accordingly as we mentioned below.

Fig. 1j-l What is the significance of cyan vs red? Should the 2nd column in 1k be red for Kek1 instead of cyan, or should the label read "Fili in PNs"? Should the 2nd column in 1l be darker blue for Sns instead of cyan?

Cyan and red are labeled according to the colors in the representative images in Fig. 1c, d and Extended Data Fig. 2c, with cyan for CSPs sending repulsive signal and red for CSPs receiving repulsive signal based on the functional studies in Figs. 2–5. As the reviewer pointed out, the 2nd column in 1f should be red. We labeled Sns in Fig. 1j for blue to

distinguish it from Hbs. To minimize confusion, we added description for the color code in the figure legend in the revised manuscript.

Fig. 1 Legend: c-i Should this read "red in c, e, g" and "cyan in d, f, h, i"?)

We thank the reviewer for catching this error. To streamline the manuscript, we have moved some of the example images into supplementary material. We have corrected the labeling error in our revised manuscript.

Extended Data Figure 3

3a VA1d ORN and VA1d PN panels appear to have been swapped

We swapped it in the revised manuscript.

Main text (last paragraph of Trans-cellular interactions of Toll2 and Ptp10D): "Simultaneous overexpression of Toll2 in VA1d-ORNs and knockdown of Ptp10D in VA1v-PNs suppressed the VA1d-PN dendrite phenotypes caused by Toll2 overexpression alone" - I believe that this should be "knockdown of Ptp10D in VA1d-PNs"

We have corrected this typo in the revised manuscript.

Fig. 6b Please clarify: Is the ssRNAseq data from ORNs, PNs, both, etc?

We mentioned PN types in the legend. To make it clearer, we added "in PNs" in the figure legend in the revised manuscript.

Please add a gene label, either to each file name or to the beginning of each supplemental movie. Perhaps this was done originally as indicated in the accompanying legend, but at present there is nothing to indicate which gene's expression is shown.

There might have been some technical issues that eliminated the label on the file names. In the revised version, we added the gene names onto the top right corner of the video files themselves to avoid this problem.

Fig. 6c Could different styles of arrowheads or other symbols be used to draw the reader's attention to the specific differences in expression that you believe could prove significant for targeting?

Great suggestion. We have changed the color of some of the arrowheads in the revised manuscript and pointed out a specific example in the legend.

Extended Data Figure 6. The authors cite other studies that support direct binding of Kirre to Hbs/Sns, but they did not detect direct binding between Fili and Kek1 or between Toll2 and Ptp10, so the biochemical basis for their repulsion remains unknown. Please show a positive control for this method in your hands, for example the binding of Kirre to Hbs/Sns.

As the reviewer suggested, we included a positive control of Sns binding to Kirre for the live brain staining in the new Extended Data Fig. 9a. In *kirre>HA>Myc* pupal brains, Sns-Fc-V5 showed binding in the antennal lobe with strong and specific pattern, which is not seen in the *Kirre* mutant pupal brains.

In addition, we performed the additional live staining experiment in the wing disc dissected from 3rd instar larvae, where *engrailed (en)-GAL4* was used to drive ectopic expression of candidate CSPs in the posterior compartment of the wing disc. When we applied Sns-Fc-V5 for live staining in the *en-GAL4 > UAS-Kirre-HA* wing disc, we observed binding of Sns-Fc-V5 specifically to the posterior compartment (Extended Data Fig. 9d), which serve as a positive control. For Fili-Kek1 and Toll2-Ptp10D, we did not detect binding in this wing disc live staining assay (Extended Data Fig. 9e, f).

Finally, we now include an orthogonal in vitro binding assay designed to detect interactions between cell surface receptors (Özkan et al., 2013), as mentioned above for Reviewer 1. The assay detects binding of bait ectodomains, captured on solid support, to oligomerized prey ectodomains, which are tagged with a chromogenic enzyme. With this assay, we recapitulated the Sns–Kirre, Hbs–Kirre, and Rst-Rst interactions, as well as the previously described Kek1–EGFR interaction, as positive controls. However, we observed no binding between Fili and Kek1, or between Ptp10D and Toll2. These new results were added to the Extended Data Fig. 8.

Discussion: "site choice" not "cite choice" in 2nd paragraph

Corrected. Thanks.